# Convergence for score-based generative modeling with polynomial complexity

**Holden Lee**
Department of Applied Mathematics and Statistics
Johns Hopkins University
`hlee283@jhu.edu`

**Jianfeng Lu**
Department of Mathematics
Duke University
`jianfeng@math.duke.edu`

**Yixin Tan**
Department of Mathematics
Duke University
`yixin.tan@duke.edu`

## Abstract

Score-based generative modeling (SGM) is a highly successful approach for learning a probability distribution from data and generating further samples. We prove the first polynomial convergence guarantees for the core mechanic behind SGM: drawing samples from a probability density $p$ given a score estimate (an estimate of $\nabla \ln p$) that is accurate in $L^2(p)$. Compared to previous works, we do not incur error that grows exponentially in time or that suffers from a curse of dimensionality. Our guarantee works for any smooth distribution and depends polynomially on its log-Sobolev constant. Using our guarantee, we give a theoretical analysis of score-based generative modeling, which transforms white-noise input into samples from a learned data distribution given score estimates at different noise scales. Our analysis gives theoretical grounding to the observation that an annealed procedure is required in practice to generate good samples, as our proof depends essentially on using annealing to obtain a warm start at each step. Moreover, we show that a predictor-corrector algorithm gives better convergence than using either portion alone.

## 1 Introduction

A key task in machine learning is to learn a probability distribution from data, in a way that allows efficient generation of additional samples from the learned distribution. Score-based generative modeling (SGM) is one empirically successful approach that *implicitly* learns the probability distribution by learning how to transform white noise into the data distribution, and gives state-of-the-art performance for generating images and audio [SE19; Dat+19; Gra+19; SE20; Son+20b; Men+21; Son+21b; Son+21a; Jin+22]. It also yields a conditional generation process for inverse problems [DN21]. The basic idea behind score-based generative modeling is to first estimate the score function from data [Son+20a] and then to sample the distribution based on the learned score function. Other approaches for generative modeling include generative adversarial networks (GANs) [Goo+14; ACB17], normalizing flows [DSB16], variational autoencoders [KW19], and energy-based models [ZML16]. While score-based generative modeling has achieved great success, its theoretical analysis is still lacking and is the focus of our work.

36th Conference on Neural Information Processing Systems (NeurIPS 2022).

## 1.1 Background

**General framework.** The *score function* of a distribution $P$ with density $p$ is defined as the gradient of the log-pdf, $\nabla \ln p$. Its significance arises from the fact that knowing the score function allows running a variety of sampling algorithms, based on discretizations of stochastic differential equations (SDE's), to sample from $p$. SGM consists of two steps: first, learning an estimate of the score function for a sequence of "noisy" versions of the data distribution $P_{\text{data}}$, and second, using the score function in lieu of the gradient of the log-pdf in the chosen sampling algorithm. We now describe each of these steps more precisely.

First, a method of adding noise to the data distribution is fixed; this takes the form of evolving a (forward) stochastic differential equation (SDE) starting from the data distribution. We fix a sequence of noise levels $\sigma_1 < \cdots < \sigma_N$. For $\sigma \in \{\sigma_1, \ldots, \sigma_N\}$, let the resulting distributions be $P_{\sigma^2}$ and the distributions conditional on the starting data point be $P_{\sigma^2}(\cdot|x)$. Typically, $\sigma_1$ is chosen so that $P_{\sigma_1^2} \approx P_{\text{data}}$ and $P_{\sigma_N^2}$ is close to some "prior" distribution that is easy to sample from, such as $N(0, \sigma_N^2 I_d)$. While the score $\nabla \ln p_{\sigma^2}$ cannot be estimated directly, it turns out that a de-noising objective that is equivalent to the score-matching objective can be calculated [SE19]. This de-noising objective can be estimated from samples $(X, \widetilde{X})$ where $\widetilde{X} \sim P_{\sigma^2}(\cdot|x)$. The objective is represented and optimized within an expressive function class, typically neural networks, to obtain a $L^2$-estimate of the score, that is, $s_\theta(x, \sigma^2)$ such that

$$\mathbb{E}_{x \sim P_{\sigma^2}}[\|s_\theta(x, \sigma^2) - \nabla \ln p_{\sigma^2}(x)\|^2] \tag{1}$$

is small.

The reason we estimate the score function $\nabla \ln p_{\sigma^2}$ is that there are a variety of sampling algorithms—based on simulating SDE's—that can sample from $p$ given access to $\nabla \ln p$, including Langevin Monte Carlo and Hamiltonian Monte Carlo. The second step is then to use the estimated score function $s_\theta(x, t)$ in lieu of the exact gradient in the sampling algorithm to successively obtain samples from $p_{\sigma_N^2}, \ldots, p_{\sigma_1^2}$. This sequence interpolates smoothly between the prior distribution (e.g., $N(0, \sigma_N^2 I_d)$) and the data distribution $P_{\text{data}}$; such an "annealing" or "homotopy" method is required in practice to generate good samples [Son+20b].

**Examples of SGM's.** There have been several instantiations of this general approach. [SE19] add gaussian noise to the data and then use Langevin diffusion at a discrete set of noise levels $\sigma_N > \cdots > \sigma_1$ as the sampling algorithm. [Son+20b] take the continuous perspective and consider a more general framework, where the forward process can be any reasonable SDE. Then a natural *reverse SDE* evolves the final distribution $p_{\sigma_N^2}$ back to the data distribution; this process can be simulated with the estimated score. They consider methods based on two different SDE's: score-matching Langevin diffusion (SMLD) based on adding Gaussian noise and denosing diffusion probabilistic models (DDPM) [Soh+15; HJA20], based on the Ornstein-Uhlenbeck process. Note that a difference with MCMC-based methods is that these SDE's are evolved for a fixed amount of time, rather than until convergence. However, they can be combined with MCMC-based methods such as Langevin diffusion in the *predictor-corrector* approach for improved convergence. [DVK21] include Hamiltonian dynamics: they augment the state space with a velocity variable and consider a critically-damped version of the Ornstein-Uhlenbeck process. Finally, we note the work of [De +21], who introduce the Diffusion Schrödinger Bridge method to learn a diffusion that more quickly transforms the prior into the data distribution.

We will give a general analysis framework for SGM's that applies to the algorithms in both [SE19] and [Son+20b].

## 1.2 Prior work and challenges for theory

Although the literature on convergence for Langevin Monte Carlo [DM17; CB18; Che+18; Dal17; DK19; MMS20; EHZ21] and related sampling algorithms is extensive, prior works mainly consider the case of exact or stochastic gradients. In contrast, by the structure of the loss function (1), the score function learned in SGM is only accurate in $L^2(p)$. This poses a significant challenge for analysis, as the stationary distribution of Langevin diffusion with $L^2(p)$-accurate gradient can be arbitrarily far from $p$ (see Appendix D). Hence, any analysis must be utilizing the short/medium-

term convergence, while overcoming the potential issue of long-term behavior of convergence to an incorrect distribution.

[BMR20] give the first theoretical analysis of SGM, and in particular, Langevin Monte Carlo with $L^2(p)$-accurate gradients. First, they show using uniform generalization bounds that optimizing the de-noising autoencoder (DAE) objective does in fact give a $L^2(p)$-accurate score function, with sample complexity depending on the complexity of the function class. They analyze convergence of LMC in Wasserstein distance. However, the error they obtain (Theorem 13) only decreases as $\varepsilon^{1/d}$ where $\varepsilon$ is the accuracy of the score estimate—so it suffers from the curse of dimensionality—and increases exponentially in the time that the process is run, the dimension, and the smoothness of the distribution, as in ODE/SDE discretization arguments that do not depend on contractivity.

[De +21] give an analysis for [Son+20b] in TV distance that requires a $L^\infty$-accurate score function and depends exponentially on the amount of time the reverse SDE is run. Although exponential dependence is bad in general, it is mollified using their Diffusion Schrödinger Bridge (DSB) approach, as it allows running for a shorter, fixed amount of time, before the forward SDE converges to the prior distribution. However, this supposes that a good solution can be found for the DSB problem, and theoretical guarantees may be difficult to obtain.

We overcome the challenges of analysis with a $L^2(p)$-accurate gradient, and give the first analysis with only polynomial dependence on running time, dimension, and smoothness of the distribution, with rates that are a fixed power of $\varepsilon$. Our convergence result is in TV distance. We assume only smoothness conditions and a bounded log-Sobolev constant of the data distribution, a weaker condition than the dissipativity condition required by [BMR20]. We introduce a general framework for analysis of sampling algorithms given $L^2$-accurate gradients (score function) based on constructing a "bad set" with small measure and showing convergence of the discretized process conditioned on not hitting the bad set. We use our framework to give an end-to-end analysis for both the algorithms in [SE19] and [Son+20b], and illuminate the relative performance of different methods in practice.

### 1.3 Notation and organization

Through out the paper, $p(x) \propto e^{-V(x)}$ denotes the target distribution in $\mathbb{R}^d$ and $V : \mathbb{R}^d \to \mathbb{R}$ is referred to as the potential. We abuse notation by identifying a measure with its density when context allows. We write $a \wedge b := \min\{a, b\}$ and $a \vee b := \max\{a, b\}$. We use $a = O(b)$ or $b = \Omega(a)$ to indicate that $a \leq Cb$ for a universal constant $C > 0$. Also, we write $a = \Theta(b)$ if there are universal constants $c' > c > 0$ such that $cb \leq a \leq cb$, and the notation $\tilde{O}(\cdot)$ means it hides polylog factors in the parameters. Definite integrals without limits are taken over $\mathbb{R}^d$.

In Section 2 we explain our main results for Langevin Monte Carlo with $L^2(p)$-accurate score estimate and use it to derive convergence bounds for the annealed LMC method of [SE19]. In Section 3, we give our main results for the predictor-corrector algorithms of [Son+20b] based on simulating reverse SDE's. Our proofs are based on a common framework which we introduce in Section 4. Full proofs are in the appendix.

## 2 Results for Langevin dynamics with estimated score

Let $p(x) \propto e^{-V(x)}$ be a probability density on $\mathbb{R}^d$ such that $V$ is $C^1$. Langevin diffusion with stationary distribution $p$ is the stochastic process defined by the SDE

$$dx_t = -\nabla V(x_t) \, dt + \sqrt{2} \, dw_t,$$

where $w_t$ is a standard Brownian Motion in $\mathbb{R}^d$. The rate of convergence to $p$ in $\chi^2$ and KL divergences are given by the Poincaré and log-Sobolev constants of $p$, respectively; see Section E.1. To obtain the Langevin Monte Carlo (LMC) algorithm, we take the Euler-Murayama discretization of the SDE. We define LMC with score estimate $s(x) \approx -\nabla V(x)$ and step size $h$ by

$$x_{(k+1)h} = x_{kh} + h \cdot s(x_{kh}) + \sqrt{2h} \cdot \xi_{kh}, \text{ where } \xi_{kh} \sim N(0, I_d). \qquad \text{(LMC-SE)}$$

We make the following assumptions on the density $p$ and the score estimate $s$, which we will use throughout this paper.

**Assumption 1.** *$p$ is a probability density on $\mathbb{R}^d$ such that the following hold.*

1. $\ln p$ is $C^1$ and $L$-smooth, that is, $\nabla \ln p$ is $L$-Lipschitz. We assume $L \geq 1$.

2. $p$ satisfies a log-Sobolev inequality with constant $C_{\mathrm{LS}}$. We assume $C_{\mathrm{LS}} \geq 1$.

3. (Moments) $\|\mathbb{E}_p x\| \leq M_1$ and $\mathbb{E}_p \|x\|^2 \leq M_2$.

We note that the uniform Lipschitzness assumption (1) helps ensure a unique strong solution to the Langevin diffusion, as in [BMR20]. One special case where one can prove Lipschitzness for all $t$ is when $p_0$ is strongly log-concave [Lee+21, Lemma 28]. Although satisfying a log-Sobolev inequality (3) is a significant assumption, it is standard for analysis of Langevin Monte Carlo [VW19]. It is much weaker than assumptions in previous works [BMR20], including log-concave distributions and distributions satisfying strong dissipativity, and is stable under bounded perturbations. See Section E.1 for background on functional inequalities.

**Assumption 2.** *Let $p$ be a given probability density on $\mathbb{R}^d$ such that $\ln p$ is $C^1$. The score estimate $s : \mathbb{R}^d \to \mathbb{R}^d$ satisfies the following.*

1. *$s$ is a $C^1$ function that is $L_s$-Lipschitz. We assume $L_s \geq 1$.*

2. *The error in the score estimate is bounded in $L^2$:*
$$\|\nabla \ln p - s\|_{L^2(p)}^2 = \mathbb{E}_p[\|\nabla \ln p(x) - s(x)\|^2] \leq \varepsilon^2.$$

## 2.1 Langevin with $L^2$-accurate score estimate

Our first main result gives an error bound between the sampled distribution and $p$, assuming $L^2$-accurate score function estimate.

**Theorem 2.1** (LMC with $L^2$-accurate score estimate)**.** *Let $p : \mathbb{R}^d \to \mathbb{R}$ be a probability density satisfying Assumption 1(1, 2) with $L \geq 1$ and $s : \mathbb{R}^d \to \mathbb{R}^d$ be a score estimate satisfying Assumption 2(2). Consider the accuracy requirement in* TV *and $\chi^2$: $0 < \varepsilon_{\mathrm{TV}} < 1$, $0 < \varepsilon_\chi < 1$, and suppose furthermore the starting distribution satisfies $\chi^2(p_0\|p) \leq K_\chi^2$. Then if*

$$\varepsilon = O\left( \frac{\varepsilon_{\mathrm{TV}} \varepsilon_\chi^3}{dL^2 C_{\mathrm{LS}}^{5/2}(\ln(2K_\chi/\varepsilon_\chi^2) \vee K_\chi)} \right), \tag{2}$$

*then running* (LMC-SE) *with score estimate $s$, step size $h = \Theta\left(\frac{\varepsilon_\chi^2}{dL^2 C_{\mathrm{LS}}}\right)$, and time $T = \Theta\left(C_{\mathrm{LS}} \ln\left(\frac{2K_\chi}{\varepsilon_\chi^2}\right)\right)$ results in a distribution $p_T$ such that $p_T$ is $\varepsilon_{\mathrm{TV}}$-far in* TV *distance from a distribution $\overline{p}_T$, where $\overline{p}_T$ satisfies $\chi^2(\overline{p}_T\|p) \leq \varepsilon_\chi^2$. In particular, taking $\varepsilon_\chi = \varepsilon_{\mathrm{TV}}$, we have the error guarantee that* $\mathrm{TV}(p_T, p) \leq 2\varepsilon_{\mathrm{TV}}$.

Note that the error bound is only achieved when running LMC for a moderate time; this is consistent with the fact that the stationary distribution of LMC with a $L^2$-score estimate can be arbitrarily far from $p$. Note also that we need a warm start in $\chi^2$-divergence: to obtain fixed errors $\varepsilon_{\mathrm{TV}}, \varepsilon_\chi$, the required accuracy for the score estimate is inversely proportional to $K_\chi$. Intuitively, we must suffer from such a dependence because if the starting distribution is very far away, then there is no guarantee that $\|\nabla \ln p(x_t) - s(x_t)\|^2$ is small on average during the sampling algorithm. Finally, although we can state a result purely in terms of TV distance, we need this more precise formulation to prove a result for annealed Langevin dynamics.

## 2.2 Annealed Langevin dynamics with estimated score

In light of the warm start requirement in Theorem 2.1, we typically cannot directly sample from $p_{\mathrm{data}}$ or its approximation. Hence, [SE19] proposed using annealed Langevin dynamics: consider a sequence of noise levels $\sigma_N > \cdots > \sigma_1 \approx 0$ giving rise to a sequence of distributions $p_{\sigma_N^2}, \ldots, p_{\sigma_1^2} \approx p_{\mathrm{data}}$, where $p_{\sigma^2} = p * \varphi_{\sigma^2}$, $\varphi_{\sigma^2}$ being the density of $N(0, \sigma^2 I_d)$. For large enough $\sigma_N, \varphi_{\sigma_N^2} \approx p_{\sigma_N^2}$ provides a warm start to $p_{\sigma_N^2}$. We then successively run LMC using score estimates for $p_{\sigma_k^2}$, with the approximate sample for $p_{\sigma_k^2}$ giving a warm start for $p_{\sigma_{k-1}^2}$. We obtain the following algorithm and error estimate.

---
**Algorithm 1** Annealed Langevin dynamics with estimated score [SE19]
---
INPUT: Noise levels $0 \leq \sigma_1 < \ldots < \sigma_M$; score function estimates $s(\cdot, \sigma_m)$ (estimates of $\nabla \ln(p * \varphi_{\sigma_m^2})$), step sizes $h_m$, and number of steps $N_m$ for $1 \leq m \leq M$.
Draw $x^{(M+1)} \sim N(0, \sigma_M^2 I_d)$.
**for** $m$ from $M$ to 1 **do**
  Starting from $x_0^{(m)} = x^{(m+1)}$, run (LMC-SE) with $s(x, \sigma_m)$ and step size $h_m$ for $N_m$ steps, and let the final sample be $x^{(m)}$.
**end for**
OUTPUT: Return $x^{(1)}$, approximate sample from $p * \varphi_{\sigma_1^2}$.
---

**Theorem 2.2** (Annealed LMC with $L^2$-accurate score estimate). *Let $p : \mathbb{R}^d \to \mathbb{R}$ be a probability density satisfying Assumption 1 for $M_1 = O(d)$, and let $p_{\sigma^2} := p * \varphi_{\sigma^2}$. Suppose furthermore that $\nabla \ln p_{\sigma^2}$ is $L$-Lipschitz for every $\sigma \geq 0$. Given $\sigma_{\min} > 0$, there exists a sequence $\sigma_{\min} = \sigma_1 < \cdots < \sigma_M$ with $M = O\left( \sqrt{d} \log\left( \frac{d C_{\mathrm{LS}}}{\sigma_{\min}^2} \right) \right)$ such that for each $m$, if*

$$\left\| \nabla \ln(p_{\sigma_m^2}) - s(\cdot, \sigma_m^2) \right\|_{L^2(p_{\sigma_m^2})}^2 = \mathbb{E}_{p_{\sigma_m^2}} \left[ \left\| \nabla \ln p_{\sigma_m^2}(x) - s(x, \sigma_m^2) \right\|^2 \right] \leq \varepsilon^2.$$

$$\text{with } \varepsilon := \widetilde{O}\left( \frac{\varepsilon_{\mathrm{TV}}^{4.5}}{d^{3.25} L^2 C_{\mathrm{LS}}^{2.5}} \right) \tag{3}$$

*then $x^{(1)}$ is a sample from a distribution $q$ such that $\mathrm{TV}(q, p_{\sigma_1^2}) \leq \varepsilon_{\mathrm{TV}}$.*

Note that we assume a score estimate with error $\varepsilon$ at all noise scales; this corresponds to using an objective function that is a maximum of the score-matching objective over all noise levels, rather than an average over all noise levels as more commonly used in practice. However, these two losses are at most a factor of $M$ apart.

The proof shows that the noise levels $\sigma_k$ can be chosen as a geometric sequence, which matches the choice used in practice [SE20]. The additional dependence on $d$ and $\varepsilon_{\mathrm{TV}}$ in Theorem 2.2 compared to Theorem 2.1 comes from requiring a sequence of $\widetilde{O}(\sqrt{d})$ noise levels and an additional factor in $\chi^2$-divergence we suffer at the beginning of each level $m$. In the next section, we will find that using a reverse SDE to evolve the samples between the noise levels—called a *predictor* step—will improve the rate and time complexity.

## 3 Results for reverse SDE's with estimated score

To improve the empirical performance of score-based generative modeling, [Son+20b] consider a general framework where noise is injected into a data distribution $p_{\mathrm{data}}$ via a forward SDE,

$$d\tilde{x}_t = f(\tilde{x}_t, t)\, dt + g(t)\, dw_t, \ \ t \in [0, T],$$

where $\widetilde{x}_0 \sim \widetilde{p}_0 := p_{\mathrm{data}}$. Let $\widetilde{p}_t$ denote the distribution of $\widetilde{x}_t$ ($\widetilde{p}_t$ is used instead of $p_t$ to distinguish with the Gaussian-convolved distribution used in Annealed Langevin dynamics as in §2.2). Remarkably, $\widetilde{x}_t$ also satisfies a reverse-time SDE,

$$d\tilde{x}_t = [f(\tilde{x}_t, t) - g(t)^2 \nabla \ln \tilde{p}_t(\tilde{x}_t)]dt + g(t)\, d\tilde{w}_t, \ t \in [0, T], \tag{4}$$

where $\tilde{w}_t$ is a backward Brownian Motion [And82]. By carefully choosing $f$ and $g$, we can expect that $\tilde{p}_T$ is approximately equal to some prior distribution $\tilde{q}_T$ (e.g., a centered Gaussian) which we can accurately sample from. Then we hope that starting with some $\tilde{y}_T \sim p_{\mathrm{prior}} = \tilde{q}_T \approx \tilde{p}_T$ and running the reverse-time process, we will get a good sample $\tilde{y}_0 \sim \tilde{q}_0 \approx p_{\mathrm{data}}$.

The case where $f \equiv 0$ and $g \equiv 1$ recovers the simple case of convolving with a Gaussian as used in §2.2; note, however that the reverse-time SDE differs from Langevin diffusion in having a larger (and time-varying) drift relative to the diffusion. [Son+20b] highlight the following two special cases. We will focus on DDPM while noting that our analysis applies more generically.

SMLD **Score-matching Langevin diffusion**: $f \equiv 0$. In this case, $\widetilde{p}_t = \widetilde{p}_0 * \varphi_{\int_0^t g(s)^2\, ds}$, so [Son+20b] call this a variance-exploding (VE) SDE. As is common for annealing-based

algorithms, [SE19; Son+20b] suggest choosing an exponential schedule, so that $g(t) = ab^t$ for constants $a, b$. We take $p_{\text{prior}} = N(0, \int_0^T g(s)^2 \, ds \cdot I_d)$.

**DDPM** **Denoising diffusion probabilistic modeling**: $f(x, t) = -\frac{1}{2} g(t)^2 x$. This is an Ornstein-Uhlenbeck process with time rescaling, $\widetilde{p}_t = M_{-\frac{1}{2} \int_0^t g(s)^2 \, ds} \sharp \widetilde{p}_0 * \varphi_{1 - e^{-\int_0^t g(s)^2 \, ds}}$, where $M_\alpha(x) = \alpha x$. [Son+20b] call this a variance-preserving (VP) SDE, as the variance converges towards $I_d$. Because it displays exponential convergence towards $N(0, I_d)$, it can be run for a smaller amount of normalized time $\int_0^t g(s)^2 \, ds$. [Son+20b] suggest the choice $g(t) = \sqrt{b + \alpha t}$. We take $p_{\text{prior}} = N(0, (1 - e^{-\int_0^t g(s)^2 \, ds}) I_d) \approx N(0, I_d)$.

To obtain an algorithm, we consider the following discretization and approximation of (4); note that in all cases of interest the integrals can be analytically evaluated. We reverse time so that $t$ corresponds to $T - t$ of the forward process. As we are free to rescale time in the SDE, we assume without loss of generality that the step sizes are constant. The predictor step is

$$z_{(k+1)h} = z_{kh} - \int_{kh}^{(k+1)h} \left[ f(z_{kh}, T - t) - g(T - t)^2 \cdot s(z_{kh}, T - kh) \right] dt$$

$$+ \int_{kh}^{(k+1)h} g(T - t) \, dw_t, \quad \text{(P)}$$

where $\int_{kh}^{(k+1)h} g(T - t) \, dw_t$ is distributed as $N(0, \int_{kh}^{(k+1)h} g(T - t)^2 \, dt \cdot I_d)$. Following [Son+20b], we call these predictor steps as the samples aim to track the distributions $\widetilde{p}_{T-kh}$. Note that we flip the time. For simplicity of presentation, we consider the case $g \equiv 1$. We note that although the choice of the schedule does matter in practice, what really matters in our theoretical analysis is the integral $\int_0^t g(s)^2 ds$. This means that different choices of $g$ are related by only a rescaling of time, i.e., for different $g$ and $\tilde{g}$, we can always choose total times $T$ and $\tilde{T}$, such that $\int_0^T g(s)^2 ds = \int_0^{\tilde{T}} \tilde{g}(s)^2 ds$. While it seems that choosing large $g(t)$ could reduce the total time $T$, in our analysis (e.g., Lemma C.15) we need the time step-size $h$ to be $O(1/g(T)^2)$ and hence the total computational cost, which is roughly $O(T/h)$, does not change significantly.

**Theorem 3.1** (Predictor with $L^2$-accurate score estimate, DDPM). *Let $p_{\text{data}} : \mathbb{R}^d \to \mathbb{R}$ be a probability density satisfying Assumption 1 with $M_2 = O(d)$, and let $\widetilde{p}_t$ be the distribution resulting from evolving the forward SDE according to DDPM with $g \equiv 1$. Suppose furthermore that $\nabla \ln \widetilde{p}_t$ is $L$-Lipschitz for every $t \geq 0$, and that each $s(\cdot, t)$ satisfies Assumption 2. Then if*

$$\varepsilon = O\left( \frac{\varepsilon_{\text{TV}}^4}{(C_{\text{LS}} + d) C_{\text{LS}}^{5/2} (L \vee L_s)^2 (\ln(C_{\text{LS}} d) \vee C_{\text{LS}} \ln(1/\varepsilon_{\text{TV}}^2))} \right),$$

*running* (P) *starting from $p_{\text{prior}}$ for time $T = \Theta\left( \ln(C_{\text{LS}} d) \vee C_{\text{LS}} \ln\left( \frac{1}{\varepsilon_{\text{TV}}} \right) \right)$ and step size $h = \Theta\left( \frac{\varepsilon_{\text{TV}}^2}{C_{\text{LS}}(C_{\text{LS}} + d)(L \vee L_s)^2} \right)$ results in a distribution $q_T$ so that $\text{TV}(q_T, p_{\text{data}}) \leq \varepsilon_{\text{TV}}$.*

A more precise statement of the Theorem can be found in the Appendix. Although we state our theorem for DDPM, we describe in Appendix C how it can be adapted to other SDE's like SMLD and the sub-VP SDE; the primary SDE-dependent bound we need is a bound on $\nabla \ln \frac{\widetilde{p}_t}{\widetilde{p}_{t+h}}$. Because the predictor is tracking a changing distribution $p_t$, we incur more error terms and worse dependence on parameters $(C_{\text{LS}}, L)$ than in LMC (Theorem 2.1). Motivated by this, we intersperse the predictor steps with LMC steps—called *corrector* steps in this context—to give additional time for the process to mix, resulting in improved dependence on parameters.

**Theorem 3.2** (Predictor-corrector with $L^2$-accurate score estimate). *Keep the setup of Theorem 3.1. Then for $\varepsilon_{\text{TV}}^3 = O\left( \frac{1}{(1 + L_s/L)^2 (1 + C_{\text{LS}}/d)(\ln(C_{\text{LS}} d) \vee C_{\text{LS}})} \right)$, if*

$$\varepsilon = O\left( \frac{\varepsilon_{\text{TV}}^4}{d L^2 C_{\text{LS}}^{5/2} \ln(1/\varepsilon_\chi^2)} \right), \quad (5)$$

*then Algorithm 2 with appropriate choices of $T = \Theta\left( \ln(C_{\text{LS}} d) \vee C_{\text{LS}} \log\left( \frac{1}{\varepsilon_{\text{TV}}} \right) \right)$, $N_m$, corrector step sizes $h_m$ and predictor step size $h$, produces a sample from a distribution $q_T$ such that $\text{TV}(q_T, p_{\text{data}}) < \varepsilon_{\text{TV}}$.*

---
**Algorithm 2** Predictor-corrector method with estimated score [Son+20b]
---
INPUT: Time $T$, predictor step size $h$; number of corrector steps $N_m$ per predictor step, corrector step sizes $h_m$

Draw $z_0 \sim p_{\text{prior}}$ from the prior distribution.

**for** $m$ from 1 to $T/h$ **do**

    (Predictor) Take a step of (P) to obtain $z_{mh}$ from $z_{(m-1)h}$, with $f, g$ as in SMLD or DDPM.

    (Corrector) Starting from $z_{mh,0} := z_{mh}$, run (LMC-SE) with $s(z, T - mh)$ and step size $h_m$ for $N$ steps, and let $z_{mh} \leftarrow z_{mh,N}$.

**end for**

OUTPUT: Return $z_T$, approximate sample from $p_{\text{data}}$.

---

The assumption on $\varepsilon_{\text{TV}}$ is for convenience in stating our bound. In comparison to using the predictor step alone (Theorem 3.1), note that in the bound on $\varepsilon$, we obtain the improved rate of the corrector step as in Theorem 2.1; this is because the predictor step only needs to track the actual distribution in $\chi^2$-divergence with error $O(1)$, and the final corrector steps are responsible for decreasing the error to $\varepsilon_{\text{TV}}$. In comparison to the Annealed Langevin sampler (Algorithm 1, Theorem 2.2), which can be viewed as using the corrector step alone, adding a predictor step provides a better warm start for the distribution at the next smaller noise level, resulting in better dependence on parameters. Thus the predictor-corrector algorithm combines the strengths of the predictor and corrector steps. For real-world data, it can be challenging to estimate TV-distance between distributions given only samples, and hence difficult to check consistency with empirical observations. However, our claim that using a corrector can improve the convergence rate of DDPM/SMLD is consistent with the simulation results in Section 4.2 of [Son+20b].

## 4   Theoretical framework and proof sketches

The main idea of our analysis framework is to convert a $L^2$ error guarantee to a $L^\infty$ error guarantee by excluding a bad set, formalized in the following theorem.

**Theorem 4.1.** *Let $(\Omega, \mathcal{F}, \mathbb{P})$ be a probability space and $\{\mathcal{F}_n\}$ be a filtration of the sigma field $\mathcal{F}$. Suppose $X_n \sim p_n$, $Z_n \sim q_n$, and $\overline{Z}_n \sim \overline{q}_n$ are $\mathcal{F}_n$-adapted random processes taking values in $\Omega$, and $B_n \subseteq \Omega$ are sets such that the following hold for every $n \in \mathbb{N}_0$.*

1. *If $Z_k \in B_k^c$ for all $0 \le k \le n - 1$, then $Z_n = \overline{Z}_n$.*

2. *$\chi^2(\overline{q}_n || p_n) \le D_n^2$.*

3. *$\mathbb{P}(X_n \in B_n) \le \delta_n$.*

*Then the following hold.*

$$\text{TV}(q_n, \overline{q}_n) \le \sum_{k=0}^{n-1} (D_k^2 + 1)^{1/2} \delta_k^{1/2} \qquad \text{TV}(p_n, q_n) \le D_n + \sum_{k=0}^{n-1} (D_k^2 + 1)^{1/2} \delta_k^{1/2} \qquad (6)$$

For our setting, we will take the "bad sets" $B_n$ to be the set of $x$ where $\|s_\theta(x) - \nabla \ln p\|$ is large, $q_n$ to be the discretized process with estimated score, and $\overline{q}_n$ to be the discretized process with estimated score except in $B_n$ where the error is large. Because $\overline{q}_n$ uses an $L^\infty$-accurate score estimate, we can use existing techniques for analyzing Langevin Monte Carlo [VW19; EHZ21; Che+21] to bound $\chi^2(\overline{q}_n || p_n)$.

*Proof.* We bound using condition 1 and Cauchy-Schwarz:

$$\mathbb{P}\left(Z_n \ne \overline{Z}_n\right) \le \mathbb{P}\left(\bigcup_{k=1}^{n-1} \{Z_k \in B_k\}\right) \le \sum_{k=0}^{n-1} \mathbb{P}\left(Z_k \in B_k\right) = \sum_{k=0}^{n-1} \mathbb{E}_{q_k} \mathbb{1}_{B_k}$$

$$\le \sum_{k=0}^{n-1} \left(\mathbb{E}_{p_k}\left(\frac{q_k}{p_k}\right)^2\right)^{1/2} \left(\mathbb{E}_{p_k} \mathbb{1}_{B_k}\right)^{1/2} = \sum_{k=0}^{n-1} (D_k^2 + 1)^{1/2} \delta_k^{1/2}.$$

The second inequality then follows from the triangle inequality and Cauchy-Schwarz:

$$\mathrm{TV}(p_n, q_n) \leq \mathrm{TV}(p_n, \bar{q}_n) + \mathrm{TV}(\bar{q}_n, q_n)$$

$$\leq \sqrt{\chi^2(\bar{q}_n\|p_n)} + \mathrm{TV}(\bar{q}_n, q_n) \leq D_n + \sum_{k=0}^{n-1}(D_k^2 + 1)^{1/2}\delta_k^{1/2}. \qquad \square$$

It now remains to give $\chi^2$ convergence bounds under $L^\infty$-accurate score estimate. The following theorem may be of independent interest.

**Theorem 4.2** (LMC under $L^\infty$ bound on gradient error). *Let $p : \mathbb{R}^d \to \mathbb{R}$ be a probability density satisfying Assumption 1(1, 2) and $s : \mathbb{R}^d \to \mathbb{R}^d$ be a score estimate $s$ with error bounded in $L^\infty$: for some $\varepsilon_1 \leq \sqrt{\frac{1}{48C_{\mathrm{LS}}}}$,*

$$\|\nabla \ln p - s\|_\infty = \max_{x \in \mathbb{R}^d} \|\nabla \ln p(x) - s(x)\| \leq \varepsilon_1.$$

*Let $N \in \mathbb{N}_0$ and $0 < h \leq \frac{1}{4392 d C_{\mathrm{LS}} L^2}$, and assume $L \geq 1$. Let $q_{nh}$ denote the $n$th iterate of LMC with step size $h$ score estimate $s$. Then*

$$\chi^2(q_{(k+1)h}\|p) \leq \exp\left(-\frac{h}{4C_{\mathrm{LS}}}\right)\chi^2(q_{kh}\|p) + 170dL^2h^2 + 5\varepsilon_1^2 h$$

*and*

$$\chi^2(q_{Nh}\|p) \leq \exp\left(-\frac{Nh}{4C_{\mathrm{LS}}}\right)\chi^2(q_0\|p) + 680dL^2hC_{\mathrm{LS}} + 20\varepsilon_1^2 C_{\mathrm{LS}} \leq \exp\left(-\frac{Nh}{4C_{\mathrm{LS}}}\right)\chi^2(q_0\|p) + 1$$

Following [Che+21], we prove this by first defining a continuous-time interpolation $q_t$ of the discrete process, and then deriving a differential inequality for $\chi^2(q_t\|p)$ using the log-Sobolev inequality for $p$. Compared to [Che+21], we incur an extra error term arising from the inaccurate gradient.

This allows us to sketch the proof of Theorem 2.1; a complete proof is in Section B.

*Proof sketch of Theorem 2.1.* We first define the bad set where the error in the score estimate is large,

$$B := \{\|\nabla \ln p(x) - s(x)\| > \varepsilon_1\}$$

for some $\varepsilon_1$ to be chosen. Then by Chebyshev's inequality, $P(B) \leq \left(\frac{\varepsilon}{\varepsilon_1}\right)^2 =: \delta$. Let $\bar{q}_{nh}$ be the discretized process, but where the score estimate is set to be equal to $\nabla \ln p$ on $B$; note it agrees with $q_{nh}$ as long as it has not hit $B$. Because $\bar{q}_{nh}$ uses a score estimate that has $L^\infty$-error $\varepsilon_1$, Theorem 4.2 gives a bound for $\chi^2(\bar{q}_{Nh}\|p)$. Then Theorem 4.1 gives

$$\mathrm{TV}(q_{nh}, \bar{q}_{nh}) \leq \sum_{k=0}^{n-1}(\chi^2(\bar{q}_{kh}\|p) + 1)^{1/2}P(B)^{1/2} \leq \sum_{k=0}^{n-1}\left(\exp\left(-\frac{kh}{8C_{\mathrm{LS}}}\right)\chi^2(q_0\|p)^{1/2} + 1\right)\delta^{1/2}$$

The theorem then follows from choosing parameters so that $\chi^2(\bar{q}_T\|p) \leq \varepsilon_\chi^2$ and $\mathrm{TV}(q_T, \bar{q}_T) \leq \varepsilon_{\mathrm{TV}}$. $\square$

We remark that the main inefficiency in the proof comes from the use of Chebyshev's inequality, and a $L^p$ bound on the error for $p > 2$ will improve the bound.

*Proof sketch of Theorem 2.2.* Choosing the sequence $\sigma_1 < \cdots < \sigma_M$ to be geometric with ratio $1 + \frac{1}{\sqrt{d}}$ ensures that the $\chi^2$-divergence between successive distributions $p_{\sigma_m^2}$ is $O(1)$. Then, choosing $\sigma_M^2 = \Omega(C_{\mathrm{LS}}d)$ ensures we have a warm start for the highest noise level: $\chi^2(p_{\mathrm{prior}}\|p_{\sigma_M^2}) = O(1)$. This uses $O\left(\sqrt{d}\log\left(\frac{dC_{\mathrm{LS}}}{\sigma_{\min}^2}\right)\right)$ noise levels. Chebyshev's inequality can be used to show that the distribution of the final sample $x^{(m)}$ for $p_{\sigma_m^2}$ is $O(\varepsilon_{\mathrm{TV}}/M)$ close to a distribution that is $O(M/\varepsilon_{\mathrm{TV}})$ in $\chi^2$-divergence from $p_{\sigma_{m+1}^2}$. This gives the warm start parameter $K_\chi = (M/\varepsilon_{\mathrm{TV}})^{1/2}$; substituting into Theorem 2.1 then gives the required bound for $\varepsilon$. Note that the TV errors accrued from each level add to $O(\varepsilon_{\mathrm{TV}})$. $\square$

To analyze the predictor-based algorithms, we also first prove convergence bounds under $L^\infty$-accurate score estimate.

**Theorem 4.3** (Predictor steps under $L^\infty$ bound on score estimate, DDPM). *Let $p : \mathbb{R}^d \to \mathbb{R}$ be a probability density satisfying Assumption 1 and $s(\cdot, t) : \mathbb{R}^d \to \mathbb{R}^d$ be a score estimate $s$ with error bounded in $L^\infty$ for each $t \in [0, T]$:*

$$\|\nabla \ln p - s(\cdot, t)\|_\infty = \max_{x \in \mathbb{R}^d} \|\nabla \ln \widetilde{p}_t(x) - s(x, t)\|] \le \varepsilon_1.$$

*Consider DDPM with $g \equiv 1$, $T \ge 1 \vee \ln(C_{\mathrm{LS}} d)$, and $h = O\left(\frac{1}{C_{\mathrm{LS}}(d + C_{\mathrm{LS}})(L \vee L_s)^2}\right)$. (Recall that $p_{kh}$ and $q_{kh}$ are the $k$-th iterate of LMC with step size $h$ and true/estimated score respectively.) Then*

$$\chi^2(q_{(k+1)h} \| p_{(k+1)h}) \le \chi^2(q_{kh} \| p_{kh}) e^{\left(-\frac{1}{8C_{\mathrm{LS}}} + 8\varepsilon_1^2\right)h} + O(\varepsilon_1^2 h + (L_s^2 + L^2 d)h^2)$$

*and if $\varepsilon_1 < \frac{1}{128 C_{\mathrm{LS}}}$,*

$$\chi^2(q_{Nh} \| p_{Nh}) \le e^{-\frac{Nh}{16 C_{\mathrm{LS}}}} \chi^2(q_0 \| p_0) + O\left(C_{\mathrm{LS}}\left(\varepsilon_1^2 + (L_s^2 + L^2 d)h\right)\right).$$

*Moreover, for $q_0 = p_{\mathrm{prior}}$, $\chi^2(q_0 \| p_0) \le e^{-T/2} C_{\mathrm{LS}} d$.*

We give a more precise statement in Section C. Note that unlike the case for LMC as in Theorem 4.2, the base density $p_t$ is also evolving in time, which produces additional error terms and necessitates a more involved analysis. The additional error terms can be bounded using the Donsker-Varadhan variational principle, concentration for distributions satisfying LSI, and error bounds between $p_t$ and $p_{t+h}$ for small $h$.

Here, we only state the result about DDPM, which has better bounds than SMLD (when $g \equiv 1$) because both the forward and backwards processes exhibit better mixing properties: the warm start improves exponentially rather than inversely with $T$, and the log-Sobolev constant is uniformly bounded by that of $p_{\mathrm{data}}$ rather than increasing. However, the analysis in Section C can be directly applied to SMLD and other models as well. We also note there is a sense in which DDPM and SMLD are equivalent under a rescaling in time and space (see discussion in Section C.2).

Note that the choice of $h$ is necessary for exponential decay of error; as if $h$ is not small enough, we would get an exponential growing instead of decaying factor in the one-step error (See Section C for details). Such an $h$ may however still be a suitable choice when used in conjunction with a corrector step. Moreover, as $\varepsilon_1 \to 0$, with appropriate choice of $T$ and $h$, $q_{Nh}$ and $p_{Nh}$ can be made arbitrarily close.

Theorem 3.1 now follows from the $L^\infty$ result (Theorem 4.3) in the same way that Theorem 2.1 follows from Theorem 4.2.

To prove Theorem 3.2, it suffices to run the corrector steps only at the lowest noise level, that is, set $N_m = 0$ for $1 \le m < T/h$, although we note that interleaving the predictor and corrector steps does empirically help with mixing. The proof follows from using the predictor and the corrector theorems in series: first apply Theorem 3.1 with $\varepsilon_\chi = O(1)$ to show that the predictor results a warm start $p_{\mathrm{data}}$, then use Theorem 2.1 to show the corrector reduces the error to the desired $\varepsilon_{\mathrm{TV}}$.

## 5  Conclusion

We introduced a general framework to analyze SDE-based sampling algorithms given a $L^2$-error score estimate, and used it to obtain the first convergence bounds for several score-based generative models with polynomial complexity in all parameters. Our analysis can potentially be adapted to other SDE's and sampling algorithms beyond Langevin Monte Carlo. There is also room for improving our analysis to better use smoothing properties of the SDE's and compare different choices of the diffusion speed $g$.

We present several interesting further directions to explore. In addition to extending the analysis to other SGM's and comparing their theoretical performance (relative to each other as well as other approaches to generative modeling), we propose the following.

**Analysis for multimodal distributions.** Our assumption of a bounded log-Sobolev constant essentially limits the analysis to distributions that are close to unimodal. However, SGM's are empirically successful at modeling multimodal distributions [SE19], and in fact perform better with multimodal distributions than other approaches such as GAN's. Can we analyze the convergence for simple multimodal distributions, such as a mixture of distributions each with bounded log-Sobolev constant? Positive results on sampling from multimodal distributions such as [GLR18] suggest this is possible, as the sequence of noised distributions is natural for annealing and tempering methods (see [GLR18, Remark 7.2]).

**Weakening conditions on the score estimate.** The assumption that we have a score estimate that is $O(1)$-accurate in $L^2$, although weaker than the usual assumptions for theoretical analysis, is in fact still a strong condition in practice that seems unlikely to be satisfied (and difficult to check) when learning complex distributions such as distributions of images. What would a reasonable weaker condition be, and in what sense can we still obtain reasonable samples?

**Guarantees for learning the score function.** Our analysis assumes a $L^2$-estimate of the score function is given, but the question remains of when we can find such an estimate. What natural conditions on distributions allow their score functions to be learned by a neural network? Various works have considered the representability of data distributions by diffusion-like processes [TR19], but the questions of optimization and generalization appear more challenging.

### Acknowledgements

We thank Andrej Risteski for helpful conversations. This work was done in part while HL was visiting the Simons Institute for the Theory of Computing. The work was supported in part by National Science Foundation via awards DMS-2012286 and CCF-1934964 (Duke Tripods).

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
