# A Computations

We start the proofs by collecting some preliminary results. In the following, we will consider the SDE

$$dx_t = f(x,t)\, dt + G(t)\, dw_t \tag{7}$$

and the interpolation of the discretization of an approximation

$$dz_t = \widehat{f}(z_{t_-}, t)\, dt + G(t)\, dw_t \tag{8}$$

when $t \geq t_-$. Let $P_t$ and $Q_t$ denote the law of $x_t$ and $z_t$, respectively. We will take $t_- = kh$ and $t \in [kh, (k+1)h)$. We will assume that $f, \widehat{f}, G$ are continuous and the functions $f(\cdot, t)$, $\widehat{f}(\cdot, t)$ are uniformly Lipschitz for each $t \in [kh, (k+1)h]$.

In this section, we will make some computations that will be used in both Sections B and C. First, we derive how the density evolves in time.

**Lemma A.1.** *Let $Q_t$ denote the law of the interpolated process* (8)*. Then*

$$\frac{\partial q_t(z)}{\partial t} = \nabla \cdot \left[ -q_t(z)\mathbb{E}\left[ \widehat{f}(z_{t_-}, t) | z_t = z \right] + \frac{G(t)G(t)^\top}{2} \nabla q_t(z) \right].$$

*Proof.* Let $q_{t|t_-}$ denote the distribution of $z_t$ conditioned on $z_{t_-}$. Then the Fokker-Planck equation gives

$$\frac{\partial q_{t|t_-}(z|z_{t_-})}{\partial t} = -\nabla q_{t|t_-}(z|z_{t_-}) \cdot [\widehat{f}(z_{t_-}, t)] + \frac{G(t)G(t)^\top}{2} \Delta q_{t|t_-}(z|z_{t_-})$$

Taking expectation with respect to $z_{t_-}$ we get

$$\frac{\partial q_t(z)}{\partial t} = \nabla \cdot \int -q_{t|t_-}(z)\widehat{f}(y,t)q_{t_-}(y)dy + \nabla \cdot \left[ \frac{G(t)G(t)^\top}{2} \nabla \int q_{t|t_-}(z|y)q_{t_-}(y)dy \right]$$

$$= \nabla \cdot q_t(z) \int \left[ -\widehat{f}(y,t)q_{k|t}(y|z)dy + \frac{G(t)G(t)^\top}{2} \nabla \int q_{t_-|t}(y|z)q_t(z)dy \right].$$

Note that for fixed $z$, $\int q_{t_-|t}(y|z)dy = 1$. Hence

$$\frac{\partial q_t(z)}{\partial t} = \nabla \cdot \left[ -q_t(z)\mathbb{E}\left[ \widehat{f}(z_{t_-}, t) | z_t = z \right] + \frac{G(t)G(t)^\top}{2} \nabla q_t(z) \right].$$

$\square$

We now use Lemma A.1 to compute how the $\chi^2$-divergence between the approximate and exact densities changes. The following generalizes the calculation of [EHZ21] in the case where $x_t$ is a non-stationary stochastic process. For simplicity of notation, from now on, we wil consider the case $G(t)$ being a scalar.

**Lemma A.2.** *Let $P_t$ and $Q_t$ be the laws of* (7) *and* (8) *for $G(t) = g(t)I_d$. Then*

$$\frac{\partial}{\partial t}\chi^2(q_t \| p_t) = -g(t)^2 \mathscr{E}_{p_t}\left( \frac{q_t}{p_t} \right) + 2\mathbb{E}\left[ \left\langle \widehat{f}(z_{t_-}, t) - f(z_t, t), \nabla \frac{q_t(z_t)}{p_t(z_t)} \right\rangle \right].$$

*Proof.* The Fokker-Planck equation gives

$$\frac{\partial p_t(x)}{\partial t} = \nabla \cdot \left[ -f(x,t)p_t(x) + \frac{g(t)^2}{2} \nabla p_t(x) \right].$$

We have

$$\frac{d}{dt}\chi^2(q_t \| p_t) = \frac{d}{dt} \int \frac{q_t(x)^2}{p_t(x)} dx = \int \left[ 2\frac{\partial q_t(x)}{\partial t} \frac{q_t(x)}{p_t(x)} - \frac{\partial p_t(x)}{\partial t} \frac{q_t(x)^2}{p_t(x)^2} \right] dx.$$

For the first term, by Lemma A.1,

$$2 \int \frac{\partial q_t(x)}{\partial t} \frac{q_t(x)}{p_t(x)} dx = 2 \int \nabla \cdot \left[ -q_t(x) \mathbb{E} \left[ \widehat{f}(z_0, t) | z_t = x \right] + \frac{g(t)^2}{2} \nabla q_t(x) \right] \cdot \frac{q_t(x)}{p_t(x)} dx$$

$$= 2 \int q_t(x) \left\langle \mathbb{E} \left[ \widehat{f}(z_0, t) | z_t = x \right], \nabla \frac{q_t(x)}{p_t(x)} \right\rangle dx - g(t)^2 \int \left\langle \nabla q_t(x), \nabla \frac{q_t(x)}{p_t(x)} \right\rangle dx. \tag{9}$$

For the second term, using integration by parts,

$$-\int \frac{\partial p_t(x)}{\partial t} \frac{q_t(x)^2}{p_t(x)^2} dx = \int \nabla \cdot \left[ f(x,t) p_t(x) - \frac{g(t)^2}{2} \nabla p_t(x) \right] \cdot \frac{q_t(x)^2}{p_t(x)^2} dx$$

$$= \int -f(x,t) p_t(x) \nabla \frac{q_t(x)^2}{p_t(x)^2} + \frac{g(t)^2}{2} \left\langle \nabla p_t(x), \nabla \frac{q_t(x)^2}{p_t(x)^2} \right\rangle dx$$

$$= -2 \int q_t(x) \left\langle f(x,t), \nabla \frac{q_t(x)}{p_t(x)} \right\rangle dx$$

$$+ g(t)^2 \int \frac{q_t(x)}{p_t(x)} \left\langle \nabla p_t(x), \nabla \frac{q_t(x)}{p_t(x)} \right\rangle dx. \tag{10}$$

Note that

$$\int \left\langle \nabla q_t(x), \nabla \frac{q_t(x)}{p_t(x)} \right\rangle - \frac{q_t(x)}{p_t(x)} \left\langle \nabla p_t(x), \nabla \frac{q_t(x)}{p_t(x)} \right\rangle = \int \left\langle \nabla \frac{q_t(x)}{p_t(x)}, \nabla \frac{q_t(x)}{p_t(x)} \right\rangle q_t(x) \, dx = \mathscr{E}_{p_t} \left( \frac{q_t}{p_t} \right).$$

Combining (9) and (10),

$$\frac{d}{dt} \chi^2(q_t \| p_t) = -g(t)^2 \mathscr{E}_{p_t} \left( \frac{q_t}{p_t} \right) + 2 \int q_t(x) \left\langle \mathbb{E} \left[ \widehat{f}(z_{t_-}, t) - f(x,t) | z_t = x \right], \nabla \frac{q_t(x)}{p_t(x)} \right\rangle dx$$

$$= -g(t)^2 \mathscr{E}_{p_t} \left( \frac{q_t}{p_t} \right) + 2 \mathbb{E} \left[ \left\langle \widehat{f}(z_{t_-}, t) - f(z_t, t), \nabla \frac{q_t(z_t)}{p_t(z_t)} \right\rangle \right].$$

$\square$

Finally, we will make good use of the following lemma to bound the second term in Lemma A.2.

**Lemma A.3** (cf. [EHZ21, Lemma 1]). *Let* $\phi_t(x) = \frac{q_t(x)}{p_t(x)}$ *and* $\psi_t(x) = \phi_t(x)/\mathbb{E}_{p_t} \phi_t^2$. *For any* $c$ *and any* $\mathbb{R}^d$-*valued random variable* $u$, *we have*

$$\mathbb{E} \left[ \left\langle u, \nabla \frac{q_t(z_t)}{p_t(z_t)} \right\rangle \right] \leq \mathbb{E} \left[ \|u\| \left\| \nabla \frac{q_t(z_t)}{p_t(z_t)} \right\| \right] \leq C \cdot \mathbb{E}_{p_t} \phi_t^2 \cdot \mathbb{E} \left[ \|u\|^2 \psi_t(z_t) \right] + \frac{1}{4C} \mathscr{E}_{p_t} \left( \frac{q_t}{p_t} \right).$$

*Proof.* Note that $\mathbb{E} \psi_t(z_t) = 1$ and the normalizing factor is $\mathbb{E}_{p_t} \phi_t^2 = \chi^2(q_t \| p_t) + 1$. By Young's inequality,

$$\mathbb{E} \left[ \left\langle u, \nabla \frac{q_t(z_t)}{p_t(z_t)} \right\rangle \right] = \mathbb{E} \left[ \left\langle u \sqrt{\frac{q_t(z_t)}{p_t(z_t)}}, \sqrt{\frac{p_t(z_t)}{q_t(z_t)}} \nabla \frac{q_t(z_t)}{p_t(z_t)} \right\rangle \right]$$

$$\leq C \mathbb{E} \left[ \|u\|^2 \frac{q_t(z_t)}{p_t(z_t)} \right] + \frac{1}{4C} \mathbb{E}_{p_t} \left[ \left\| \nabla \frac{q_t(x)}{p_t(x)} \right\|^2 \right]$$

$$= C \mathbb{E} \left[ \|u\|^2 \frac{q_t(z_t)}{p_t(z_t)} \right] + \frac{1}{4C} \mathscr{E}_{p_t} \left( \frac{q_t}{p_t} \right).$$

$\square$

# B  Analysis for LMC

Let $p$ be the probability density we wish to sample from. Suppose that we have an estimate $s$ of the score $\nabla \ln p$. Our main theorem says that if the $L^2$ error $\mathbb{E}_p \| \nabla \ln p - s \|^2$ is small enough, then running LMC with $s$ for an *appropriate* time results in a density that is close in *TV distance* to a density that is close in $\chi^2$-*divergence* to $p$. The following is a more precise version of Theorem 2.1.

**Theorem B.1** (LMC with $L^2$-accurate score estimate)**.** *Let $p : \mathbb{R}^d \to \mathbb{R}$ be a probability density satisfying Assumption 1 with $L \geq 1$ and $s : \mathbb{R}^d \to \mathbb{R}^d$ be a score estimate satisfying Assumption 2(2). Consider the accuracy requirement in* TV *and $\chi^2$: $0 < \varepsilon_{\mathrm{TV}} < 1$, $0 < \varepsilon_\chi < 1$, and suppose furthermore the starting distribution satisfies $\chi^2(p_0||p) \leq K_\chi^2$. Then if*

$$\varepsilon \leq \frac{\varepsilon_{\mathrm{TV}}\varepsilon_\chi^3}{174080\sqrt{5}dL^2C_{\mathrm{LS}}^{5/2}(C_T \ln(2K_\chi/\varepsilon_\chi^2) \vee 2K_\chi)},$$

*then running* (LMC-SE) *with score estimate $s$ and step size $h = \frac{\varepsilon_\chi^2}{2720dL^2C_{\mathrm{LS}}}$ for any time $T \in [T_{\min}, C_T T_{\min}]$, where $T_{\min} = 4C_{\mathrm{LS}} \ln\left(\frac{2K_\chi}{\varepsilon_\chi^2}\right)$, results in a distribution $p_T$ such that $p_T$ is $\varepsilon_{\mathrm{TV}}$-far in TV distance from a distribution $\bar{p}_T$, where $\bar{p}_T$ satisfies $\chi^2(\bar{p}_T||p) \leq \varepsilon_\chi^2$. In particular, taking $\varepsilon_\chi = \varepsilon_{\mathrm{TV}}$, we have the error guarantee that $\mathrm{TV}(p_T, p) = 2\varepsilon_{\mathrm{TV}}$.*

The main difficulty is that the stationary distribution of LMC using the score estimate may be arbitrarily far from $p$, even if the $L^2$ error of the score estimate is bounded. (See Section D.) Thus, a long-time convergence result does not hold, and an upper bound on $T$ is required, as in the theorem statement.

We instead proceed by showing that *conditioned on not hitting a bad set*, if we run LMC using $s$, the $\chi^2$-divergence to the stationary distribution will decrease. This means that the closeness of the overall distribution (in TV distance, say) will decrease in the short term, despite it will increase in the long term, as the probability of hitting the bad set increases. This does not contradict the fact that the stationary distribution is different from $p$. By running for a moderate amount of time (just enough for mixing), we can ensure that the probability of hitting the bad set is small, so that the resulting distribution is close to $p$. Note that we state the theorem with a $C_T$ parameter to allow a range of times that we can run LMC for.

More precisely, we prove Theorem B.1 in two steps.

**LMC under $L^\infty$ gradient error (Section B.1, Theorem 4.2).** First, consider a simpler problem: proving a bound for $\chi^2$ divergence for LMC with score estimate $s$, when $\|s - \nabla \ln p\|$ is bounded everywhere, not just on average. For this, we follow the argument in [Che+21] for showing convergence of LMC in Rényi divergence; this also gives a bound in $\chi^2$-divergence. We define an interpolation of the discrete process and derive a upper bound for the derivative of Rényi divergence, $\partial_t \mathcal{R}_q(q_t||p)$, using the log-Sobolev inequality for $p$. In the original proof, the error comes from the discretization error; here we have an additional error term coming from an inaccurate gradient, which is bounded by assumption. Note that a $L^2$ bound on $\nabla f - s$ is insufficient to give an upper bound, as we need to bound $\mathbb{E}_{q_t\psi_t}[\|\nabla f - s\|^2]$ for a different measure $q_t\psi_t$ that we do not have good control over. An $L^\infty$ bound works regardless of the measure.

**Defining a bad set and bounding the hitting time (Section B.2).** The idea is now to reduce to the case of $L^\infty$ error by defining the "bad set" $B$ to be the set where $\|s - \nabla f\| \geq \varepsilon_1$, where $\varepsilon \ll \varepsilon_1 \ll 1$. This set has small measure by Chebyshev's inequality. Away from the bad set, Theorem 4.2 applies; it then suffices to bound the probability of hitting $B$. Technically, we define a coupling with a hypothetical process where the $L^\infty$ error is always bounded, and note that the processes disagree exactly when it hits $B$; this is the source of the TV error.

We consider the probability of being in $B$ at times $0, h, 2h, \ldots$. we note that Theorem B.1 bounds the $\chi^2$-divergence of this hypothetical process $X_t$ at time $t$ to $p$. If the distribution were actually $p$, then the probability $X_t \in B'$ is exactly $p(B')$; we expect the probability to be small even if the distribution is close to $p$. Indeed, by Cauchy-Schwarz, we can bound the probability $X \in B$ in terms of $P(B)$ and $\chi^2(q_t||p)$; this bound is given in Theorem 4.1. Note that the eventual bound depends on $\chi^2(q_t||p)$, so we have to assume a warm start, that is, a reasonable bound on $\chi^2(q_0||p)$.

## B.1 LMC under $L^\infty$ gradient error

The following gives a long-time convergence bound for LMC with inaccurate gradient, with error bounded in $L^\infty$; this may be of independent interest.

**Theorem 4.2** (LMC under $L^\infty$ bound on gradient error). *Let $p : \mathbb{R}^d \to \mathbb{R}$ be a probability density satisfying Assumption 1(1, 2) and $s : \mathbb{R}^d \to \mathbb{R}^d$ be a score estimate $s$ with error bounded in $L^\infty$: for some $\varepsilon_1 \leq \sqrt{\frac{1}{48 C_{\mathrm{LS}}}}$,*

$$\|\nabla \ln p - s\|_\infty = \max_{x \in \mathbb{R}^d} \|\nabla \ln p(x) - s(x)\|] \leq \varepsilon_1.$$

*Let $N \in \mathbb{N}_0$ and $0 < h \leq \frac{1}{4392 d C_{\mathrm{LS}} L^2}$, and assume $L \geq 1$. Let $q_{nh}$ denote the $n$th iterate of LMC with step size $h$ score estimate $s$. Then*

$$\chi^2(q_{(k+1)h}\|p) \leq \exp\left(-\frac{h}{4 C_{\mathrm{LS}}}\right) \chi^2(q_{kh}\|p) + 170 d L^2 h^2 + 5\varepsilon_1^2 h$$

*and*

$$\chi^2(q_{Nh}\|p) \leq \exp\left(-\frac{Nh}{4 C_{\mathrm{LS}}}\right) \chi^2(q_0\|p) + 680 d L^2 h C_{\mathrm{LS}} + 20 \varepsilon_1^2 C_{\mathrm{LS}} \leq \exp\left(-\frac{Nh}{4 C_{\mathrm{LS}}}\right) \chi^2(q_0\|p) + 1$$

Following [Che+21], convergence in Rényi divergence can also be derived; we only consider $\chi^2$-divergence because we will need a warm start in $\chi^2$-divergence for our application. Note that by letting $N \to \infty$ and $h \to 0$, we obtain the following.

**Corollary B.2.** *Keep the assumptions in Theorem 4.2. The stationary distribution $q$ of Langevin diffusion with score estimate $s$ satisfies*

$$\chi^2(q\|p) \leq 20 C_{\mathrm{LS}} \varepsilon_1^2.$$

*Proof of Theorem 4.2.* We follow the proof of [Che+21, Theorem 4], except that we work with the $\chi^2$ divergence directly, rather than the Rényi divergence, and have an extra term from the inaccurate gradient (17). Given $t \geq 0$, let $t_- = h \lfloor \frac{t}{h} \rfloor$. Define the interpolated process by

$$dz_t = s(z_{t_-}) \, dt + \sqrt{2} \, dw_t, \tag{11}$$

and let $q_t$ denote the distribution of $X_t$ at time $t$, when $X_0 \sim q_0$.

By Lemma A.2,

$$\frac{\partial}{\partial t} \chi^2(q_t\|p) = -2\mathscr{E}_p\left(\frac{q_t}{p}\right) + 2\mathbb{E}\left[\left\langle s(z_{t_-}) - \nabla \ln p(z_t), \nabla \frac{q_t(z_t)}{p(z_t)}\right\rangle\right]. \tag{12}$$

By the proof of Theorem 4 in [Che+21],

$$\left\|\nabla \ln p(x_t) - \nabla \ln p(x_{t_-})\right\|^2 \leq 9 L^2 (t - t_-)^2 \left\|\nabla \ln p(x_t)\right\|^2 + 6 L^2 \left\|B_t - B_{t_-}\right\|^2.$$

Then

$$\left\|s(z_{t_-}) - \nabla \ln p(z_t)\right\|^2 \leq 2\left\|\nabla \ln p(z_{t_-}) - \nabla \ln p(z_t)\right\|^2 + 2\left\|s(z_{t_-}) - \nabla \ln p(z_{t_-})\right\|^2$$
$$\leq 18 L^2 (t - t_-)^2 \|\nabla \ln p(z_t)\|^2 + 12 L^2 \left\|B_t - B_{t_-}\right\|^2 + 2\varepsilon_1^2. \tag{13}$$

Let $\phi_t := q_t/p$ and $\psi_t := \frac{\phi_t}{\mathbb{E}_p(\phi_t^2)}$. By Lemma A.3,

$$2\mathbb{E}\left[\left\langle s(z_{t_-}) - \nabla \ln p(z_t), \nabla \frac{q_t(z_t)}{p(z_t)}\right\rangle\right] \leq 2\mathbb{E}_p \phi_t^2 \cdot \mathbb{E}\left[\left\|s(z_{t_-}) - \nabla \ln p(z_t)\right\|^2 \psi_t(z_t)\right] + \frac{1}{2}\mathscr{E}_p\left(\frac{q_t}{p}\right)$$
$$\leq A_1 + A_2 + A_3 + \frac{1}{2}\mathscr{E}_p(\phi_t) \tag{14}$$

where $A_1, A_2, A_3$ are obtained by substituting in the 3 terms in (13), and given in (15), (16), and (17). Let $V(x) = -\ln p(x)$. We consider each term in turn.

$$A_1 := 36 L^2 (t - t_-)^2 \mathbb{E}_p \phi_t^2 \cdot \mathbb{E}\left[\|\nabla V(z_t)\|^2 \psi_t(z_t)\right] \tag{15}$$
$$\leq 36 L^2 (t - t_-)^2 \mathbb{E}_p \phi_t^2 \cdot \left(\frac{4\mathscr{E}_p(\phi_t)}{\mathbb{E}_p \phi_t^2} + 2dL\right) \qquad \text{by [Che+21, Lemma 16]}$$
$$\leq \frac{1}{2}\mathscr{E}_p(\phi_t) + 72 d L^3 (t - t_-)^2 (\chi^2(q_t\|p) + 1)$$

when $h^2 \leq \frac{1}{288L^2}$. By [Che+21, p. 15]

$$A_2 := 24L^2 \mathbb{E}_p \phi_t^2 \cdot \mathbb{E} \left[ \left\| B_t - B_{t_-} \right\|^2 \psi_t(z_t) \right] \tag{16}$$

$$\leq 24L^2 \mathbb{E}_p \phi_t^2 \cdot \left( 14dL^2(t - t_-) + 32hC_{\mathrm{LS}} \frac{\mathscr{E}_p(\phi_t)}{\mathbb{E}_p \phi_t^2} \right)$$

$$\leq 336dL^2(t - t_-)(\chi^2(q_t \| p) + 1) + \frac{1}{2} \mathscr{E}_p(\phi_t)$$

when $h \leq \frac{1}{1536L^2 C_{\mathrm{LS}}}$. Finally,

$$A_3 := 4\varepsilon_1^2 \mathbb{E}_p \phi_t^2 = 4\varepsilon_1^2 (\chi^2(q_t \| p) + 1). \tag{17}$$

Combining (12), (14), (15), (16), and (17) gives

$$\frac{\partial}{\partial t} \chi^2(q_t \| p) \leq -\frac{1}{2} \mathscr{E}_p(\phi_t) + (\chi^2(q_t \| p) + 1)(72dL^3(t - t_-)^2 + 336dL^2(t - t_-) + 4\varepsilon_1^2)$$

$$\leq -\frac{1}{2C_{\mathrm{LS}}} \chi^2(q_t \| p) + (\chi^2(q_t \| p) + 1)(72dL^3(t - t_-)^2 + 336L^2 d(t - t_-) + 4\varepsilon_1^2)$$

$$\leq -\frac{1}{4C_{\mathrm{LS}}} \chi^2(q_t \| p) + (72dL^3(t - t_-)^2 + 336dL^2(t - t_-) + 4\varepsilon_1^2)$$

if $h \leq \left( \frac{1}{12 \cdot 72dL^3 C_{\mathrm{LS}}} \right)^{1/2} \wedge \frac{1}{12 \cdot 336dC_{\mathrm{LS}}}$ and $\varepsilon_1 \leq \left( \frac{1}{48C_{\mathrm{LS}}} \right)^{1/2}$. Then for $t \in [kh, (k+1)h)$,

$$\frac{\partial}{\partial t} \left( \chi^2(q_t \| p) \exp\left( \frac{t - t_-}{4C_{\mathrm{LS}}} \right) \right) = \exp\left( \frac{t - t_-}{4C_{\mathrm{LS}}} \right) (72dL^3(t - t_-)^2 + 336dL^2(t - t_-) + 4\varepsilon_1^2)$$

$$\leq 73dL^3(t - t_-)^2 + 337dL^2(t - t_-) + 5\varepsilon_1^2.$$

Integrating over $t \in [kh, (k+1)h)$ gives

$$\chi^2(q_{(k+1)h} \| p) \leq \exp\left( -\frac{h}{4C_{\mathrm{LS}}} \right) \chi^2(q_{kh} \| p) + \frac{73}{3} dL^3 h^3 + \frac{337}{2} dL^2 h^2 + 5\varepsilon_1^2 h$$

$$\leq \exp\left( -\frac{h}{4C_{\mathrm{LS}}} \right) \chi^2(q_{kh} \| p) + 170dL^2 h^2 + 5\varepsilon_1^2 h$$

using $h \leq \frac{1}{12\sqrt{2}L}$. Unfolding the recurrence and summing the geometric series gives

$$\chi^2(q_{kh} \| p) \leq \exp\left( -\frac{kh}{4C_{\mathrm{LS}}} \right) \chi^2(q_0 \| p) + 680dL^2 h C_{\mathrm{LS}} + 20\varepsilon_1^2 C_{\mathrm{LS}}$$

$$\leq \exp\left( -\frac{kh}{4C_{\mathrm{LS}}} \right) \chi^2(q_0 \| p) + 1$$

when $h \leq \frac{1}{1360dL^2 C_{\mathrm{LS}}}$ and $\varepsilon_1^2 \leq \frac{1}{40C_{\mathrm{LS}}}$. We can check that the given condition on $h$ and the fact that $LC_{\mathrm{LS}} \geq 1$ (Lemma E.5) imply all the required inequalities on $h$. $\qquad\square$

### B.2 Proof of Theorem B.1

*Proof of Theorem B.1.* We first define the bad set where the error in the score estimate is large,

$$B := \{ \| \nabla \ln p(x) - s(x) \| > \varepsilon_1 \}$$

for some $\varepsilon_1$ to be chosen.

Given $t \geq 0$, let $t_- = h \left\lfloor \frac{t}{h} \right\rfloor$. Given a bad set $B$, define the interpolated process by

$$d\bar{z}_t = b(\bar{z}_{t_-}) \, dt + \sqrt{2} \, dw_t, \tag{18}$$

$$\text{where } b(z) = \begin{cases} s(z), & z \notin B \\ \nabla \ln p(z), & z \in B \end{cases}.$$

In other words, run LMC using the score estimate as long as the point is in the good set at the previous discretization step, and otherwise use the actual gradient $\nabla \ln p$. Let $\bar{q}_t$ denote the distribution of $\bar{z}_t$ when $\bar{z}_0 \sim q_0$; note that $q_{nh}$ is the distribution resulting from running LMC with estimate $b$ for $n$ steps and step size $h$. Note that this auxiliary process is defined only for purposes of analysis; it cannot be used for practical algorithm as we do not have access to $\nabla f$.

We can couple this process with LMC using $s$ so that as long as $X_t$ does not hit $B$, the processes agree, thus satisfying condition 1 of Theorem 4.1.

Then by Chebyshev's inequality,

$$P(B) \leq \left(\frac{\varepsilon}{\varepsilon_1}\right)^2 =: \delta.$$

Let $T = Nh$. Then by Theorem 4.2,

$$\chi^2(\widetilde{q}_{kh}||p) \leq \exp\left(-\frac{kh}{4C_{\mathrm{LS}}}\right)\chi^2(q_0||p) + 680dL^2hC_{\mathrm{LS}} + 20\varepsilon_1^2 C_{\mathrm{LS}} \leq \exp\left(-\frac{kh}{4C_{\mathrm{LS}}}\right)\chi^2(q_0||p) + 1.$$

For this to be bounded by $\varepsilon_\chi^2$, it suffices for the terms to be bounded by $\frac{\varepsilon_\chi^2}{2}, \frac{\varepsilon_\chi^2}{4}, \frac{\varepsilon_\chi^2}{4}$; this is implied by

$$T \geq 4C_{\mathrm{LS}}\ln\left(\frac{2K_\chi}{\varepsilon_\chi^2}\right) =: T_{\min}$$

$$h = \frac{\varepsilon_\chi^2}{4392dL^2C_{\mathrm{LS}}}$$

$$\varepsilon_1 = \frac{\varepsilon_\chi}{4\sqrt{5C_{\mathrm{LS}}}}.$$

(We choose $h$ so that the condition in Theorem 4.2 is satisfied; note $\varepsilon_\chi \leq 1$.) By Theorem 4.1,

$$
\begin{aligned}
\mathrm{TV}(q_{Nh}, \bar{q}_{Nh}) &\leq \sum_{k=0}^{N-1}(1 + \chi^2(q_{kh}||p))^{1/2}P(B)^{1/2} \\
&\leq \left(\sum_{k=0}^{N-1}\exp\left(-\frac{kh}{8C_{\mathrm{LS}}}\right)\chi^2(q_0||p)^{1/2} + 2\right)\delta^{1/2} \\
&\leq \left(\left(\sum_{k=0}^{\infty}\exp\left(-\frac{kh}{8C_{\mathrm{LS}}}\right)K_\chi\right) + 2N\right)\frac{\varepsilon}{\varepsilon_1} \\
&\leq \frac{\varepsilon}{\varepsilon_1}\left(\frac{16C_{\mathrm{LS}}}{h}K_\chi + 2N\right).
\end{aligned}
$$

In order for this to be $\leq \varepsilon_{\mathrm{TV}}$, it suffices for

$$\varepsilon \leq \varepsilon_1\varepsilon_{\mathrm{TV}}\left(\frac{1}{4N} \wedge \frac{h}{32C_{\mathrm{LS}}K_\chi}\right).$$

Supposing that we run for time $T$ where $T_{\min} \leq T \leq C_T T_{\min}$, we have that $N = \frac{T}{h} \leq \frac{C_T T_{\min}}{h}$. Thus it suffices for

$$
\begin{aligned}
\varepsilon &\leq \varepsilon_1\varepsilon_{\mathrm{TV}}\left(\frac{h}{4C_T T_{\min}} \wedge \frac{h}{32C_{\mathrm{LS}}K_\chi}\right) \\
&= \frac{\varepsilon_\chi}{4\sqrt{5C_{\mathrm{LS}}}} \cdot \varepsilon_{\mathrm{TV}} \cdot \frac{\varepsilon_\chi^2}{2720dL^2C_{\mathrm{LS}}}\left(\frac{1}{16C_T C_{\mathrm{LS}}\ln(2K_\chi/\varepsilon_\chi^2)} \wedge \frac{1}{32C_{\mathrm{LS}}K_\chi}\right) \\
&= \frac{\varepsilon_{\mathrm{TV}}\varepsilon_\chi^3}{174080\sqrt{5}dL^2C_{\mathrm{LS}}^{5/2}(C_T\ln(2K_\chi/\varepsilon_\chi^2) \vee 2K_\chi)}. \qquad \square
\end{aligned}
$$

## B.3 Proof of Theorem 2.2

We restate the theorem for convenience.

**Theorem 2.2** (Annealed LMC with $L^2$-accurate score estimate). *Let $p : \mathbb{R}^d \to \mathbb{R}$ be a probability density satisfying Assumption 1 for $M_1 = O(d)$, and let $p_{\sigma^2} := p * \varphi_{\sigma^2}$. Suppose furthermore that $\nabla \ln p_{\sigma^2}$ is $L$-Lipschitz for every $\sigma \geq 0$. Given $\sigma_{\min} > 0$, there exists a sequence $\sigma_{\min} = \sigma_1 < \cdots < \sigma_M$ with $M = O\left(\sqrt{d} \log\left(\frac{dC_{\mathrm{LS}}}{\sigma_{\min}^2}\right)\right)$ such that for each $m$, if*

$$\left\|\nabla \ln(p_{\sigma_m^2}) - s(\cdot, \sigma_m^2)\right\|_{L^2(p_{\sigma_m^2})}^2 = \mathbb{E}_{p_{\sigma_m^2}}\left[\left\|\nabla \ln p_{\sigma_m^2}(x) - s(x, \sigma_m^2)\right\|^2\right] \leq \varepsilon^2.$$

$$\text{with } \varepsilon := \widetilde{O}\left(\frac{\varepsilon_{\mathrm{TV}}^{4.5}}{d^{3.25}L^2 C_{\mathrm{LS}}^{2.5}}\right) \tag{3}$$

*then $x^{(1)}$ is a sample from a distribution $q$ such that $\mathrm{TV}(q, p_{\sigma_1^2}) \leq \varepsilon_{\mathrm{TV}}$.*

*Proof.* We choose

$$h_M = \cdots = h_2 = \Theta\left(\frac{1}{dL^2 C_{\mathrm{LS}}}\right) \qquad\qquad h_1 = \Theta\left(\frac{dL^2 C_{\mathrm{LS}}}{\varepsilon_{\mathrm{TV}}^2}\right)$$

$$T_{M-1} = \cdots = T_2 = \Theta\left(C_{\mathrm{LS}} \ln\left(\frac{M}{\varepsilon_{\mathrm{TV}}}\right)\right) \qquad T_1 = \Theta\left(C_{\mathrm{LS}} \ln\left(\frac{1}{\varepsilon_{\mathrm{TV}}}\right)\right),$$

and $T_M = 0$, $N_m = T_m/h$.

Choose the sequence $\sigma_{\min}^2 = \sigma_1^2 < \cdots < \sigma_M^2$ to be geometric with ratio $1 + \Theta\left(\frac{1}{\sqrt{d}}\right)$. Note that

$$\chi^2(N(0, \sigma_2^2 I_d)\|N(0, \sigma_1^2 I_d)) = \frac{\sigma_1^d}{\sigma_2^{2d}}(2\sigma_2^{-2} - \sigma_1^{-2})^{-d/2} - 1 = \left(\frac{\sigma_2^2}{\sigma_1^2}\right)^{-d/2}\left(2 - \left(\frac{\sigma_2}{\sigma_1}\right)^2\right)^{-\frac{d}{2}}.$$

For $\sigma_2^2 = (1 + \varepsilon)\sigma_1^2$, this equals $(1 + \varepsilon)^{-d/2}(1 - \varepsilon)^{-d/2} = (1 - \varepsilon^2)^{-d/2} - 1$. For $\varepsilon = \Theta\left(\frac{1}{\sqrt{d}}\right)$, this is $d \cdot O\left(\frac{1}{d}\right) = O(1)$. Hence, the $\chi^2$-divergence between successive distributions $p_{\sigma_m^2}$ is $O(1)$. Choosing $\sigma_M^2 = \Omega(d(M_1 + C_{\mathrm{LS}}))$ ensures we have a warm start for the highest noise level by Lemma E.9: $\chi^2(p_{\mathrm{prior}}\|p_{\sigma_M^2}) = O(1)$. This uses $O\left(\sqrt{d} \log\left(\frac{dC_{\mathrm{LS}}}{\sigma_{\min}^2}\right)\right)$ noise levels.

Write $p_m = p_{\sigma_m^2}$ for short. Let $q_m$ be the distribution of the final sample $x^{(m)}$. We show by downwards induction on $m$ that there is $\overline{q}_m$ such that

$$\mathrm{TV}(q_m, \overline{q}_m) \leq \frac{(M + 1) - m}{M + 1}\varepsilon_{\mathrm{TV}}$$

$$\chi^2(\overline{q}_m\|p_m) \leq \left(\frac{\varepsilon_{\mathrm{TV}}}{4(M + 1)}\right)^2.$$

For $m = M$, this follows from the assumption on $\varepsilon$ and Theorem 2.1 with $K_\chi = O(1)$ (given by the warm start).

Fix $m < M$ and suppose it holds for $m+1$. We use the closeness between $q_{m+1}$ and $p_{m+1}$ combined with $\chi^2(p_{m+1}\|p_m) = O(1)$ to obtain compute how close $q_{m+1}$ and $p_m$ are. Because the triangle inequality does not hold for $\chi^2$, we will incur an extra TV error.

Let $\overline{q}_{m,m+1}$ be the distribution of the final sample if $x_0^{(m+1)} \sim \overline{q}_m$. We have $\mathrm{TV}(q_{m+1}, \overline{q}_{m,m+1}) \leq \mathrm{TV}(q_m, \overline{q}_m) \leq \frac{(M+1)-m}{M+1}\varepsilon_{\mathrm{TV}}$.

By Markov's inequality, when $\chi^2(p_{m+1}\|p_m) \leq 1$,

$$\mathbb{P}_{p_{m+1}}\left(\frac{p_{m+1}}{p_m} \geq \frac{8(M + 1)}{\varepsilon_{\mathrm{TV}}}\right) \leq \frac{\chi^2(p_{m+1}\|p_m) + 1}{8(M + 1)/\varepsilon_{\mathrm{TV}}} \leq \frac{\varepsilon_{\mathrm{TV}}}{4(M + 1)}.$$

Let $\overline{q}_{m+1,m} = \mathbb{1}_{\left\{\frac{p_{m+1}}{p_m} \leq \frac{8(M+1)}{\varepsilon_{\mathrm{TV}}}\right\}} \overline{q}_{m+1} \Big/ \int_{\left\{\frac{p_{m+1}}{p_m} \leq \frac{8(M+1)}{\varepsilon_{\mathrm{TV}}}\right\}} \overline{q}_{m+1}.$  Note that (using $\mathrm{TV}(\overline{q}_{m+1}, p_{m+1}) \leq \sqrt{\chi^2(\overline{q}_{m+1}||p_{m+1})} \leq \frac{\varepsilon_{\mathrm{TV}}}{4(M+1)}$)

$$\mathbb{P}_{\overline{q}_{m+1}}\left(\frac{p_{m+1}}{p_m} \geq \frac{8(M+1)}{\varepsilon_{\mathrm{TV}}}\right) \leq \mathbb{P}_{p_{m+1}}\left(\frac{p_{m+1}}{p_m} \geq \frac{8(M+1)}{\varepsilon_{\mathrm{TV}}}\right) + \mathrm{TV}(\overline{q}_{m+1}, p_{m+1})$$

$$\leq \frac{\varepsilon_{\mathrm{TV}}}{4(M+1)} + \frac{\varepsilon_{\mathrm{TV}}}{4(M+1)} \leq \frac{1}{2}. \tag{19}$$

so $\overline{q}_{m+1,m} \leq 2\overline{q}_{m+1}$ and

$$\chi^2(\overline{q}_{m+1,m}||p_m) + 1 \leq 2(\chi^2(\overline{q}_{m+1}||p_m) + 1)$$

$$= \int_{\left\{\frac{p_{m+1}}{p_m} \leq \frac{8(M+1)}{\varepsilon_{\mathrm{TV}}}\right\}} \frac{\overline{q}_{m+1}(x)^2}{p_{m+1}(x)^2} \cdot \frac{p_{m+1}(x)}{p_m(x)} p_{m+1}(x)\, dx$$

$$\leq \frac{8(M+1)}{\varepsilon_{\mathrm{TV}}}(\chi^2(\overline{q}_{m+1}||p_{m+1}) + 1) \leq \frac{16(M+1)}{\varepsilon_{\mathrm{TV}}}.$$

Let $\overline{q}'_{m+1,m}$ be the distribution of $x_{N_m}^{(m)}$ when $x_0^{(m)} \sim \overline{q}_{m+1,m}$. Then by assumption on $\varepsilon$ (3) and Theorem 2.1 (with $K_\chi = 4\sqrt{\frac{M+1}{\varepsilon_{\mathrm{TV}}}}$, $\varepsilon_\chi = \frac{\varepsilon_{\mathrm{TV}}}{4(M+1)}$, and $\varepsilon_{\mathrm{TV}} \leftarrow \frac{\varepsilon_{\mathrm{TV}}}{2(M+1)}$), there is $\overline{q}_m$ such that $\mathrm{TV}(\overline{q}'_{m,m+1}, \overline{q}_m) \leq \frac{\varepsilon_{\mathrm{TV}}}{2(M+1)}$ and $\chi^2(\overline{q}_m||p_m) \leq \frac{\varepsilon_{\mathrm{TV}}}{4(M+1)}$. It remains to bound

$$\mathrm{TV}(q_m, \overline{q}_m) \leq \mathrm{TV}(q_m, \overline{q}'_{m,m+1}) + \mathrm{TV}(\overline{q}'_{m,m+1}, \overline{q}_m)$$

$$\leq \mathrm{TV}(q_{m+1}, \overline{q}_{m,m+1}) + \frac{\varepsilon_{\mathrm{TV}}}{2(M+1)}$$

$$\leq \mathrm{TV}(q_{m+1}, \overline{q}_{m+1}) + \mathrm{TV}(\overline{q}_{m+1}, \overline{q}_{m+1,m}) + \frac{\varepsilon_{\mathrm{TV}}}{2(M+1)}$$

$$\leq \frac{(M+1) - (m+1)}{M+1}\varepsilon_{\mathrm{TV}} + \mathbb{P}_{\overline{q}_{m+1}}\left(\frac{p_{m+1}}{p_m} \geq \frac{8(M+1)}{\varepsilon_{\mathrm{TV}}}\right) + \frac{\varepsilon_{\mathrm{TV}}}{2(M+1)}$$

$$\leq \frac{(M+1) - (m+1)}{M+1}\varepsilon_{\mathrm{TV}} + \frac{\varepsilon_{\mathrm{TV}}}{2(M+1)} + \frac{\varepsilon_{\mathrm{TV}}}{2(M+1)} = \frac{(M+1) - m}{M+1}\varepsilon_{\mathrm{TV}},$$

where we use (19) in the last line. This finishes the induction step.

Finally, the theorem follows by taking $m = 1$ and noting

$$\mathrm{TV}(q_1, p_1) \leq \mathrm{TV}(q_1, \overline{q}_1) + \mathrm{TV}(\overline{q}_1, p_1)$$

$$\leq \mathrm{TV}(q_1, \overline{q}_1) + \sqrt{\chi^2(\overline{q}_1||p_1)} \leq \frac{M\varepsilon_{\mathrm{TV}}}{M+1} + \frac{\varepsilon_{\mathrm{TV}}}{4(M+1)} \leq \varepsilon_{\mathrm{TV}}. \qquad \square$$

## C Analysis for SGM based on reverse SDE's

In this section, we analyze score-based generative models based on reverse SDE's. In Section C.2, we prove convergence of the predictor algorithm under $L^\infty$-accurate score estimate (Theorem 4.3, restated as C.1) using lemmas proved in Section C.3, C.4, C.5, and C.6. In Section C.7, we prove convergence of the predictor algorithm under $L^2$-accurate score estimate (Theorem 3.1, restated as C.16). In Section C.8, we prove convergence of the predictor-corrector algorithm (Theorem 3.2).

### C.1 Discretization and Score Estimation

With a change of variable in (4), we define the sampling process $x_t$ on $[0, T]$ by

$$dx_t = [-f(x_t, T - t) + g(T - t)^2 \nabla \ln \tilde{p}_{T-t}(x_t)]\, dt + g(T - t)\, dw_t, \;\; x_0 \sim \tilde{p}_T.$$

Denoting the distribution of $x_t$ by $p_t$ and running the process from 0 to $T$, we will exactly obtain $p_T = \tilde{p}_0$, which is the data distribution. In practice, we need to discretize this process and replace the score function $\nabla \ln \tilde{p}_{T-t}$ with the estimated score $s$. With a general Euler-Maruyama method, we would obtain $\{z_k\}_{k=0}^N$ defined by

$$z_{(k+1)h} = z_{kh} - h \cdot [f(z_{kh}, T - kh) - g(T - kh)^2 s(z_{kh}, T - kh)] + \sqrt{h} \cdot g(T - kh)\eta_{k+1}, \tag{20}$$

where $h = T/N$ is the step size and $\eta_k$ is a sequence of independent Gaussian random vectors. As we run (20) from 0 to $N$ with $h$ small enough, we should expect that the distribution of $z_T$ is close to that of $x_T$. However, in both SMLD or DDPM models, for fixed $z_k$, the integration

$$\int_{kh}^{(k+1)h} f(z_{kh}, T - t)\, dt \ \text{ and } \ s(z_{kh}, T - kh) \cdot \int_{kh}^{(k+1)h} g(T - t)^2\, dt$$

can be exactly computed, as can the diffusion term. Therefore, we can consider the following process $z_t$ as an "interpolation" of (20):

$$dz_t = [-f(z_{kh}, T - t) + g(T - t)^2 s(z_{kh}, T - kh)]\, dt + g(T - t)\, dw_t, \ \ t \in [kh, (k+1)h]. \tag{21}$$

Note that by running this process instead, we can reduce the discretization error. Now if we denote the distribution of $z_t$ by $q_t$, with $q_0 \approx p_0$, we can expect that $q_T$ is close to $p_T$. Here the estimated score $s$ satisfies for all $x$

$$\|s(x, T - kh) - \nabla \ln \tilde{p}_{T-kh}(x)\| \leq \varepsilon_{kh}, \quad k = 0, 1, \dots, N. \tag{22}$$

Observe that in either SMLD or DDPM, the function $g(t)^2$ is Lipschitz on $[0, T]$. So in the following sections, we will assume that $g(t)^2$ is $L_g$-Lipschitz on $[0, T]$.

## C.2 Predictor

In this section, we present the main result (Theorem C.1) on the one-step error of the predictor in $\chi^2$-divergence, which can be obtained by directly applying the Gronwall's inequality to the differential inequality derived in Lemma C.3. Note that Theorem C.1 is a more precise version of Theorem 4.3; see the remark following the theorem.

**Theorem C.1.** *With the setting in Section C.1, assume $g$ is non-decreasing and let*

$$0 < h \leq \min_{kh \leq t \leq (k+1)h} \frac{1}{g(T - kh)^2(28L^2 + 10C_t + \mathbb{E}_{p_t}\|x\|^2 + 64C_{t,L} + 128C_{d,L} + 360L_s^2(\tilde{R}_t + 2C_t R_d))}$$

*where $C_t$ is the log-Sobolev constant of $p_t$, bounded in Lemma E.7. Suppose that $\nabla \ln p_t$ is $L$-Lipschitz for all $t \in [kh, (k+1)h]$, $s(\cdot, kh)$ is $L_s$-Lipschitz, $L, L_s \geq 1$, and $\varepsilon_{kh}$ is such that (22) holds. Then*

$$\chi^2(q_{(k+1)h}\|p_{(k+1)h}) \leq \left[\chi^2(q_{kh}\|p_{kh}) + \int_{kh}^{(k+1)h} C_{t,kh}\, dt\right] e^{\int_{kh}^{(k+1)h}(-\frac{1}{8C_t} + 8\varepsilon_{kh}^2)g(T-t)^2\, dt}$$

*Here,*

$$C_{t,kh} = \left[8\varepsilon_{kh}^2 + E \cdot (t - kh)g(T - kh)^2\right] g(T - t)^2$$

*and*

$$E = 9(4L_s^2 + 1) + 8C_{d,L}$$

$$C_{t,L} = \begin{cases} 32L^2 & \text{in SMLD,} \\ (88C_t^2 + 400)L^2 & \text{in DDPM,} \end{cases}$$

$$C_{d,L} = \begin{cases} 76L^2 d & \text{in SMLD,} \\ 6 + 94L^2 d & \text{in DDPM} \end{cases} \leq 100L^2 d$$

$$\tilde{R}_t = 9(C_t + 1)$$

$$R_d = 300d + 12$$

*are defined in (24), (27), (28), (30) and (31), respectively.*

*Proof.* The theorem follows from applying Gronwall's inequality to the result of Lemma C.3. $\square$

**Remark.** Note that in DDPM, $E = O(L_s^2 + L^2 d)$. Therefore, when $g \equiv 1$, $C_{t,kh} = O(\varepsilon_1^2 + (L_s^2 + L^2 d)h)$, where we denote the upper bound of $\varepsilon_{kh}$ for all $k \in \{0, ..., N\}$ by $\varepsilon_1$. Using the bound on

the log-Sobolev constant (Lemma E.7) and second moment (Lemma E.8) for DDPM, we note that the restriction on $h$ for all steps is implied by

$$h = O \left( \frac{1}{\mathbb{E}_{p_{\text{data}}} \|x\|^2 + C_{\text{LS}}(C_{\text{LS}} + d)(L \vee L_s)^2} \right)$$

with appropriate constants. Then we can conclude the first inequality in Theorem 4.3 by combining Theorem C.1 and Lemma E.7 and the second inequality from unfolding the first one and evaluating the geometric series. Likewise, we have the following analogue for SMLD, for which we omit the proof.

**Theorem C.2** (Predictor steps under $L^\infty$ bound on score estimate, SMLD). *Let $p : \mathbb{R}^d \to \mathbb{R}$ be a probability density satisfying Assumption 1 and $s(\cdot, t) : \mathbb{R}^d \to \mathbb{R}^d$ be a score estimate $s$ with error bounded in $L^\infty$ for each $t \in [0, T]$:*

$$\|\nabla \ln p - s(\cdot, t)\|_\infty = \max_{x \in \mathbb{R}^d} \|\nabla \ln \widetilde{p}_t(x) - s(x, t)\|] \le \varepsilon_1.$$

*Consider SMLD. Let $C_T = C_{\text{LS}} + T$. Let $g \equiv 1$, $T \ge C_{\text{LS}}d$, and $h = O \left( \frac{1}{\mathbb{E}_{p_0} \|x\|^2 + C_T d(L \vee L_s)^2} \right)$. Then*

$$\chi^2(q_{(k+1)h}||p_{(k+1)h}) \le \chi^2(q_{kh}||p_{kh}) e^{(-\frac{1}{8C_{T-kh}} + 8\varepsilon_1^2)h} + O(\varepsilon_1^2 h + (L_s^2 + L^2 d)h^2)$$

*and letting $t = T - Nh$, if $\varepsilon_1 < \frac{1}{128 C_T}$,*

$$\chi^2(q_{Nh}||p_{Nh}) \le \left( \frac{C_{\text{LS}} + t}{C_{\text{LS}} + T} \right)^{\frac{1}{16}} \chi^2(q_0||p_0) + O \left( \ln \left( \frac{C_{\text{LS}} + T}{C_{\text{LS}} + t} \right) (\varepsilon_1^2 + (L_s^2 + L^2 d)h) \right).$$

*Moreover, for $q_0 = p_{\text{prior}}$, $q_0 = \varphi_T$, $\chi^2(q_0||p_0) \le \frac{C_{\text{LS}}d}{T}$.*

**Remark.** We note that in a sense SMLD and DDPM are equivalent, as we can get from one to the other by rescaling in time and space. First we recall that, as discussed in Section 3, all the SMLD models are equivalent under rescaling in time. Therefore we can assume $g(t) = e^{t/2}$ and consider the forward SDE for SMLD

$$dx_t = e^{t/2} dw_t,$$

where $w_t$ is a standard Brownian Motion. Now let $y_t = e^{-t/2} x_t$; then

$$dy_t = -\frac{1}{2} y_t dt + dw_t,$$

which is exactly DDPM with $g(t) = 1$. Note that Theorem C.2 uses a different parameterization for SMLD and the resulting complexity is slightly worse.

### C.3 Differential Inequality

Now we prove a differential inequality involving $\chi^2(q_t||p_t)$. As in [Che+21], the key difficulty is to bound the discretization error. We decompose it into two error terms and bound them in Lemma C.4 and Lemma C.5 separately.

In the following, we will let

$$G_{kh,t} := \int_{kh}^t g(T - s)^2 \, ds. \tag{23}$$

**Lemma C.3.** *Let $(q_t)_{0 \le t \le T}$ denote the law of the interpolation (21). With the setting in Lemma C.1, we have for $t \in [kh, (k+1)h]$,*

$$\frac{d}{dt} \chi^2(q_t||p_t) \le g(T - t)^2 \left[ \left( -\frac{1}{8C_t} + 8\varepsilon_{kh}^2 \right) \chi^2(q_t||p_t) + \left[ 8\varepsilon_{kh}^2 + E \cdot (t - kh)g(T - kh)^2 \right] \right],$$

*where $C_t$ is the LSI constant of $p_t$, $\varepsilon_{kh}$ is the $L^\infty$-score estimation error at time $kh$ and $E$ is defined in (24).*

*Proof.* By Lemma A.2 with

$$\widehat{f}(z_{kh}, t) \leftarrow -f(z_{kh}, T-t) + g(T-t)^2 s(z_{kh}, T-kh)$$
$$f(z, t) \leftarrow -f(z, T-t) + g(T-t)^2 \nabla \ln \widetilde{p}_{T-t}(z),$$

we have

$$
\frac{d}{dt}\chi^2(q_t\|p_t) = -g(T-t)^2 \mathscr{E}_{p_t}\left(\frac{q_t}{p_t}\right) + 2\mathbb{E}\left[\left\langle \left(-f(z_{kh}, T-t) + g(T-t)^2 s(z_{kh}, T-kh)\right)\right.\right.
$$
$$
\left.\left. - \left(-f(z, T-t) + g(T-t)^2 \nabla \ln \widetilde{p}_{T-t}(z)\right), \nabla\frac{q_t}{p_t}\right\rangle\right]
$$
$$
= -g(T-t)^2 \mathscr{E}_{p_t}\left(\frac{q_t}{p_t}\right) + 2\mathbb{E}\left[\left\langle f(z_t, T-t) - f(z_{kh}, T-t), \nabla\frac{q_t(z_t)}{p_t(z_t)}\right\rangle\right]
$$
$$
+ 2g(T-t)^2 \mathbb{E}\left[\left\langle s(z_{kh}, T-kh) - \nabla \ln \tilde{p}_{T-t}(z_t), \nabla\frac{q_t(z_t)}{p_t(z_t)}\right\rangle\right]
$$
$$
=: -g(T-t)^2 \mathscr{E}_{p_t}\left(\frac{q_t}{p_t}\right) + A + B.
$$

By Lemma C.4,

$$
A \le g(T-t)^2 \left[2(\chi^2(q_t\|p_t) + 1)\mathbb{E}\left[\|z_t - z_{kh}\|^2 \psi_t(z_t)\right] + \frac{1}{8}\mathscr{E}_{p_t}\left(\frac{q_t}{p_t}\right)\right],
$$

while by Lemma C.5,

$$
B \le \frac{1}{2}g(T-t)^2 \mathscr{E}_{p_t}\left(\frac{q_t}{p_t}\right) + 8g(T-t)^2 L_s^2(\chi^2(q_t\|p_t) + 1)\mathbb{E}\left[\|z_t - z_{kh}\|^2 \psi_t(z_t)\right]
$$
$$
+ 8\left[\varepsilon_{kh}^2 + G_{kh,t}C_{d,L}\right]g(T-t)^2(\chi^2(q_t\|p_t) + 1).
$$

Therefore, for $h \le \frac{1}{72g(T-kh)^2(4L_s^2+1)(\tilde{R}_t \vee 2C_t R_d)} \wedge \frac{1}{128g(T-kh)^2 C_{d,L}}$, using Lemma C.15,

$$
\frac{d}{dt}\chi^2(q_t\|p_t) \le -\frac{3}{8}g(T-t)^2 \mathscr{E}_{p_t}\left(\frac{q_t}{p_t}\right) + g(T-t)^2(8L_s^2 + 2)(\chi^2(q_t\|p_t) + 1)\mathbb{E}\left[\|z_t - z_{kh}\|^2 \psi_t(z_t)\right]
$$
$$
+ 8\left[\varepsilon_{kh}^2 + G_{kh,t}C_{d,L}\right]g(T-t)^2(\chi^2(q_t\|p_t) + 1)
$$
$$
\le -\frac{3}{8}g(T-t)^2 \mathscr{E}_{p_t}\left(\frac{q_t}{p_t}\right)
$$
$$
+ 9g(T-t)^2(4L_s^2 + 1)G_{kh,t}\left[\tilde{R}_t \mathscr{E}_{p_t}\left(\frac{q_t}{p_t}\right) + R_{t,kh}(\chi^2(q_t\|p_t) + 1)\right]
$$
$$
+ 8\left[\varepsilon_{kh}^2 + G_{kh,t}C_{d,L}\right]g(T-t)^2(\chi^2(q_t\|p_t) + 1)
$$
$$
\le -\frac{2}{8}g(T-t)^2 \mathscr{E}_{p_t}\left(\frac{q_t}{p_t}\right) + g(T-t)^2\frac{1}{8C_t}\chi^2(q_t\|p_t) + 8g(T-t)^2\varepsilon_{kh}^2\chi^2(q_t\|p_t)
$$
$$
+ g(T-t)^2\left[8\varepsilon_{kh}^2 + 8C_{d,L}G_{kh,t} + 9(4L_s^2 + 1)G_{kh,t}R_d\right].
$$

Using the fact that $p_t$ satisfies a log-Sobolev inequality with constant $C_t$,

$$
\frac{d}{dt}\chi^2(q_t\|p_t) \le -\frac{2}{8C_t}g(T-t)^2\chi^2(q_t\|p_t) + \frac{1}{8C_t}g(T-t)^2\chi^2(q_t\|p_t) + 8g(T-t)^2\varepsilon_{kh}^2\chi^2(q_t\|p_t)
$$
$$
+ g(T-t)^2\left[8\varepsilon_{kh}^2 + 8C_{d,L}G_{kh,t} + 9(4L_s^2 + 1)G_{kh,t}R_d\right]
$$
$$
\le \left(-\frac{1}{8C_t} + 8\varepsilon_{kh}^2\right)g(T-t)^2\chi^2(q_t\|p_t) + g(T-t)^2[8\varepsilon_{kh}^2 + E(t-kh)g(T-kh)^2].
$$

where

$$
E = 9(4L_s^2 + 1) + 8C_{d,L}. \tag{24}
$$

$\square$

In order to bound the error terms $A$ and $B$, we will use Lemma A.3. Let $\phi_t(x) = \frac{q_t(x)}{p_t(x)}$ and $\psi_t(x) = \phi_t(x)/\mathbb{E}_{p_t}\phi_t^2$. Then $\mathbb{E}\psi_t(z_t) = 1$ and in fact the normalizing factor $\mathbb{E}_{p_t}\phi_t^2 = \chi^2(q_t\|p_t)+1$. We first deal with error term $A$.

**Lemma C.4.** *In the setting of Lemma C.3, we have the following bound for term $A$:*

$$2\mathbb{E}\left[\left\langle f(z_t, T-t) - f(z_{kh}, T-t), \nabla\frac{q_t(z_t)}{p_t(z_t)}\right\rangle\right]$$

$$\leq g(T-t)^2\left[2(\chi^2(q_t\|p_t)+1)\mathbb{E}\left[\|z_t - z_{kh}\|^2\,\psi_t(z_t)\right] + \frac{1}{8}\mathscr{E}_{p_t}\left(\frac{q_t}{p_t}\right)\right].$$

*Proof.* In SMLD, $f(x,t) = 0$ and hence $A = 0$; while in DDPM, $f(x,t) = -\frac{1}{2}g(t)^2 x$. Therefore, by Lemma A.3,

$$2\mathbb{E}\left[\left\langle f(z_t, T-t) - f(z_{kh}, T-t), \nabla\frac{q_t(z_t)}{p_t(z_t)}\right\rangle\right]$$

$$= -g(T-t)^2\mathbb{E}\left[\left\langle z_t - z_{kh}, \nabla\frac{q_t(z_t)}{p_t(z_t)}\right\rangle\right]$$

$$\leq g(T-t)^2\left[2\cdot\mathbb{E}_{p_t}\phi_t^2\cdot\mathbb{E}\left[\|z_t - z_{kh}\|^2\,\psi_t(z_t)\right] + \frac{1}{8}\mathscr{E}_{p_t}\left(\frac{q_t}{p_t}\right)\right]$$

$$= g(T-t)^2\left[2(\chi^2(q_t\|p_t)+1)\mathbb{E}\left[\|z_t - z_{kh}\|^2\,\psi_t(z_t)\right] + \frac{1}{8}\mathscr{E}_{p_t}\left(\frac{q_t}{p_t}\right)\right]. \qquad \square$$

Now we bound error term $B$.

**Lemma C.5.** *In the setting of Lemma C.3, we have the following bound for term $B$:*

$$B \leq \frac{1}{2}g(T-t)^2\mathscr{E}_{p_t}\left(\frac{q_t}{p_t}\right) + 8g(T-t)^2 L_s^2(\chi^2(q_t\|p_t)+1)\mathbb{E}\left[\|z_t - z_{kh}\|^2\,\psi_t(z_t)\right]$$

$$+ 8\left[\varepsilon_{kh}^2 + G_{kh,t}C_{d,L}\right]g(T-t)^2(\chi^2(q_t\|p_t)+1).$$

*Proof.* We first decompose the error:

$$\mathbb{E}\left[\left\langle s(z_{kh}, T-kh) - \nabla\ln\tilde{p}_{T-t}(z_t), \nabla\frac{q_t(z_t)}{p_t(z_t)}\right\rangle\right] = \mathbb{E}\left[\left\langle s(z_{kh}, T-kh) - s(z_t, T-kh), \nabla\frac{q_t(z_t)}{p_t(z_t)}\right\rangle\right]$$

$$+ \mathbb{E}\left[\left\langle s(z_t, T-kh) - \nabla\ln p_{kh}(z_t), \nabla\frac{q_t(z_t)}{p_t(z_t)}\right\rangle\right]$$

$$+ \mathbb{E}\left[\left\langle \nabla\ln p_{kh}(z_t) - \nabla\ln p_t(z_t), \nabla\frac{q_t(z_t)}{p_t(z_t)}\right\rangle\right]$$

$$=: B_1 + B_2 + B_3.$$

Now we bound these error terms separately. For $B_1$, by the Lipschitz assumption, we have by Lemma A.3, for a constant $C_2 > 0$ to be chosen later,

$$B_1 \leq \mathbb{E}\left[L_s\,\|z_{kh} - z_t\|\cdot\left\|\nabla\frac{q_t(z_t)}{p_t(z_t)}\right\|\right]$$

$$\leq 4L_s^2\cdot\mathbb{E}_{p_t}\phi_t^2\cdot\mathbb{E}\left[\|z_t - z_{kh}\|^2\,\psi_t(z_t)\right] + \frac{1}{16}\mathscr{E}_{p_t}\left(\frac{q_t}{p_t}\right)$$

$$= 4L_s^2(\chi^2(q_t\|p_t)+1)\mathbb{E}\left[\|z_t - z_{kh}\|^2\,\psi_t(z_t)\right] + \frac{1}{16}\mathscr{E}_{p_t}\left(\frac{q_t}{p_t}\right).$$

For $B_2$, recalling the assumption that $\|s(x, T-kh) - \nabla\ln p_{kh}(x)\| \leq \varepsilon_{kh}$ for all $x$, we have by Lemma A.3

$$B_2 \leq 4\mathbb{E}\left[\|s(z_t, T-kh) - \nabla\ln p_{kh}(z_t)\|^2\,\psi_t(z_t)\right]\cdot\mathbb{E}_{p_t}[\phi_t^2] + \frac{1}{16}\mathscr{E}_{p_t}\left(\frac{q_t}{p_t}\right)$$

$$\leq 4\varepsilon_{kh}^2(\chi^2(q_t\|p_t)+1) + \frac{1}{16}\mathscr{E}_{p_t}\left(\frac{q_t}{p_t}\right). \qquad (25)$$

Now for the last error term $B_3$, we have by Lemma A.3 that

$$B_3 \leq 4\mathbb{E}_{p_t}\phi_t^2 \cdot \mathbb{E}\left[\|\nabla \ln p_{kh}(z_t) - \nabla \ln p_t(z_t)\|^2 \psi_t(z_t)\right] + \frac{1}{16}\mathscr{E}_{p_t}\left(\frac{q_t}{p_t}\right)$$

$$\leq 4K_{t,kh}(\chi^2(q_t\|p_t) + 1) + \frac{1}{16}\mathscr{E}_{p_t}\left(\frac{q_t}{p_t}\right). \tag{26}$$

Here $K_{t,kh}$ is the bound for $\mathbb{E}\left[\psi_t(z_t)\|\nabla \ln p_{kh}(z_t) - \nabla \ln p_t(z_t)\|^2\right]$ obtained in Lemma C.13:

$$K_{t,kh} := G_{kh,t}\left[\frac{C_{t,L}}{\chi^2(q_t\|p_t) + 1} \cdot \mathscr{E}_{p_t}\left(\frac{q_t}{p_t}\right) + C_{d,L}\right]$$

where $C_{t,L}$ and $C_{d,L}$ are constants defined in (27) and (28) respectively. Hence

$$B_3 \leq 4G_{kh,t}\left[C_{t,L}\mathscr{E}_{p_t}\left(\frac{q_t}{p_t}\right) + C_{d,L}(\chi^2(q_t\|p_t) + 1)\right] + \frac{1}{16}\mathscr{E}_{p_t}\left(\frac{q_t}{p_t}\right).$$

Combining all these results, we finally obtain the bound for error term $B$ in Lemma C.3: for $h \leq \frac{1}{64C_{t,L}g(T-kh)^2}$,

$$B = 2g(T - t)^2(B_1 + B_2 + B_3)$$

$$\leq \frac{3}{8}g(T - t)^2\mathscr{E}_{p_t}\left(\frac{q_t}{p_t}\right) + 8g(T - t)^2 L_s^2(\chi^2(q_t\|p_t) + 1)\mathbb{E}\left[\|z_t - z_{kh}\|^2 \psi_t(z_t)\right]$$

$$+ 8C_{t,L}g(T - t)^2 G_{kh,t}\mathscr{E}_{p_t}\left(\frac{q_t}{p_t}\right)$$

$$+ 8\left[\varepsilon_{kh}^2 + G_{kh,t}C_{d,L}\right]g(T - t)^2(\chi^2(q_t\|p_t) + 1)$$

$$\leq \frac{1}{2}g(T - t)^2\mathscr{E}_{p_t}\left(\frac{q_t}{p_t}\right) + 8g(T - t)^2 L_s^2(\chi^2(q_t\|p_t) + 1)\mathbb{E}\left[\|z_t - z_{kh}\|^2 \psi_t(z_t)\right]$$

$$+ 8\left[\varepsilon_{kh}^2 + G_{kh,t}C_{d,L}\right]g(T - t)^2(\chi^2(q_t\|p_t) + 1). \qquad \square$$

## C.4 Change of Measure

As shown in Lemma C.4 and Lemma C.5, the key to the proof of Lemma C.3 is bounding the discretization error $A$ and $B$. The difficulty is that these errors usually have the form of $\mathbb{E}_{\psi_t q_t}\left[\|u(x)\|^2\right]$ for some function $u : \mathbb{R}^d \to \mathbb{R}^d$, while it is usually easier to bound those expectations over the original probability measure or our target distribution $p_t$. Therefore, as discussed in [Che+21, Section 5.1], our task is to bound these error terms under a complicated change of measure. We first state such a result with respect to the gradient of the potential.

**Lemma C.6.** *[Che+21, Lemma 16] Assume that $p(x) \propto e^{-V(x)}$ is a density in $\mathbb{R}^d$ and $\nabla V(x)$ is L-Lipschitz. Then for any probability density $q$, it holds that*

$$\mathbb{E}_q\left[\|\nabla V\|^2\right] \leq 4\mathbb{E}_p\left[\left\|\nabla\sqrt{\frac{q(x)}{p(x)}}\right\|^2\right] + 2dL = \mathbb{E}_q\left[\left\|\nabla \ln \frac{q(x)}{p(x)}\right\|^2\right] + 2dL.$$

*Proof.* Define the Langevin diffusion w.r.t. $p(x)$:

$$dx_t = -\nabla V(x_t)\,dt + \sqrt{2}\,dw_t,$$

where $B_t$ is a standard Brownian Motion in $\mathbb{R}^d$. Let $\mathcal{L}$ be the corresponding infinitesimal generator, i.e., $\mathcal{L}f = \langle \nabla V, \nabla f \rangle - \Delta f$. Observe that $\mathcal{L}V = \|\nabla V\|^2 - \Delta V$ and $\mathbb{E}_p \mathcal{L}f = 0$ for any $f$, so

$$\mathbb{E}_q \left[ \|\nabla V\|^2 \right] = \mathbb{E}_q \mathcal{L}V + \mathbb{E}_q \Delta V$$

$$\leq \int \mathcal{L}V \left( \frac{q(x)}{p(x)} - 1 \right) p(x)dx + dL = \int \left\langle \nabla V, \nabla \frac{q(x)}{p(x)} \right\rangle p(x)dx + dL$$

$$= 2 \int \left\langle \sqrt{\frac{q(x)}{p(x)}} \nabla V, \nabla \sqrt{\frac{q(x)}{p(x)}} \right\rangle p(x)dx + dL$$

$$\leq \frac{1}{2} \mathbb{E}_q \left[ \|\nabla V\|^2 \right] + 2\mathbb{E}_p \left[ \left\| \nabla \sqrt{\frac{q(x)}{p(x)}} \right\|^2 \right] + dL.$$

Rearrange this inequality to obtain the desired result. $\qquad\square$

Now applying this Lemma to $p = p_t$ and $q = \psi_t q_t$, we get immediately the following corollary. Note that $\psi_t q_t$ is a density function because $\int \psi_t(x) q_t(x)\, dx = \int \frac{q_t(x)}{p_t(x)} q_t(x)\, dx / \mathbb{E}_{p_t} \phi_t^2 = 1$ and $\psi_t(x) q_t(x) \geq 0$ for any $x \in \mathbb{R}^d$.

**Corollary C.7.** *In the setting of Lemma C.3, it holds that*

$$\mathbb{E} \left[ \psi_t(z_t) \|\nabla \ln p_t(z_t)\|^2 \right] \leq \frac{4}{\chi^2(q_t\|p_t) + 1} \cdot \mathscr{E}_{p_t} \left( \frac{q_t}{p_t} \right) + 2dL.$$

*Proof.* Applying Lemma C.6 to the density $\psi_t q_t$ yields

$$\mathbb{E}_{\psi_t q_t} \left[ \|\nabla \ln p_t(x)\|^2 \right] \leq \mathbb{E}_{\psi_t q_t} \left[ \left\| \nabla \ln \frac{\psi_t(x) q_t(x)}{p_t(x)} \right\|^2 \right] + 2dL = \frac{4}{\chi^2(q_t\|p_t) + 1} \cdot \mathscr{E}_{p_t} \left( \frac{q_t}{p_t} \right) + 2dL.$$

$\qquad\square$

Note that we cannot expect analogous results for a general $u(x)$ as in Lemma C.6. In the general case, we apply the Donsker-Varadhan variational principle, which states that for probability measures $p$ and $q$,

$$\mathbb{E}_q \|u(x)\|^2 \leq \mathrm{KL}(q\|p) + \ln \mathbb{E}_p \exp \|u(x)\|^2.$$

Towards this end, we first need to analyze $\mathrm{KL}(\psi_t q_t \| p_t)$.

**Lemma C.8.** *Let $\phi_t(x) = \frac{q_t(x)}{p_t(x)}$ and $\psi_t(x) = \phi_t(x)/\mathbb{E}_{p_t}\phi_t^2$. If $p_t$ satisfies a LSI with constant $C_t$, then*

$$\mathrm{KL}(\psi_t q_t \| p_t) \leq \frac{2C_t}{\chi^2(q_t\|p_t) + 1} \cdot \mathscr{E}_{p_t} \left( \frac{q_t}{p_t} \right).$$

*Proof.* Since $p_t$ satisfies LSI with constant $C_t$,

$$\mathrm{KL}(\psi_t q_t \| p_t) \leq \frac{C_t}{2} \int \left\| \nabla \ln \frac{\psi_t(x) q_t(x)}{p_t(x)} \right\|^2 \psi_t(x) q_t(x) dx$$

$$= 2C_t \int \left\| \nabla \ln \frac{q_t(x)}{p_t(x)} \right\|^2 \psi_t(x) q_t(x) dx$$

$$= 2C_t \int \left\| \nabla \frac{q_t(x)}{p_t(x)} \right\|^2 \frac{\psi_t(x) p_t(x)^2}{q_t(x)} dx$$

$$= \frac{2C_t}{\chi^2(q_t\|p_t) + 1} \cdot \int \left\| \nabla \frac{q_t(x)}{p_t(x)} \right\|^2 p_t(x) dx$$

$$= \frac{2C_t}{\chi^2(q_t\|p_t) + 1} \cdot \mathscr{E}_{p_t} \left( \frac{q_t}{p_t} \right). \qquad\square$$

With this in hand, we are ready to bound the second moment of $\psi_t q_t$ as well as the variance of a Gaussian random vector with respect to this measure:

**Lemma C.9.** *With the setting of Lemma C.3, we have*

$$\mathbb{E}\left[\psi_t(z_t)\,\|z_t\|^2\right] \leq \frac{2C_t^2}{\chi^2(q_t\|p_t)+1}\cdot\mathscr{E}_{p_t}\left(\frac{q_t}{p_t}\right) + \frac{1}{2}\mathbb{E}_{p_t}\left[\|x\|^2\right] + \frac{1}{2}C_t,$$

*where $C_t$ is the LSI constant of $p_t$, which is bounded in Lemma E.6, and the second moment of $p_t$ is bounded in Lemma E.8.*

*Proof.* Since $p_t$ has LSI constant $C_t$, by Donsker-Varadhan variational principle,

$$\mathbb{E}\left[\psi_t(z_t)\,\|z_t\|^2\right] = \frac{2}{s}\mathbb{E}_{\psi_t q_t}\left[\frac{s}{2}\|x\|^2\right] \leq \frac{2}{s}\left[\mathrm{KL}(\psi_t q_t\|p_t) + \ln\mathbb{E}_{p_t}\left[e^{\frac{s}{2}\|x\|^2}\right]\right]$$

for any $s > 0$. By Lemma E.1, for any $s \in [0, \frac{1}{C_t})$, we have

$$\mathbb{E}_{p_t}\left[e^{\frac{s}{2}\|x\|^2}\right] \leq \frac{1}{\sqrt{1-C_t\cdot s}}\exp\left[\frac{s}{2(1-C_t\cdot s)}(\mathbb{E}_{p_t}\|x\|)^2\right].$$

Now choose $s = \frac{1}{2C_t}$, we have

$$\mathbb{E}_{p_t}\left[e^{\frac{s}{2}\|x\|^2}\right] \leq \sqrt{2}\exp\left[\frac{1}{2C_t}(\mathbb{E}_{p_t}\|x\|)^2\right].$$

Hence

$$\mathbb{E}\left[\psi_t(z_t)\,\|z_t\|^2\right] \leq C_t\cdot\left[\mathrm{KL}(\psi_t q_t\|p_t) + \frac{1}{2C_t}\mathbb{E}_{p_t}\left[\|x\|^2\right] + \frac{\ln 2}{2}\right].$$

Now with the bound of $\mathrm{KL}(\psi_t q_t\|p_t)$ in Lemma C.8, we obtain

$$\mathbb{E}\left[\psi_t(z_t)\,\|z_t\|^2\right] \leq \frac{2C_t^2}{\chi^2(q_t\|p_t)+1}\cdot\mathscr{E}_{p_t}\left(\frac{q_t}{p_t}\right) + \frac{1}{2}\mathbb{E}_{p_t}\left[\|x\|^2\right] + \frac{1}{2}C_t. \qquad \square$$

**Lemma C.10.** *With the setting of Lemma C.3,*

$$\mathbb{E}\left[\psi_t(z_t)\left\|\int_{kh}^t g(T-s)dw_s\right\|^2\right] \leq 2\int_{kh}^t g(T-s)^2\,ds\cdot\left[\frac{8C_t}{\chi^2(q_t\|p_t)+1}\cdot\mathscr{E}_{p_t}\left(\frac{q_t}{p_t}\right) + d + 8\ln 2\right],$$

*where $C_t$ is the LSI constant of $p_t$.*

*Proof.* Note that $\int_{kh}^t g(T-s)dw_s$ is a Gaussian random vector with variance $\int_{kh}^t g(T-s)^2 ds\cdot I_d$. Using the Donsker-Varadhan variational principle, for any random variable $X$,

$$\tilde{\mathbb{E}}X \leq \mathrm{KL}(\tilde{\mathbb{P}}\|\mathbb{P}) + \ln\mathbb{E}\exp X.$$

Applying this to $X = c\left(\left\|\int_{kh}^t g(T-s)dw_s\right\| - \mathbb{E}\left\|\int_{kh}^t g(T-s)dw_s\right\|\right)^2$ for a constant $c > 0$ to be chosen later, we can bound

$$\tilde{\mathbb{E}}\left\|\int_{kh}^t g(T-s)dw_s\right\|^2 \leq 2\mathbb{E}\left[\left\|\int_{kh}^t g(T-s)dw_s\right\|^2\right]$$
$$+ \frac{2}{c}\left[\mathrm{KL}(\tilde{\mathbb{P}}\|\mathbb{P}) + \ln\mathbb{E}\exp\left(c\left(\left\|\int_{kh}^t g(T-s)dw_s\right\| - \mathbb{E}\left\|\int_{kh}^t g(T-s)dw_s\right\|\right)^2\right)\right],$$

where $\frac{d\tilde{\mathbb{P}}}{d\mathbb{P}} = \psi_t(z_t)$. Now following [Che+21, Theorem 4], we set $c = \frac{1}{8\int_{kh}^t g(s)^2 ds}$, so that

$$\mathbb{E}\exp\left[\frac{\left(\left\|\int_{kh}^t g(T-s)dw_s\right\| - \mathbb{E}\left\|\int_{kh}^t g(T-s)dw_s\right\|\right)^2}{8\int_{kh}^t g(s)^2 ds}\right] \leq 2.$$

Next, using the LSI for $p_t$, we have

$$\mathrm{KL}(\tilde{\mathbb{P}}||\mathbb{P}) = \mathbb{E}_{\psi_t q_t} \ln \psi_t = \mathbb{E}_{\psi_t q_t} \ln \frac{\phi_t}{\mathbb{E}_{p_t} \phi_t^2} = \frac{1}{2} \mathbb{E}_{\psi_t q_t} \ln \frac{\phi_t^2}{(\mathbb{E}_{p_t} \phi_t^2)^2}$$

$$= \frac{1}{2} \left[ \mathbb{E}_{\psi_t q_t} \ln \frac{\phi_t^2}{\mathbb{E}_{p_t} \phi_t^2} - \ln \mathbb{E}_{p_t} \phi_t^2 \right] = \frac{1}{2} \left[ \mathbb{E}_{\psi_t q_t} \ln \frac{\psi_t q_t}{p_t} - \ln \mathbb{E}_{p_t} \phi_t^2 \right].$$

Noting that $\mathbb{E}_{p_t} \phi_t^2 = \chi^2(q_t||p_t) + 1 \geq 1$, we have that

$$\mathrm{KL}(\tilde{\mathbb{P}}||\mathbb{P}) \leq \frac{1}{2} \mathrm{KL}(\psi_t q_t || p_t) \leq \frac{C_t}{\chi^2(q_t||p_t) + 1} \cdot \mathscr{E}_{p_t}\left(\frac{q_t}{p_t}\right),$$

where the last inequality is due to Lemma C.8. We have proved

$$\mathbb{E}\left[ \psi_t(z_t) \left\| \int_{kh}^t g(T-s)\, dw_s \right\|^2 \right]$$

$$\leq 2d \int_{kh}^t g(T-s)^2\, ds + 16 \int_{kh}^t g(T-s)^2 ds \cdot \left[ \frac{C_t}{\chi^2(q_t||p_t) + 1} \cdot \mathscr{E}_{p_t}\left(\frac{q_t}{p_t}\right) + \ln 2 \right]$$

$$\leq 2 \int_{kh}^t g(T-s)^2\, ds \cdot \left[ \frac{8C_t}{\chi^2(q_t||p_t) + 1} \cdot \mathscr{E}_{p_t}\left(\frac{q_t}{p_t}\right) + d + 8 \ln 2 \right]. \qquad \square$$

### C.5 Perturbation Error

In the previous section, we bound errors in the form of $\mathbb{E}_{\psi_t q_t} \|u(x)\|^2$ with a change of measure technique, where $\|u(x)\|^2$ is easy to bound with respect to the original measure or $p_t$. However, this is not always the case for the errors we are considering. In this section, we aim to bound $\mathbb{E}_{\psi_t q_t}\left[ \|\nabla \ln p_{kh}(x) - \nabla \ln p_t(x)\|^2 \right]$, where, as discussed in Lemma C.13, $p_{kh}$ can be regarded as a perturbed version of $p_t$ with some Gaussian noise. We first provide a point-wise bound for SMLD (Lemma C.11) and DDMP (Lemma C.12), respectively and then use them to bound the expectation with respect to $\psi_t q_t$.

**Lemma C.11.** *Suppose that $p(x) \propto e^{-V(x)}$ is a probability density on $\mathbb{R}^d$, where $V(x)$ is $L$-smooth, and let $\varphi_{\sigma^2}(x)$ be the density function of $N(0, \sigma^2 I_d)$. Then for $L \leq \frac{1}{2\sigma^2}$,*

$$\left\| \nabla \ln \frac{p(x)}{(p * \varphi_{\sigma^2})(x)} \right\| \leq 6L\sigma d^{1/2} + 2L\sigma^2 \|\nabla V(x)\|.$$

*Proof.* Note that

$$\nabla \ln p * \varphi_{\sigma^2}(x) = \frac{\int_{\mathbb{R}^d} -\nabla V(y) e^{-V(y)} e^{-\frac{\|x-y\|^2}{2\sigma^2}}\, dy}{\int_{\mathbb{R}^d} e^{-V(y)} e^{-\frac{\|x-y\|^2}{2\sigma^2}}\, dy} = -\mathbb{E}_{p_{x,\sigma^2}} \nabla V(y),$$

where $p_{x,\sigma^2}$ denotes the probability density

$$p_{x,\sigma^2}(y) \propto p(y) e^{-\frac{\|y-x\|^2}{2\sigma^2}}$$

so when $V$ is $L$-smooth,

$$\left\| \nabla \ln \frac{p(x)}{p * \varphi_{\sigma^2}(x)} \right\| = \left\| \mathbb{E}_{p_{x,\sigma^2}} [\nabla V(y) - \nabla V(x)] \right\|$$

$$\leq \mathbb{E}_{p_{x,\sigma^2}} [L \|y - x\|]$$

We now write

$$\mathbb{E}_{p_{x,\sigma^2}} \|y - x\| \leq \mathbb{E}_{p_{x,\sigma^2}} \left\| y - \mathbb{E}_{p_{x,\sigma^2}} y \right\| + \left\| \mathbb{E}_{p_{x,\sigma^2}} y - y^* \right\| + \|y^* - x\|,$$

where $y^* \in \mathrm{argmax}_y\, p_{x,\sigma^2}(y)$ is a mode of the distribution $p_{x,\sigma^2}$. We now bound each of these terms.

1. For the first term, note that $p_{x,\sigma^2}$ is $\left(\frac{1}{\sigma^2} - L\right)$-strongly convex, so satisfies a Poincaré inequality with constant $\left(\frac{1}{\sigma^2} - L\right)^{-1}$. Thus

$$\mathbb{E}_{p_{x,\sigma^2}} \|y - x\| \leq \mathbb{E}_{p_{x,\sigma^2}} [\|y - \mathbb{E}_{p_{x,\sigma^2}} y\|^2]^{1/2}$$

$$= \left(\sum_{i=1}^{d} \operatorname{Var}_{p_{x,\sigma^2}}(y_i)\right)^{1/2} \leq \left(d \left(\frac{1}{\sigma^2} - L\right)^{-1}\right)^{1/2}.$$

2. For the second term, by Lemma E.3, noting that $V(y) + \frac{\|x-y\|^2}{2\sigma^2}$ is $\left(\frac{1}{\sigma^2} + L\right)$-smooth,

$$\left\|\mathbb{E}_{p_{x,\sigma^2}} y - y^*\right\| \leq \left(\frac{1}{\sigma^2} - L\right)^{-1/2} d^{1/2} \left(5 + \ln\left(\left(\frac{1}{\sigma^2} - L\right)^{-1}\left(\frac{1}{\sigma^2} + L\right)\right)\right)^{1/2}$$

$$\leq \left(\frac{1}{\sigma^2} - L\right)^{-1/2} d^{1/2} \left(5 + \ln\frac{1 + L\sigma^2}{1 - L\sigma^2}\right)^{1/2}$$

$$\leq \sqrt{7} \left(\frac{1}{\sigma^2} - L\right)^{-1/2} d^{1/2},$$

where the last inequality uses $\sigma^2 \leq \frac{1}{2L}$.

3. For the third term, we note that the mode satisfies

$$\nabla V(y^*) + \frac{y^* - x}{\sigma^2} = 0$$

$$-\frac{y^* - x}{\sigma^2} = \nabla V(y^*) = (\nabla V(y^*) - \nabla V(x)) + \nabla V(x)$$

$$\frac{1}{\sigma^2} \|y^* - x\| \leq \|\nabla V(x)\| + L \|y^* - x\|$$

$$\|y^* - x\| \leq \left(\frac{1}{\sigma^2} - L\right)^{-1} \|\nabla V(x)\|.$$

Putting these together and using $\left(\frac{1}{\sigma^2} - L\right)^{-1} \leq 2$, we obtain

$$\left\|\nabla \ln \frac{p(x)}{p * \varphi_{\sigma^2}(x)}\right\| \leq (\sqrt{7} + 1)L \left(\frac{1}{\sigma^2} - L\right)^{-1/2} d^{1/2} + L \left(\frac{1}{\sigma^2} - L\right)^{-1} \|\nabla V(x)\|$$

$$\leq 6L\sigma d^{1/2} + 2L\sigma^2 \|\nabla V(x)\|. \qquad \square$$

**Lemma C.12.** *With the setting in Lemma C.11 and the notation* $p_\alpha(x) = \alpha^d p(\alpha x)$ *for* $\alpha \geq 1$, *we have that for* $L \leq \frac{1}{2\alpha^2\sigma^2}$,

$$\left\|\nabla \ln \frac{p(x)}{(p_\alpha * \varphi_{\sigma^2})(x)}\right\| \leq 6\alpha^2 L\sigma d^{1/2} + (\alpha + 2\alpha^3 L\sigma^2)(\alpha - 1)L \|x\| + (\alpha - 1 + 2\alpha^3 L\sigma^2) \|\nabla V(x)\|.$$

*Proof.* Note $p_\alpha(x)$ is also a probability density in $\mathbb{R}^d$. By the triangle inequality,

$$\left\|\nabla \ln \frac{p(x)}{(p_\alpha * \varphi_{\sigma^2})(x)}\right\| \leq \left\|\nabla \ln \frac{p(x)}{p_\alpha(x)}\right\| + \left\|\nabla \ln \frac{p_\alpha(x)}{(p_\alpha * \varphi_{\sigma^2})(x)}\right\|.$$

Without loss of generality, we can assume that $p(x) = e^{-V(x)}$; then $p_\alpha(x) = \alpha^d e^{-V(\alpha x)}$. Hence

$$\left\|\nabla \ln \frac{p(x)}{p_\alpha(x)}\right\| = \|\alpha \nabla V(\alpha x) - \nabla V(x)\|$$

$$\leq \|\alpha \nabla V(\alpha x) - \alpha \nabla V(x)\| + \|\alpha \nabla V(x) - \nabla V(x)\|$$

$$\leq \alpha(\alpha - 1)L \|x\| + (\alpha - 1) \|\nabla V(x)\|.$$

Since $\alpha \nabla V(\alpha x)$ is $\alpha^2 L$-Lipschitz, by Lemma C.11,

$$\left\| \nabla \ln \frac{p_\alpha(x)}{(p_\alpha * \varphi_{\sigma^2})(x)} \right\| \leq 6\alpha^2 L \sigma d^{1/2} + 2\alpha^3 L \sigma^2 \|\nabla V(\alpha x)\|.$$

By the Lipschitz assumption,

$$\|\nabla V(\alpha x)\| \leq \|\nabla V(\alpha x) - \nabla V(x)\| + \|\nabla V(x)\| \leq (\alpha - 1)L\|x\| + \|\nabla V(x)\|.$$

The result follows from combining the three inequalities above. $\qquad\square$

**Lemma C.13.** *In the setting of Lemma C.3, we have for $t \in [kh, (k+1)h]$,*

$$\mathbb{E}\left[\psi_t(z_t)\|\nabla \ln p_{kh}(z_t) - \nabla \ln p_t(z_t)\|^2\right]$$

$$\leq G_{kh,t} \cdot \left[\frac{C_{t,L}}{\chi^2(q_t\|p_t)+1}G_{kh,t}\mathscr{E}_{p_t}\left(\frac{q_t}{p_t}\right) + C_{d,L}\right],$$

*where*

$$C_{t,L} = \begin{cases} 32L^2 & \text{in SMLD,} \\ (88C_t^2 + 400)L^2 & \text{in DDPM,} \end{cases} \tag{27}$$

*and*

$$C_{d,L} = \begin{cases} 76L^2 d & \text{in SMLD,} \\ 6 + 94L^2 d & \text{in DDPM} \end{cases} \leq 100L^2 d. \tag{28}$$

*Proof.* In both SMLD and DDPM models, we have the following relationship for $t \in [kh, (k+1)h]$:

$$p_{kh} = (p_t)_\alpha * \varphi_{\sigma^2}.$$

where $p_\alpha(x) = \alpha^d p(\alpha x)$. In SMLD, $\alpha = 1$ and $\sigma^2 = \int_{kh}^t g(T-s)^2 \, ds$, while in DDPM, $\alpha = e^{\frac{1}{2}\int_{kh}^t g(T-s)^2 \, ds}$ and $\sigma^2 = 1 - e^{-\int_{kh}^t g(T-s)^2 \, ds}$. Now for SMLD,

$$\mathbb{E}\left[\psi_t(z_t)\|\nabla \ln p_{kh}(z_t) - \nabla \ln p_t(z_t)\|^2\right]$$

$$\leq 72L^2\sigma^2 d + 8L^2\sigma^4 \mathbb{E}\left[\psi_t(z_t)\|\nabla \ln p_t(z_t)\|^2\right] \qquad \text{by Lemma C.11}$$

$$\leq 72\sigma^2 L^2 d + \frac{32L^2\sigma^4}{\chi^2(q_t\|p_t)+1} \cdot \mathscr{E}_{p_t}\left(\frac{q_t}{p_t}\right) + 16\sigma^4 L^3 d \qquad \text{by Corollary C.7}$$

$$\leq G_{kh,t}^2\left(\frac{32L^2}{\chi^2(q_t\|p_t)+1} + 16L^3 d\right) + G_{kh,t} \cdot 72L^2 d$$

$$\leq G_{kh,t}^2 \frac{32L^2}{\chi^2(q_t\|p_t)+1} + G_{kh,t} \cdot 76L^2 d,$$

where in the last inequality we use the fact that $g$ is increasing, so that for $h \leq \frac{1}{4Lg(T-kh)^2}$,

$$G_{kh,t}L = \int_{kh}^t g(T-s)^2 \, ds \cdot L \leq h \cdot g(T-kh)^2 \cdot L \leq \frac{1}{4}.$$

Recall that to use Lemma C.11, it suffices that $L \leq \frac{1}{2\alpha^2\sigma^2}$, and so it suffices that $h \leq \frac{1}{4Lg(T-kh)^2}$ in SMLD.

For DDPM, observe that for $h \leq \frac{1}{4g(T-kh)^2}$,

$$\alpha \leq 1 + \int_{kh}^t g(T-s)^2 ds \leq 1 + (t-kh)g(T-kh)^2 \leq 1 + \frac{1}{4}$$

$$\sigma^2 = 1 - e^{-\int_{kh}^t g(T-s)^2 ds} \leq \int_{kh}^t g(T-s)^2 ds \leq (t-kh)g(T-kh)^2 \leq \frac{1}{4}.$$

By Lemma C.12, using the assumption that $L \geq 1$, we obtain

$$
\mathbb{E}\left[\psi_t(z_t) \left\|\nabla \ln p_{kh}(z_t) - \nabla \ln p_t(z_t)\right\|^2\right]
$$

$$
\leq 72\alpha^4 L^2 \sigma^2 d + 4(\alpha + 2\alpha^3 L\sigma^2)^2(\alpha-1)^2 L^2 \mathbb{E}\left[\psi(z_t) \|z_t\|^2\right]
$$

$$
+ 4(\alpha - 1 + 2\alpha^3 L\sigma^2)^2 \mathbb{E}\left[\psi_t(z_t) \|\nabla \ln p_t(z_t)\|^2\right]
$$

$$
\leq 72\alpha^4 L^2 \sigma^2 d + 44 L^2 G_{kh,t}^2 \mathbb{E}\left[\psi(z_t) \|z_t\|^2\right]
$$

$$
+ 100 L^2 G_{kh,t}^2 \mathbb{E}\left[\psi_t(z_t) \|\nabla \ln p_t(z_t)\|^2\right]
$$

$$
\leq 44 L^2 d G_{kh,t}
$$

$$
+ 44 L^2 \left[\frac{2C_t^2}{\chi^2(q_t\|p_t)+1}\mathscr{E}_{p_t}\left(\frac{q_t}{p_t}\right) + \frac{1}{2}\mathbb{E}_{p_t} \|x\|^2 + \frac{1}{2}C_t\right] G_{kh,t}^2
$$

$$
+ 100 L^2 \left[\frac{4}{\chi^2(q_t\|p_t)+1}\mathscr{E}_{p_t}\left(\frac{q_t}{p_t}\right) + 2dL\right] G_{kh,t}^2
$$

$$
\leq L^2 G_{kh,t}\left[G_{kh,t}\left(\frac{88C_t^2 + 400}{\chi^2(q_t\|p_t)+1}\right)\mathscr{E}_{p_t}\left(\frac{q_t}{p_t}\right)\right.
$$

$$
\left. + 44d + G_{kh,t}\left(22(\mathbb{E}_{p_t} \|x\|^2 + C_t) + 200Ld\right)\right]
$$

$$
\leq G_{kh,t}\left[G_{kh,t}\frac{88C_t^2 + 400}{\chi^2(q_t\|p_t)+1}\mathscr{E}_{p_t}\left(\frac{q_t}{p_t}\right) + 6 + 94L^2 d\right],
$$

where we used Lemma C.9 and Corollary C.7. Here, we use the assumption that $h \leq \frac{1}{4g(T-kh)^2(\mathbb{E}_{p_t} \|x\|^2 + C_t)}$. $\qquad\square$

## C.6 Auxiliary Lemmas

In this section, we continue with bounding errors in the form of $\mathbb{E}_{\psi_t q_t} \|u(x)\|^2$. However, we only decompose them into errors which we have already bounded in the previous two sections. The following two lemmas will be directly applied in the proof of Lemma C.4 and Lemma C.5.

**Lemma C.14.** *With the setting of Lemma C.3, we have the following bound of the second moment of estimated score function with respect to $\psi_t q_t$:*

$$
\mathbb{E}\left[\psi_t(z_t) \|s(z_t, T - kh)\|^2\right] \leq \frac{4C_{t,L}G_{kh,t} + 8}{\chi^2(q_t\|p_t)+1} \cdot \mathscr{E}_{p_t}\left(\frac{q_t}{p_t}\right) + 4(\varepsilon_{kh}^2 + C_{d,L} + dL),
$$

*where $C_{t,L}$ and $C_{d,L}$ are constants defined in Lemma C.13.*

*Proof.* Note that by the triangle inequality,

$$
\|s(x, T - kh)\| \leq \|s(x, T - kh) - \nabla \ln \tilde{p}_{T-kh}(x)\|
$$
$$
+ \|\nabla \ln \tilde{p}_{T-kh}(x) - \nabla \ln \tilde{p}_{T-t}(x)\| + \|\nabla \ln \tilde{p}_{T-t}(x)\|,
$$

and hence,

$$
\|s(x, T - kh)\|^2 \leq 4\|s(x, T - kh) - \nabla \ln \tilde{p}_{T-kh}(x)\|^2
$$
$$
+ 4\|\nabla \ln \tilde{p}_{T-kh}(x) - \nabla \ln \tilde{p}_{T-t}(x)\|^2 + 2\|\nabla \ln \tilde{p}_{T-t}(x)\|^2.
$$

Recall that we need to bound this second moment of estimated score function with respect to $\psi_t q_T$. For the first term, as $\|s(x.T - kh) - \nabla \ln p_{kh}(x)\|$ is $\varepsilon_{kh}$-bounded, we have trivial bound that

$$
\mathbb{E}_{\psi_t q_t} \|s(x, T - kh) - \nabla \ln \tilde{p}_{T-kh}(x)\|^2 \leq \varepsilon_{kh}^2.
$$

By Lemma C.13, the second term is bounded by

$$\mathbb{E}_{\psi_t q_t}\left[\|\nabla \ln p_{kh}(z_t) - \nabla \ln p_t(z_t)\|^2\right]$$

$$\leq G_{kh,t} \cdot \left[\frac{C_{t,L}}{\chi^2(q_t\|p_t)+1}G_{kh,t}\mathscr{E}_{p_t}\left(\frac{q_t}{p_t}\right) + C_{d,L}\right]$$

for constant $C_{t,L}$ and $C_{d,L}$ defined in (27) and (28) respectively. The last term is bounded in Corollary C.7 by

$$\mathbb{E}\left[\psi_t(z_t)\|\nabla \ln p_t(z_t)\|^2\right] \leq \frac{4}{\chi^2(q_t\|p_t)+1}\mathscr{E}_{p_t}\left(\frac{q_t}{p_t}\right) + 2dL.$$

Combining these three inequalities, we obtain that for $h \leq \frac{1}{g(T-kh)^2}$,

$$\mathbb{E}\left[\psi_t(z_t)\|s(z_t, T-kh)\|^2\right]$$

$$\leq \frac{4C_{t,L}+8}{\chi^2(q_t\|p_t)+1}G_{kh,t}\mathscr{E}_{p_t}\left(\frac{q_t}{p_t}\right) + 4(\varepsilon_{kh}^2 + C_{d,L} + dL). \qquad \square$$

Now we bound $\mathbb{E}\left[\psi_t(z_t)\|z_t - z_{kh}\|^2\right]$.

**Lemma C.15.** *In the setting of Lemma C.3, if*

$$h \leq \frac{1}{g(T-kh)^2(8L^2 + 20L + 3L_s + 10C_t + \mathbb{E}_{p_t}\|x\|^2)},$$

*then*

$$\mathbb{E}\left[\psi_t(z_t)\|z_t - z_{kh}\|^2\right] \leq \frac{9}{2}G_{kh,t}\left[\frac{\tilde{R}_t}{\chi^2(q_t\|p_t)+1} \cdot \mathscr{E}_{p_t}\left(\frac{q_t}{p_t}\right) + R_{t,kh}\right],$$

*where $\tilde{R}_t$ and $R_d$ are defined in (30) and (31) respectively.*

*Proof.* Note that

$$\|z_t - z_{kh}\|$$

$$= \left\|G_{kh,t}s(z_{kh}, T-kh) - \int_{kh}^t f(z_{kh}, T-s)ds + \int_{kh}^t g(T-s)dw_s\right\|$$

$$\leq G_{kh,t}\|s(z_{kh}, T-kh)\| + \frac{1}{2}\left\|z_{kh}\int_{kh}^t g(T-s)^2 ds\right\| + \left\|\int_{kh}^t g(T-s)dw_s\right\|$$

$$\leq G_{kh,t}\left[\|s(z_{kh}, T-kh)\| + \frac{1}{2}\|z_{kh}\|\right] + \left\|\int_{kh}^t g(T-s)dw_s\right\|$$

$$\leq G_{kh,t}\left[\|s(z_t, T-kh)\| + L_s\|z_t - z_{kh}\| + \frac{1}{2}\|z_t\| + \frac{1}{2}\|z_t - z_{kh}\|\right] + \left\|\int_{kh}^t g(T-s)dw_s\right\|$$

$$= G_{kh,t}\left[\|s(z_t, T-kh)\| + \frac{1}{2}\|z_t\|\right] + \left(L_s + \frac{1}{2}\right)g(T-kh)^2 \cdot h\|z_t - z_{kh}\| + \left\|\int_{kh}^t g(T-s)dw_s\right\|,$$

where the next-to-last line is due to the fact that the estimated score function is $L_s$-Lipschitz. We also use the fact that $g(t)$ is an increasing function and hence $g(T-t) \leq g(T-kh)$ for any $t \in [kh, (k+1)h]$. Hence if $h \leq \frac{1}{3(L_s+1/2)g(T-kh)^2}$, then

$$\|z_t - z_{kh}\| \leq \frac{3}{2}G_{kh,t}\left[\|s(z_t, T-kh)\| + \frac{1}{2}\|z_t\|\right] + \frac{3}{2}\left\|\int_{kh}^t g(T-s)dw_s\right\|.$$

Therefore, by the fact that $(a+b)^2 \leq 2a^2 + 2b^2$ for any $a, b > 0$,

$$\|z_t - z_{kh}\|^2 \leq \frac{9}{2}G_{kh,t}^2\left[2\|s(z_t, T-kh)\|^2 + \frac{1}{2}\|z_t\|^2\right] + \frac{9}{2}\left\|\int_{kh}^t g(T-s)dw_s\right\|^2. \qquad (29)$$

With the results of Lemma C.14 and Lemma C.9, we have

$$2\mathbb{E}\left[\psi_t(z_t)\,\|s(z_t, T - kh)\|^2\right] + \frac{1}{2}\mathbb{E}\left[\psi_t(z_t)\,\|z_t\|^2\right]$$

$$\leq \frac{8C_{t,L}G_{kh,t} + C_t^2 + 16}{\chi^2(q_t\|p_t) + 1} \cdot \mathscr{E}_{p_t}\left(\frac{q_t}{p_t}\right) + 8(\varepsilon_{kh}^2 + C_{d,L} + dL) + \frac{1}{4}\mathbb{E}_{p_t}\|x\|^2 + \frac{1}{4}C_t.$$

Now plugging this and the result of Lemma C.10 into (29), we get that

$$\mathbb{E}\left[\psi_t(z_t)\,\|z_t - z_{kh}\|^2\right] \leq \frac{9}{2}G_{kh,t}^2 \cdot 8(\varepsilon_{kh}^2 + C_{d,L} + dL)$$

$$+ \frac{9}{2}G_{kh,t}^2 \cdot \left[\frac{8C_{t,L}G_{kh,t} + C_t^2 + 16}{\chi^2(q_t\|p_t) + 1} \cdot \mathscr{E}_{p_t}\left(\frac{q_t}{p_t}\right) + \frac{1}{4}\mathbb{E}_{p_t}\|x\|^2 + \frac{1}{4}C_t\right]$$

$$+ 9G_{kh,t} \cdot \left[\frac{8C_t}{\chi^2(q_t\|p_t) + 1} \cdot \mathscr{E}_{p_t}\left(\frac{q_t}{p_t}\right) + d + 8\ln 2\right].$$

Hence, using the assumption on $h$,

$$\mathbb{E}\left[\psi_t(z_t)\,\|z_t - z_{kh}\|^2\right] \leq \frac{9}{2}G_{kh,t}\left[\frac{K_1}{\chi^2(q_t\|p_t) + 1} \cdot \mathscr{E}_{p_t}\left(\frac{q_t}{p_t}\right) + K_2\right],$$

where

$$K_1 := 8C_{t,L}G_{kh,t}^2 + (C_t^2 + 16)G_{kh,t} + 16C_t$$

$$\leq 8(88C_t^2 + 400L^2)\frac{1}{400L + 100C_t^2} + (C_t^2 + 16)\frac{1}{20L + 10C_t} + 8C_t$$

$$\leq 8 + C_t + 1 + 8C_t = 9(C_t + 1)$$

and

$$K_2 := \left[\frac{1}{4}(\mathbb{E}_{p_t}\|x\|^2 + C_t) + 8(\varepsilon_{kh}^2 + C_{d,L} + dL)\right]G_{kh,t} + 2d + 16\ln 2$$

$$\leq \left[\frac{1}{4}(\mathbb{E}_{p_t}\|x\|^2 + C_t) + 8(\varepsilon_{kh}^2 + 256L^2d + dL)\right]\left(\frac{1}{\mathbb{E}_{p_t}\|x\|^2 + C_t + 8L^2}\right) + 2d + 16\ln 2$$

$$\leq \frac{1}{4} + 300d + 16\ln 2 \leq 300d + 12.$$

Hence the lemma holds by setting

$$\tilde{R}_t = 9(C_t + 1), \tag{30}$$
$$R_d = 300d + 12. \tag{31}$$

$\square$

## C.7   Proof of Theorem 3.1

We state a more precise version of Theorem 3.1. The structure of the proof is similar to that of Theorem 2.1.

**Theorem C.16** (Predictor with $L^2$-accurate score estimate, DDPM). *Let $p_{\text{data}} : \mathbb{R}^d \to \mathbb{R}$ be a probability density satisfying Assumption 1 with $M_2 = O(d)$, and let $\widetilde{p}_t$ be the distribution resulting from evolving the forward SDE according to DDPM with $g \equiv 1$. Suppose furthermore that $\nabla \ln \widetilde{p}_t$ is $L$-Lipschitz for every $t \geq 0$, and that each $s(\cdot, t)$ satisfies Assumption 2. Then if*

$$\varepsilon = O\left(\frac{\varepsilon_{\text{TV}}\varepsilon_\chi^3}{(C_{\text{LS}} + d)C_{\text{LS}}^{5/2}(L \vee L_s)^2(\ln(C_{\text{LS}}d) \vee C_{\text{LS}}\ln(1/\varepsilon_{\text{TV}}^2))}\right),$$

*running* (P) *starting from $p_{\text{prior}}$ for time $T = \Theta\left(\ln(C_{\text{LS}}d) \vee C_{\text{LS}}\ln\left(\frac{1}{\varepsilon_{\text{TV}}}\right)\right)$ and step size $h = \Theta\left(\frac{\varepsilon_\chi^2}{C_{\text{LS}}(C_{\text{LS}} + d)(L \vee L_s)^2}\right)$ results in a distribution $q_T$ such that $q_T$ is $\varepsilon_{\text{TV}}$-far in TV distance from a distribution $\bar{q}_T$, where $\bar{q}_T$ satisfies $\chi^2(\bar{q}_T\|p_{\text{data}}) \leq \varepsilon_\chi^2$. In particular, taking $\varepsilon_\chi = \varepsilon_{\text{TV}}$, we have $\text{TV}(q_T\|p_{\text{data}}) \leq 2\varepsilon_{\text{TV}}$.*

*Proof of Theorem C.16.* We first define the bad sets where the error in the score estimate is large,

$$B_t := \{\|\nabla \ln p_t(x) - s(x, T - t)\| > \varepsilon_1\} \tag{32}$$

for some $\varepsilon_1$ to be chosen.

Given $t \geq 0$, let $t_- = h \lfloor \frac{t}{h} \rfloor$. Given a bad set $B$, define the interpolated process by

$$d\bar{z}_t = -\left[f(z_{t_-}, T - t) - g(T - t)^2 b(z_{kh}, T - kh)\right] dt + g(T - t) dw_t, \tag{33}$$

where $b(z, t) = \begin{cases} s(z, t), & z \notin B_t \\ \nabla \ln p_t(z), & z \in B_t \end{cases}$.

In other words, simulate the reverse SDE using the score estimate as long as the point is in the good set (for the current $p_t$) at the previous discretization step, and otherwise use the actual gradient $\nabla \ln p_t$. Let $\bar{q}_t$ denote the distribution of $\bar{z}_t$ when $\bar{z}_0 \sim q_0$; note that $q_{nh}$ is the distribution resulting from running LMC with estimate $b$ for $n$ steps and step size $h$. Note that this process is defined only for purposes of analysis, as we do not have access to $\nabla \ln p_t$.

We can couple this process with the predictor algorithm using $s$ so that as long as $x_{mh} \notin B_{mh}$, the processes agree, thus satisfying condition 1 of Theorem 4.1.

Then by Chebyshev's inequality,

$$P(B_t) \leq \left(\frac{\varepsilon}{\varepsilon_1}\right)^2 =: \delta.$$

Let $T = Nh$, and let $K_\chi = \chi^2(q_0||p_0)$. Then by Theorem 4.3,

$$\chi^2(\bar{q}_{kh}||p_{kh}) = \exp\left(-\frac{kh}{16C_{\text{LS}}}\right) \chi^2(q_0||p_0) + O\left(C_{\text{LS}}(\varepsilon_1^2 + (L_s^2 + L^2 d)h)\right)$$

$$= \exp\left(-\frac{kh}{4C_{\text{LS}}}\right) \chi^2(\mu_0||p) + O(1).$$

For this to be bounded by $\varepsilon_\chi^2$, it suffices for the terms to be bounded by $\frac{\varepsilon_\chi^2}{2}, \frac{\varepsilon_\chi^2}{4}, \frac{\varepsilon_\chi^2}{4}$; this is implied by

$$T \geq 32 C_{\text{LS}} \ln\left(\frac{2K_\chi}{\varepsilon_\chi^2}\right) =: T_{\min}$$

$$h = O\left(\frac{\varepsilon_\chi^2}{C_{\text{LS}}(C_{\text{LS}} + d)(L \vee L_s)^2}\right)$$

$$\varepsilon_1 = O\left(\frac{\varepsilon_\chi}{\sqrt{C_{\text{LS}}}}\right).$$

(We choose $h$ so that the condition in Theorem 4.3 is satisfied; note $\varepsilon_\chi \leq 1$.) By Theorem 4.1,

$$\text{TV}(q_{nh}, \bar{q}_{nh}) \leq \sum_{k=0}^{n-1} (1 + \chi^2(q_{kh}||p))^{1/2} P(B_{kh})^{1/2}$$

$$\leq \left(\sum_{k=0}^{n-1} \exp\left(-\frac{kh}{32C_{\text{LS}}}\right) \chi^2(q_0||p)^{1/2} + O(1)\right) \delta^{1/2}$$

$$\leq \left(\left(\sum_{k=0}^{\infty} \exp\left(-\frac{kh}{32C_{\text{LS}}}\right) K_\chi\right) + O(n)\right) \frac{\varepsilon}{\varepsilon_1}$$

$$\leq \frac{\varepsilon}{\varepsilon_1}\left(\frac{64C_{\text{LS}}}{h} K_\chi + O(n)\right).$$

In order for this to be $\leq \varepsilon_{\text{TV}}$, it suffices for

$$\varepsilon \leq \varepsilon_1 \varepsilon_{\text{TV}} \cdot O\left(\frac{1}{n} \wedge \frac{h}{C_{\text{LS}} K_\chi}\right).$$

Supposing that we run for time $T = \Theta(T_{\min})$, we have that $n = \frac{T}{h} = O\left(\frac{C_T T_{\min}}{h}\right)$. Thus it suffices for

$$
\varepsilon = \varepsilon_1 \varepsilon_{\mathrm{TV}} \cdot O\left(\frac{h}{T_{\min}} \wedge \frac{h}{32 C_{\mathrm{LS}} K_\chi}\right)
$$

$$
= O\left(\frac{\varepsilon_\chi}{\sqrt{C_{\mathrm{LS}}}} \cdot \varepsilon_{\mathrm{TV}} \cdot \frac{\varepsilon_\chi^2}{C_{\mathrm{LS}}(C_{\mathrm{LS}}+d)(L \vee L_s)^2}\left(\frac{1}{C_{\mathrm{LS}}\ln(2K_\chi/\varepsilon_\chi^2)} \wedge \frac{1}{C_{\mathrm{LS}} K_\chi}\right)\right)
$$

$$
= O\left(\frac{\varepsilon_{\mathrm{TV}} \varepsilon_\chi^3}{C_{\mathrm{LS}}^{5/2}(C_{\mathrm{LS}}+d)(L \vee L_s)^2(\ln(2K_\chi/\varepsilon_\chi^2) \vee K_\chi)}\right).
$$

Finally, note that for $T = \Omega(\ln(C_{\mathrm{LS}}d))$, we have $K_\chi = O(1)$ by Lemma E.9. Substituting $K_\chi = O(1)$ then gives the desired bound. $\qquad \square$

## C.8   Proof of Theorem 3.2

We now prove the main theorem on the predictor-corrector algorithm with $L^2$-accurate score estimate.

**Theorem 3.2** (Predictor-corrector with $L^2$-accurate score estimate). *Keep the setup of Theorem 3.1. Then for $\varepsilon_{\mathrm{TV}}^3 = O\left(\frac{1}{(1+L_s/L)^2(1+C_{\mathrm{LS}}/d)(\ln(C_{\mathrm{LS}}d)\vee C_{\mathrm{LS}})}\right)$, if*

$$
\varepsilon = O\left(\frac{\varepsilon_{\mathrm{TV}}^4}{dL^2 C_{\mathrm{LS}}^{5/2} \ln(1/\varepsilon_\chi^2)}\right), \tag{5}
$$

*then Algorithm 2 with appropriate choices of $T = \Theta\left(\ln(C_{\mathrm{LS}}d) \vee C_{\mathrm{LS}} \log\left(\frac{1}{\varepsilon_{\mathrm{TV}}}\right)\right)$, $N_m$, corrector step sizes $h_m$ and predictor step size $h$, produces a sample from a distribution $q_T$ such that $\mathrm{TV}(q_T, p_{\mathrm{data}}) < \varepsilon_{\mathrm{TV}}$.*

For simplicity, we consider the predictor-corrector algorithm in the case where all the corrector steps are at the end (but see the discussion following the proof for the general case). The result will follow from chaining together the guarantee on the predictor algorithm (Theorem C.16) and LMC (Theorem 2.1).

*Proof of Theorem 3.2.* Let $M = T/h$. We take $h = \Theta\left(\frac{1}{(L \vee L_s)^2 C_{\mathrm{LS}}(C_{\mathrm{LS}}+d)}\right)$, number of corrector steps $N_0 = \cdots = N_{T/h-1} = 0$ and $N_M = T_c/h_M$, where $T_c = \Theta\left(C_{\mathrm{LS}}\ln\left(\frac{2}{\varepsilon_\chi^2}\right)\right)$ and $h_M = \Theta\left(\frac{\varepsilon_\chi^2}{dL^2 C_{\mathrm{LS}}}\right)$. Let the distribution of $z_{T,0}$ be $q_{T,0}$. By Theorem C.16, if $T = \Theta(\ln(C_{\mathrm{LS}}d) \vee C_{\mathrm{LS}}\ln(1/\varepsilon_{\mathrm{TV}}))$, then

$$
\varepsilon = O\left(\frac{\varepsilon_{\mathrm{TV}}}{(L \vee L_s)^2(C_{\mathrm{LS}}+d)C_{\mathrm{LS}}^{5/2}(\ln(C_{\mathrm{LS}}d) \vee C_{\mathrm{LS}}\ln(1/\varepsilon_{\mathrm{TV}}))}\right),
$$

then there exists $\bar{q}_{T,0}$ such that $\mathrm{TV}(q_{T,0}, \bar{q}_{T,0}) = \varepsilon_{\mathrm{TV}}/2$ and $\chi^2(\bar{q}_{T,0}\|p_{\mathrm{data}}) = 1$. Then using Theorem 2.1 with $\varepsilon_{\mathrm{TV}} \leftarrow \varepsilon_{\mathrm{TV}}/2$ and $K_\chi = 1$, plus the triangle inequality gives that if

$$
\varepsilon = O\left(\frac{\varepsilon_{\mathrm{TV}} \varepsilon_\chi^3}{dL^2 C_{\mathrm{LS}}^{5/2} \ln(1/\varepsilon_{\mathrm{TV}})}\right),
$$

then there is $\bar{q}_T$ such that $\mathrm{TV}(q_T, \bar{q}_T) = \varepsilon_{\mathrm{TV}}$ and $\chi^2(\bar{q}_T\|p_{\mathrm{data}}) = \varepsilon_\chi^2$. Finally, setting $\varepsilon_{\mathrm{TV}}, \varepsilon_\chi \leftarrow \varepsilon_{\mathrm{TV}}/2$ gives $\mathrm{TV}(q_T, p_{\mathrm{data}}) \leq \varepsilon_{\mathrm{TV}}$.

We note that for $\varepsilon_{\mathrm{TV}}^3 = O\left(\frac{1}{(1+L_s/L)^2(1+C_{\mathrm{LS}}/d)(\ln(C_{\mathrm{LS}}d)\vee C_{\mathrm{LS}})}\right)$, the second condition on $\varepsilon$ is more constraining, giving the theorem. $\qquad \square$

**Remark.** *We can also analyze a setting where predictor and corrector steps are interleaved; for instance, if $N = 1$, then interleaving the one-step inequalities in Theorem 4.2 and 4.3 gives a recurrence*

$$\chi^2(q_{(k+1)h,0}\|p_{(k+1)h}) \leq \exp\left(-\frac{h_{\text{pred}}}{16C_{\text{LS}}}\right)\chi^2(q_{kh,1}\|p_{kh}) + O(dL^2h^2 + \varepsilon_1^2 h)$$

$$\chi^2(q_{(k+1)h,1}\|p_{(k+1)h)}) \leq \exp\left(-\frac{h_{\text{corr}}}{4C_{\text{LS}}}\right)\chi^2(q_{(k+1)h,0}\|p_{(k+1)h}) + O(\varepsilon_1^2 h + (L_s^2 + L^2 d)h^2);$$

*we can then follow the proof of Theorem 3.1. While this does not improve the parameter dependence under the assumptions of Theorem 3.2, it can potentially allow for larger step sizes (beyond what is allowed by Theorem 3.1), as error accrued in the predictor step can be exponentially damped by the corrector step.*

## D    Stationary distribution of LD with score estimate can be arbitrarily far away

We show that the stationary distribution of Langevin dynamics with $L^2$-accurate score estimate can be arbitrarily far from the true distribution. We can construct a counterexample even in one dimension, and take the true distribution as a standard Gaussian $p(x) = \frac{1}{\sqrt{2\pi}}e^{-x^2/2}$. We will take the score estimate to also be in the form $\nabla \ln q$, so that the stationary distribution of LMC with the score estimate is $q$. The main idea of the construction is to set $q$ to disagree with $p$ only in the tail of $p$, where it has a large mode; this error will fail to be detected under $L^2(p)$.

**Theorem D.1.** *Let $p$ be the density function of $N(0,1)$. There exists an absolute constant $C$ such that given any $\varepsilon > 0$, there exists a distribution $q$ such that*

1. *$\ln q$ is $C$-smooth.*

2. *$\mathbb{E}_p[\|\nabla \ln p - \nabla \ln q\|^2] < \varepsilon$*

3. *$\mathrm{TV}(p,q) > 1 - \varepsilon$.*

*Proof.* Take a smooth non-negative function $g$ supported on $[-1, 1]$, with $\max|g''| \leq c$ and $g(0) = 1$. We consider a family of distributions for $L > 0$ with density

$$q_L(x) \propto e^{-V_L(x)}, \qquad \text{and} \quad V_L(x) := \frac{x^2}{2} - L^2 g\left(\frac{2}{L}(x - L)\right).$$

Thus the score function for $q_L$ is given by

$$V_L'(x) = x - (2L)g'\left(\frac{2}{L}(x - L)\right).$$

We compute the $L^2(p)$ error between the score functions associated with $p$ and $q_L$.

$$\begin{aligned}
\mathbb{E}_p(V_L'(x) - x)^2 &= \frac{1}{\sqrt{2\pi}}\int_{-\infty}^{\infty}(2L)^2\left|g'\left(\frac{2}{L}(x - L)\right)\right|^2 e^{-x^2/2}\, dx \\
&\leq \frac{1}{\sqrt{2\pi}}(2L)^2 e^{-L^2/8}\int_{-\infty}^{\infty}\left|g'\left(\frac{2}{L}(x - L)\right)\right|^2 dx \\
&= \frac{1}{\sqrt{2\pi}}2L^3 e^{-L^2/8}\int_{-\infty}^{\infty}|g'(y)|^2\, dy,
\end{aligned}$$

where in the first inequality we have used that $g(\frac{2}{L}(x - L))$ has support $[\frac{L}{2}, \frac{3L}{2}]$, since $g$ has support $[-1, 1]$. Thus the $L^2(p)$-error of the score function goes to 0 as $L \to \infty$.

Moreover, as

$$\left|V_L''(x)\right| = \left|1 - 4g''\left(\frac{2}{L}(x - L)\right)\right| \leq 1 + 4\max_y\left|g''(y)\right| \leq 1 + 4c,$$

the distribution $q_L$ satisfies the required smoothness (Lipschitz score) assumption. Note that $q_L$ has a large mode concentrated at $x = L$ as

$$V_L(L) = \frac{L^2}{2} - L^2 g(0) = -\frac{L^2}{2},$$

while $p$ has vanishing density there, which is in fact the reason that $L^2(p)$-loss of the score estimate is not able to detect the difference between the two distributions. As the height (and width) of the mode becomes arbitrarily large compared to $x = 0$, we have $q_L([\frac{L}{2}, \frac{3L}{2}]) \to 1$, whereas $p_L([\frac{L}{2}, \frac{3L}{2}]) \to 0$. Hence $\mathrm{TV}(p_L, q_L) \to 1$. $\qquad\square$

# E  Useful facts

In this section, we collect some facts and technical lemmas used throughout the paper.

## E.1  Facts about probability distributions

Given a probability measure $P$ on $\mathbb{R}^d$ with density $p$, we say that a Poincaré inequality (PI) holds with constant $C_{\mathrm{P}}$ if for any probability measure $q$,

$$\chi^2(q\|p) \leq C_{\mathrm{P}} \mathscr{E}_p\left(\frac{q}{p}\right) := C_{\mathrm{P}} \int_{\mathbb{R}^d} \left\|\nabla \frac{q(x)}{p(x)}\right\|^2 p(x) dx. \tag{PI}$$

Alternatively, for any $C^1$ function $f$,

$$\mathrm{Var}_p(f) \leq C_{\mathrm{P}} \int_{\mathbb{R}^d} \|\nabla f\|^2 \, p(x) \, dx.$$

We say that a log-Sobolev inequality (LSI) holds with constant $C_{\mathrm{LS}}$ if for any probability measure $q$,

$$\mathrm{KL}(q\|p) \leq \frac{C_{\mathrm{LS}}}{2} \int_{\mathbb{R}^d} \left\|\nabla \ln \frac{q(x)}{p(x)}\right\|^2 q(x) dx. \tag{LSI}$$

We call the Poincaré constant and log-Sobolev constant the smallest $C_{\mathrm{P}}$, $C_{\mathrm{LS}}$ for which the inequalities hold for all $q$. If $p$ satisfies a log-Sobolev inequality with constant, then $p$ satisfies a Poincaré inequality with the same constant; hence the Poincaré constant is at most the log-Sobolev constant, $C_{\mathrm{P}} \leq C_{\mathrm{LS}}$. If $p \propto e^{-V}$ is $\alpha$-strongly log-concave, that is, $V \succeq \alpha I_d$, then $p$ satisfies a log-Sobolev inequality with constant $1/\alpha$.

We collect some properties of distributions satisfying LSI or PI.

**Lemma E.1** (Herbst, Sub-exponential and sub-gaussian concentration given log-Sobolev inequality, [BGL13, Pr. 5.4.1])**.** *Suppose that $\mu$ satisfies a log-Sobolev inequality with constant $C_{\mathrm{LS}}$. Let $f$ be a 1-Lipschitz function. Then*

1. *(Sub-exponential concentration) For any $t \in \mathbb{R}$,*

$$\mathbb{E}_\mu e^{tf} \leq e^{t\mathbb{E}_\mu f + \frac{C_{\mathrm{LS}} t^2}{2}}.$$

2. *(Sub-gaussian concentration) For any $t \in \left[0, \frac{1}{C_{\mathrm{LS}}}\right)$,*

$$\mathbb{E}_\mu e^{\frac{tf^2}{2}} \leq \frac{1}{\sqrt{1 - C_{\mathrm{LS}} t}} \exp\left[\frac{t}{2(1 - C_{\mathrm{LS}} t)} (\mathbb{E}_\mu f)^2\right].$$

**Lemma E.2** (Gaussian measure concentration for LSI, [BGL13, §5.4.2])**.** *Suppose that $\mu$ satisfies a log-Sobolev inequality with constant $C_{\mathrm{LS}}$. Let $f$ be a $L$-Lipschitz function. Then*

$$\mu\left(|f - \mathbb{E}_\mu f| \geq r\right) \leq 2e^{-\frac{r^2}{2C_{\mathrm{LS}} L^2}}.$$

**Lemma E.3** ([GLR18, Lemma G.10]). *Let $V : \mathbb{R}^d \to \mathbb{R}$ be a $\alpha$-strongly convex and $\beta$-smooth function and let $P$ be a probability measure with density function $p(x) \propto e^{-V(x)}$. Let $x^* = \operatorname{argmin}_x V(x)$ and $\overline{x} = \mathbb{E}_P x$. Then*

$$\|x^* - \overline{x}\| \leq \sqrt{\frac{d}{\alpha}} \left( \sqrt{\ln\left(\frac{\beta}{\alpha}\right)} + 5 \right). \tag{34}$$

**Theorem E.4** ( [BL02, Theorem 5.1], [Har04]). *Suppose the $d$-dimensional gaussian $N(0, \Sigma)$ has density $\gamma$. Let $p = h \cdot \gamma$ be a probability density.*

*1. If $h$ is log-concave, and $g$ is convex, then*

$$\int_{\mathbb{R}^d} g(x - \mathbb{E}_p x) p(x) \, dx \leq \int_{\mathbb{R}^d} g(x)\gamma(x) \, dx.$$

*2. If $h$ is log-convex,[1] and $g(x) = \langle x, y \rangle^\alpha$ for some $y \in \mathbb{R}^d$, $\alpha > 0$, then*

$$\int_{\mathbb{R}^d} g(x - \mathbb{E}_p x) p(x) \, dx \geq \int_{\mathbb{R}^d} g(x)\gamma(x) \, dx.$$

**Lemma E.5.** *Let $P$ be a probability measure on $\mathbb{R}^d$ with density function $p$ such that $\ln p$ is $C^1$ and $L$-smooth and $P$ satisfies a Poincaré inequality with constant $C_{\mathrm{P}}$. Then $LC_{\mathrm{P}} \geq 1$.*

*Proof.* By the Poincaré inequality and Lemma E.4(2), since $p$ is equal to the density of $N(0, \frac{1}{L}I_d)$ multiplied by a log-convex function,

$$C_{\mathrm{P}} \geq \mathbb{E}_P(x_1 - \mathbb{E}_P x_1)^2 \geq \mathbb{E}_{N(0, \frac{1}{L}I_d)} x_1^2 = \frac{1}{L}. \qquad \square$$

## E.2 Lemmas on SMLD and DDPM

We give bounds on several quantities associated with the SMLD and DDPM processes at time $t$: the log-Sobolev constants (Lemma E.7), the second moment (Lemma E.8), and the warm start parameter (Lemma E.9).

First, we note that for SMLD and DDPM, the conditional distribution of $\widetilde{x}_t$ given $\widetilde{x}_0$ is

$$\text{SMLD:} \qquad \widetilde{x}_t | \widetilde{x}_0 \sim N\left( x(0), \int_0^t g(s)^2 \, ds \cdot I_d \right)$$

$$\text{DDPM:} \qquad \widetilde{x}_t | \widetilde{x}_0 \sim N\left( x(0) e^{-\frac{1}{2}\int_0^t g(s)^2 \, ds}, (1 - e^{-\int_0^t g(s)^2 \, ds}) I_d \right).$$

Hence

$$\widetilde{p}_t^{\mathrm{SMLD}} = p_0 * N\left( 0, \int_0^t g(s)^2 \, ds \cdot I_d \right) \tag{35}$$

$$\widetilde{p}_t^{\mathrm{DDPM}} = M_{e^{-\frac{1}{2}\int_0^t g(s)^2 \, ds}} {}_\# p_0 * N(0, (1 - e^{-\int_0^t g(s)^2 \, ds}) I_d) \tag{36}$$

where $M_c$ is multiplication by $c$.

**Lemma E.6** ([Cha04]). *Let $p, p'$ be two probability densities on $\mathbb{R}^d$. If $p$ and $p'$ satisfy log-Sobolev inequalities with constants $C_{\mathrm{LS}}$ and $C'_{\mathrm{LS}}$, then $p * p'$ satisfies a log-Sobolev inequality with constant $C_{\mathrm{LS}} + C'_{\mathrm{LS}}$.*

**Lemma E.7** (Log-Sobolev constant for SMLD and DDPM). *Let $\widetilde{p}_t^{\mathrm{SMLD}}$ and $\widetilde{p}_t^{\mathrm{DDPM}}$ denote the distribution of the SMLD/DDPM processes at time $t$, when started at $p_0$. Let $C_{\mathrm{LS}}$ be the log-Sobolev constant of $p_0$. Then*

$$C_{\mathrm{LS}}(\widetilde{p}_t^{\mathrm{SMLD}}) \leq C_{\mathrm{LS}} + \int_0^t g(s)^2 \, ds$$

$$C_{\mathrm{LS}}(\widetilde{p}_t^{\mathrm{DDPM}}) \leq (C_{\mathrm{LS}} - 1) e^{-\int_0^t g(s)^2 \, ds} + 1 \leq \max\{C_{\mathrm{LS}}, 1\}.$$

---

[1]Note that the sign is flipped in the theorem statement in [BL02].

Note that the analogous statement for the Poincaré constant $C_P$ holds for Lemma E.6 and E.7.

*Proof.* Note that if $\mu$ has log-Sobolev constant $C_{\mathrm{LS}}$ and $T$ is a smooth $L$-Lipschitz map, then $T_{\#}\mu$ has log-Sobolev constant $\leq L^2 C_{\mathrm{LS}}$. Applying Lemma E.6 to (35) and (36) then finishes the proof. $\qquad\square$

**Lemma E.8** (Second moment for SMLD and DDPM). *Suppose that $\tilde{p}_0$ has finite second moment, then for $t \in [0, T]$:*

$$\mathbb{E}_{\tilde{p}_t}\left[\|x\|^2\right] = \mathbb{E}_{\tilde{p}_0}\left[\|x\|^2\right] + d\beta(t) \qquad\qquad \text{in SMLD,}$$

$$\mathbb{E}_{\tilde{p}_t}\left[\|x\|^2\right] = e^{-\beta(t)}\mathbb{E}_{\tilde{p}_0}\left[\|x\|^2\right] + d(1 - e^{-\beta(t)}) \leq \max\left\{\mathbb{E}_{\tilde{p}_0}\left[\|x\|^2\right], d\right\} \qquad \text{in DDPM,}$$

*where $\beta(t) = \int_0^t g(s)^2 ds$.*

*Proof.* Recall that in SMLD, $\tilde{x}_t \sim N(\tilde{x}_0, \beta(t) \cdot I_d)$. Let $y \sim N(0, \beta(t) \cdot I_d)$ be independent of $\tilde{x}_0$. Then

$$\mathbb{E}_{\tilde{p}_t}\left[\|x\|^2\right] = \mathbb{E}\left[\|\tilde{x}_0 + y\|^2\right] = \mathbb{E}\left[\|\tilde{x}_0\|^2\right] + \mathbb{E}\left[\|y\|^2\right] = \mathbb{E}\left[\|\tilde{x}_0\|^2\right] + d\beta(t).$$

In DDMP, $\tilde{x}_t \sim N(e^{-\frac{1}{2}\beta(t)}\tilde{x}_0, (1 - e^{-\beta(t)}) \cdot I_d)$. Choose $y \sim N(0, (1 - e^{-\beta(t)}) \cdot I_d)$ independent of $\tilde{x}_0$, then

$$\mathbb{E}_{\tilde{p}_t}\left[\|x\|^2\right] = \mathbb{E}\left[\left\|e^{-\frac{1}{2}\beta(t)}\tilde{x}_0 + y\right\|^2\right] = \mathbb{E}\left[\left\|e^{-\frac{1}{2}\beta(t)}\tilde{x}_0\right\|^2\right] + \mathbb{E}\left[\|y\|^2\right]$$

$$= e^{-\beta(t)}\mathbb{E}\left[\|\tilde{x}_0\|^2\right] + d(1 - e^{-\beta(t)}). \qquad\square$$

**Lemma E.9** (Warm start for SMLD and DDPM). *Suppose that $p$ has log-Sobolev constant at most $C_{\mathrm{LS}}$ and $\|\mathbb{E}_{y\sim p}y\| \leq M_1$. Let $\varphi_{\sigma^2}$ denote the density of $N(0, \sigma^2 I_d)$. Then for any $\sigma^2$,*

$$\chi^2(\varphi_{\sigma^2}\|p * \varphi_{\sigma^2}) \leq 4\exp\left(\frac{d(2M_1 + 8C_{\mathrm{LS}})}{\sigma^2}\right)$$

*Hence, letting $\sigma^2_{\mathrm{SMLD}} = \int_0^t g(s)^2\, ds$ and $\sigma^2_{\mathrm{DDPM}} = 1 - e^{-\int_0^t g(s)^2\, ds}$,*

$$\chi^2(\varphi_{\sigma^2_{\mathrm{SMLD}}}\|\widehat{p}_t^{\mathrm{SMLD}}) \leq 4\exp\left(\frac{d(2M_1 + 8C_{\mathrm{LS}})}{\sigma^2_{\mathrm{SMLD}}}\right)$$

$$\chi^2(\varphi_{\sigma^2_{\mathrm{DDPM}}}\|\widehat{p}_t^{\mathrm{DDPM}}) \leq 4\exp\left(\frac{d\left(2e^{-\frac{1}{2}\int_0^t g(s)^2\, ds}M_1 + 8e^{-\int_0^t g(s)^2\, ds}C_{\mathrm{LS}}\right)}{\sigma^2_{\mathrm{DDPM}}}\right).$$

*Proof.* Let $R_x = (M_1 + 2\sqrt{C_{\mathrm{LS}}})\|x\|$. For a fixed $x$, note that $\mathbb{E}_{y\sim p}\langle y, x\rangle \leq \|\mathbb{E}_{y\sim p}y\|\|x\| \leq M_1\|x\|$ by assumption. Then by Lemma E.2,

$$\mathbb{P}(\langle y, x\rangle \geq R_x) \leq \mathbb{P}(|\langle y, x\rangle - \mathbb{E}_{y\sim p}\langle y, x\rangle| \geq 2\sqrt{C_{\mathrm{LS}}}\|x\|) \leq 2e^{-\frac{(2\sqrt{C_{\mathrm{LS}}}\|x\|)^2}{2C_{\mathrm{LS}}\|x\|^2}} \leq 2e^{-2} \leq \frac{1}{2}.$$

Hence

$$(p * \varphi_{\sigma^2})(x) = \left(\frac{1}{2\pi\sigma^2}\right)^{d/2} \int_{\mathbb{R}^d} e^{-\frac{\|x+y\|^2}{2\sigma^2}} p(y)\, dy$$

$$\geq \left(\frac{1}{2\pi\sigma^2}\right)^{d/2} e^{-\frac{\|x\|^2}{2\sigma^2}} \int_{\mathbb{R}^d} e^{-\frac{\langle x,y\rangle}{\sigma^2}} p(y)\, dy$$

$$\geq \left(\frac{1}{2\pi\sigma^2}\right)^{d/2} e^{-\frac{\|x\|^2}{2\sigma^2}} \int_{\langle y,x\rangle \leq R_x} e^{-\frac{\langle x,y\rangle}{\sigma^2}} p(y)\, dy$$

$$\geq \left(\frac{1}{2\pi\sigma^2}\right)^{d/2} e^{-\frac{\|x\|^2}{2\sigma^2}} \int_{\langle y,x\rangle \leq R_x} e^{-(M_1\|x\|+2\sqrt{C_{\mathrm{LS}}}\|x\|)/\sigma^2} p(y)\, dy$$

$$\geq \left(\frac{1}{2\pi\sigma^2}\right)^{d/2} e^{-\frac{\|x\|^2}{2\sigma^2}} e^{-\frac{\|x\|^2}{8\sigma^2 d} - \frac{2M_1^2 d}{\sigma^2} - \frac{\|x\|^2}{8\sigma^2 d} - \frac{8C_{\mathrm{LS}} d}{\sigma^2}} \int_{\langle y,x\rangle \leq R_x} p(y)\, dy$$

$$\geq \left(\frac{1}{2\pi\sigma^2}\right)^{d/2} e^{-\frac{\|x\|^2}{2\sigma^2\left(1-\frac{1}{2d}\right)^{-1}}} e^{-\frac{d(8C_{\mathrm{LS}}+2M_1^2)}{\sigma^2}} \cdot \frac{1}{2}$$

$$\geq \frac{1}{2} e^{-\frac{d(8C_{\mathrm{LS}}+2M_1^2)}{\sigma^2}} \left(1-\frac{1}{2d}\right)^{d/2} \varphi_{\frac{\sigma^2}{1-\frac{1}{2d}}}.$$

Using the fact that $\chi^2(N(0,\Sigma_2)\|N(0,\Sigma_1)) = \frac{|\Sigma_1|^{1/2}}{|\Sigma_2|}|(2\Sigma_2^{-1}-\Sigma_1^{-1})|^{-\frac{1}{2}} - 1$, we have

$$\chi^2(\varphi_{\sigma^2}\|p * \varphi_{\sigma^2}) + 1 \leq 2 \cdot e^{\frac{d(8C_{\mathrm{LS}}+2M_1^2)}{\sigma^2}} \left(1-\frac{1}{2d}\right)^{-\frac{d}{2}} \left[\chi^2\left(\varphi_{\sigma^2}\|\varphi_{\frac{\sigma^2}{1-\frac{1}{2d}}}\right) + 1\right]$$

$$= 2 \cdot e^{\frac{d(8C_{\mathrm{LS}}+2M_1^2)}{\sigma^2}} \left(1-\frac{1}{2d}\right)^{-\frac{d}{2}} \left(1-\frac{1}{2d}\right)^{-\frac{d}{2}} \left(2-\left(1-\frac{1}{2d}\right)\right)^{-\frac{d}{2}}$$

$$\leq 2 \cdot e^{\frac{d(8C_{\mathrm{LS}}+2M_1^2)}{\sigma^2}} \left(1-\frac{1}{2d}\right)^{-d} \leq 4 e^{\frac{d(8C_{\mathrm{LS}}+2M_1^2)}{\sigma^2}}.$$

The corollary inequalities then follow from (35) and (36), where for DDPM, we use the fact that $M_{e^{-\frac{1}{2}\int_0^t g(s)^2\, ds}\#}p_0$ has mean $e^{-\frac{1}{2}\int_0^t g(s)^2\, ds} \cdot \mathbb{E}_p x$ and log-Sobolev constant $(e^{-\frac{1}{2}\int_0^t g(s)^2\, ds})^2 C_{\mathrm{LS}}$.

$\square$