# OpenReview forum: "Convergence for score-based generative modeling with polynomial complexity"
_NeurIPS.cc/2022/Conference — NeurIPS 2022 Accept_

### Official Review · Reviewer_VLZN · 2022-07-08

**Rating:** 7
**Confidence:** 4
**Soundness:** 3 good
**Presentation:** 3 good
**Contribution:** 3 good

**Summary:**

In this paper, the authors study the convergence of score-based generative models (SGMs) (also called denoising diffusion models) w.r.t. the total variation distance and under the assumption that a logarithmic Sobolev inequality holds for the distribution. They also assume that the score network is close to the true score w.r.t. the $L_2$ distance and not the $L_\infty$ distance contrary to previous works. They provide quantitative convergence for the annealed algorithm first introduced in [1] and the time-reverse algorithms considered in [2], including the predictor-corrector scheme. The paper is concluded by a short presentation of the theoretical framework including an explanation of the translations of $L_2$ errors to $L_\infty$ errors. The analysis in the case of the $L_\infty$ error follows the lines of [3].

* [1] -- Song and Ermon (2019) -- Generative modeling by estimating gradients of the data distribution
* [2] -- Song et al. (2021) -- Score-based generative modeling through stochastic differential equations
* [3] -- Chewi et al. (2021) -- Analysis of Langevin Monte Carlo from Poincare to Log-Sobolev

**Questions:**

In this section, I wrap up my main comments and questions for the authors:

* What is $C_T$ in Theorem 2.1 ? It is a bit hard to know.

* In Theorem 2.2, is the sequence of $(\sigma_i)_{i=1}^M$ explicit? It seems so and in fact it seems to be geometric but it would be interesting to specify it in the discussion following Theorem 2.2

* In (2) the backward Brownian motion is never introduced. I would have preferred if the authors had sticked with the backward formulation given a positive time element, i.e.
$d x_{T-t} = -f(x_{T-t}, T-t) + g^2(T-t) \nabla \log p_{T-t}(x_{T-t}) + g(T-t) d w_t$, which is also closer to the form that is actually used in the proof.

* It is a bit underwhelming that the authors do not study the influence of $g$ in this paper. The choice of the schedule does matter in practice and is an important topic of investigation in the community.

* There is an error in Theorem 4.1. Indeed after line 246, the equality $\sum_{k=0}^{n-1} \mathbb{P}(X_k \in B_k) = \sum_{k=0}^{n-1} \mathbb{E}_{q_k} \mathbb{1}_k$ is false. In fact, we have $\sum_{k=0}^{n-1} \mathbb{P}(X_k \in B_k) = \sum_{k=0}^{n-1} \mathbb{E}_{p_k} \mathbb{1}_k$. This simplifies the result since we get that $\mathrm{TV}(q_n, \bar{q}_n) \leq \sum_{k=0}^{n-1} \delta_k$ and $\mathrm{TV}(p_n, q_n) \leq D_n + \sum+{k=0}^{n-1} \delta_k$.

* Related to the comment on the mixture of Gaussians : in [1] it is shown that if two distributions $p_0$ and $p_1$ have finite $\chi_2$ divergence and satisfy logarithmic-Sobolev inequalities then their mixture also satisfies a logarithmic Sobolev inequality. Hence, I think that the authors could mitigate their caveat here.

* There are a few typos in the appendix. For example, line 718 $f(x,t) ==$ should be $f(x,t) =$ and below line 706, $p$ should be $p_t$.

* I really enjoyed reading the proof of Theorem C.1. The proof can be decomposed as follows: first the author derive a differential inequality on $\chi^2(p_t|q_t)$. To do so the authors use an extension of the computations of [2] to the non-homogeneous setting. Then, they control the errors arising from this decomposition. In particular, they need to control errors involving $\phi_t = (q_t / p_t) / \mathbb{E}_{p_t}[\phi_t^2]$ (Section C.4). This is done with the Donsker-Varadhan variational theorem. A perturbation error needs to be controlled in Section C.5. The differential inequality then gives rise to a $L_\infty$ control which when combined with the results allowing to go from the $L_2$ error to the $L_\infty$ error concludes the proof. I would have appreciated such a discussion in Section C.1 to give an overview of the rest of the proof.

* The results of Section C.4 heavily rely on the Donsker-Varadhan variational principle. To my knowledge this principle only applies to bounded function (this comes from the dual representation of the Kullback-Leibler divergence). I'm assuming that the result can be extended to any function but could the authors provide a reference for this result?

[1] Schlichting (2019) -- Poincaré and log–Sobolev inequalities for mixtures
[2] Chewi et al. (2021) -- Analysis of Langevin Monte Carlo from Poincare to Log-Sobolev


**Limitations:**

The authors discuss the limitations of their work in the conclusion.
They are aware of the limitations of their framework regarding the use of the logarithmic Sobolev inequality.
I would have appreciated a deeper discussion on the assumption that $\nabla \log p_t$ needs to be Lipschitz.

**Strengths And Weaknesses:**

STRENGTHS:

* The paper is very well-written with clear exposition. The study is also well motivated. Diffusion models are an attractive research area with a lot of empirical results but very few theoretical ones.
* The results cover diffusion models with Brownian motion forward (SMLD) or Ornstein-Ulhenbeck forward (DDPM). They also cover the case of predictor-corrector schemes.
* The approach to prove the convergence of diffusion models is new and likely to be fruitful in the future.

WEAKNESSES:

* Some of the assumptions are too strong. For example proving that a given density satisfies a logarithmic-Sobolev inequality is not trivial, except in easy setting such as strongly convex potentials. For example, to the best of my knowledge, it is not clear that a dissipativity condition suffices to prove some logarithmic Sobolev inequality. The closest condition I could find is the work of [1] which proves that under some dissipativity condition which might depend on the local behavior of the density a logarithmic Sobolev condition is satisfied. Is there a realistic setting in which the density satisfies a logarthmic Sobolev inequality.
* There is one strong assumption throughout this paper, which is that $\nabla \log p_t$ is Lipschitz with a uniform constant $L$. While it seems intuitive that $\log p_t$ should be Lipschitz, proving it seems quite hard.

[1] Cattiaux et al. (2007) -- Lyapunov conditions for logarithmic Sobolev and Super Poincaré inequality

---

> ### Author Response · Authors · 2022-08-02
> **Response to Reviewer VLZN**
>
> Thank you for supporting the paper, catching the typos, and suggesting the
> clarifications. We will fix the issues in the revision. Here are our responses:
>
> It was commented that **some of the assumptions are too strong** or hard to check in practice.  We note that although the **logarithmic-Sobolev inequality (LSI)** is hard to check for real-world data, this assumption is much weaker than those in previous works [1]. Also, as [2] pointed out, the class of measures satisfying LSI is large and is stable under bounded perturbation. We also recall that it has been shown that the dissipativity condition implies LSI (e.g. see Section 4.1 of [1]). As for **the uniform-lipschitz assumption**, it's not trivial to prove it in the LSI case, and hence we leave it as an assumption for now. But one can show the uniform-lipshitzness holds when the target distribution is log-concave (see Lemma 28 in [3]).
>
> Regarding the **error in Theorem 4.1**: Yes, and actually there is an error in the statement of the theorem: condition 1 should read “If $Z_k \in B^c_k$” and “$X_k \in B_k$” should be replaced by “$Z_k \in B_k$” in the proof. Thus the use of Cauchy-Schwarz is necessary to transfer between $X_k$ and $Z_k$. We will fix that in the revision.
>
> Regarding the extension of **Donsker-Varadhan variational principle** to any functions, we note that we can derive the inequality for general functions by approximation with bounded functions (see Lemma 4.10 in [4]).
>
> Regarding **the comment on the mixture of Gaussians**, thanks for providing the reference and we will include this in future work or possibly in the revision.
>
> As for the constant **$C_T$ in Theorem 2.1**, it stands for the LSI constant of $p_T$; and the \textbf{geometric sequence $\{\sigma_i\}$} is specified in the proof sketch of Theorem 2.2 (line 272). Thank you for pointing these out, and we will make them more clear in the revision.
>
> Regarding **the influence of $g$**, although the choice of the schedule does matter in practice, our theoretical analysis shows that what really matters is the integral $\int_0^t g(s)^2 ds$. This means that different choices of $g$ is only a rescaling of time, i.e. for different $g$ and $\tilde g$, we can always choose proper running time $T$ and $\tilde T$, such that $\int_0^T g(s)^2 ds = \int_0^ {\tilde {T}} \tilde g(s)^2 ds$. Although it seems that choosing large $g(t)$ could reduce the total running time $T$, our analysis (e.g. Lemma C.15, line 849) shows that we need the time step-size $h$ to be $O(1/g(T)^2)$ and hence the total computation, which is roughly $O(T/h)$, doesn't change too much. As a result, theoretically we don't see too many differences of choosing different $g$.
>
> As for the **introduction of backward Brownian Motion**, sorry for the lack of that. For a fixed interval $[0,T]$, we say $ w_{t} $ is a backward Brownian Motion if $w_{T-t}$ is a Brownian Motion on $[0,T]$. We will include this in the revision.
>
>
> [1] Adam Block, Youssef Mroueh, and Alexander Rakhlin. “Generative modeling with denoising auto-encoders and Langevin sampling”. In: arXiv preprint arXiv:2002.00107 (2020)
>
> [2] Santosh Vempala and Andre Wibisono. “Rapid convergence of the unadjusted langevin algorithm: Isoperimetry suffices”. In: Advances in neural information processing systems 32 (2019), pp. 8094–8106.
>
> [3] Lee H, Pabbaraju C, Sevekari A, et al. Universal Approximation for Log-concave Distributions using Well-conditioned Normalizing Flows[J]. arXiv preprint arXiv:2107.02951, 2021.
>
> [4] Van Handel R. Probability in high dimension[R]. PRINCETON UNIV NJ, 2014.

---

> > ### Comment · Reviewer_VLZN · 2022-08-07
> > **Response**
> >
> > Thanks a lot for your rebuttal. I just have a few remarks.
> >
> > Thank you for pointing out the fact that LSI can be checked under dissipativity conditions. Digging up the references I think that the correct result to cite here would be [1, Theorem 1.9]. I slightly disagree that this result implies that LSI is satisfied under dissipativity conditions. Indeed, one need the extra assumption that the Hessian satisfies a global lower bound. Hence, local sharp maximizer can have a dramatic effect on the LSI constant. However, I thank the authors for pointing me to this line of work that I was not aware of.
> >
> > I also think that the authors missed one of my comments: "There is one strong assumption throughout this paper, which is that $\nabla \log p_t$ is Lipschitz with a uniform constant . While it seems intuitive that $\nabla \log p_t$ should be Lipschitz, proving it seems quite hard."
> > Could the authors comment further on this?
> >
> > [1] Cattiaux et al. (2008) -- A note on Talagrand transportation inequality and LSI

---

> > > ### Author Response · Authors · 2022-08-08
> > > **Lipschitzness of gradient**
> > >
> > > You are correct that dissipativity is only known to imply a log-Sobolev inequality under a Hessian bound. We will note this caveat in the revision. As [1] assumes a Hessian bound, we note that our results are strictly more general.
> > >
> > > To expand on our comment on the uniform-lipschitz assumption (Lipschitzness of $\nabla \log p_t$), it is unclear to us whether Lipschitzness of $\nabla \log p_0$ implies Lipschitzness for all $t$ (and we suspect it might not be true). Hence we leave it as an assumption for now, which has to be checked on a case-by-case basis for the distribution of interest.
> > > We note that this assumption appears in [1] as well, under Assumption 2, and helps ensure a unique strong solution to the Langevin diffusion. (Note that the distributions $p_t$ for DDPM and SMLD differ by a scaling by the remark after Theorem C.2, so it suffices to assume Lipschitzness for SMLD, where $p_t=p_0*N(0,t)$.)
> > > One special case where one can prove Lipschitzness for all $t$ is when $p_0$ is log-concave (see Lemma 28 in [3]).

---

### Official Review · Reviewer_MwKa · 2022-07-09

**Rating:** 7
**Confidence:** 5
**Soundness:** 4 excellent
**Presentation:** 3 good
**Contribution:** 4 excellent

**Summary:**

This paper gives a new theoretical framework to analyze the complexity of score-based generative modeling methods, from Langevin dynamics (including Annealed Langevin dynamics, [SE19]) to forward/backward processes such as DDPM [HJA20] (including the predictor/corrector improvement suggested in [Son+20b]). The authors develop a good set/bad set approach to control the $\chi_2$ divergence of the target distribution $p$ wrt a distribution $\bar{q}$, which is arbitrarily close to the estimated distribution $q$ in total variation distance. Their analysis relies on assumptions commonly used in this field: L2-accuracy of $\nabla \ln q$ wrt $\nabla \ln p$, Log-Sobolev property for $p$ and smoothness properties of $\ln p$. In contrast to previous works (especially [De+21]), the error term in their bound does not grow exponentially in time and does not suffer from the curse of dimensionality. This is the first known analysis which provides a polynomial dependence on the parameters of the problem.

**Questions:**


- I suggest the authors to move Assumptions 1 and 2 outside of Section 2 (preferably before), since they are used in Section 3 again. More precisely, I advise the authors to introduce these assumptions out of their analysis sections.
- I suggest the authors to move Appendix E at the beginning of the supplement and to recall the definition of $\chi_2$-divergence and total variation distance in this section.
- The authors should recall the notations $p_t$, $q_t$ in Theorem 4.3 when stating the result.
- In the main paper and the supplements, $\tilde{q}$ often stands for $\bar{q}$, which leads to some confusion.


**Limitations:**

- I think there is an error in Theorem 2.2 in the polynomial dependence of $d$; it would rather be $d^{3.25}$ than $d^{2.5}$.

**Strengths And Weaknesses:**

Strengths:
- High quality paper, very well-written, the results are well presented and clear, the authors provide intuition and interpretation for their results, the theoretical framework is solid.
- The paper is well organized and didactic: Background & related work; Analysis of Langevin Dynamics; Analysis of Forward/Backward SDE; General approach and main technical lemmas.
- Their convergence bound is better than the bounds stated in state-of-the-art theorems, under close assumptions.
- Their good set/bad set approach is interesting and convincing.

Weaknesses:

- In my opinion, there is an error in Theorem 4.1 (line 246, the last term in the inequality is the expectation wrt to $p_k$ rather than $q_k$), which leads to a suboptimal bound for TV($q_n$, $\bar{q_n}$) in the paper. Indeed, the actual bound should be $\sum_{k=0}^{n-1}\delta_k$, which is better than the one derived by the authors (since $\delta_k \leq 1$, $\delta_k \leq 1+D_k$). This suboptimal bound seems to make the proof harder than it is. However, from my understanding, this suboptimal bound does not change the conclusions of the paper.

---

> ### Author Response · Authors · 2022-08-02
> **Response to Reviewer MwKa**
>
> Thank you very much for the constructive comments and suggestions. Here are our responses to the issues you pointed out:
>
> Regarding the **error in Theorem 4.1**: Yes, and actually there is an error in the statement of the theorem: in condition 1, "If $X_k\in B^c$ should be replaced by “If $Z_k \in B^c_k$”, and in the proof, “$X_k \in B_k$” should be replaced by “$Z_k \in B_k$”. Thus the use of Cauchy-Schwarz is necessary to transfer between $X_k$ and $Z_k$. Thank you for pointing this out and we will fix that in the revision.
>
> As for the **abuse of $\tilde{q}$ and $\bar{q}$**, we apologize for the confusion and we will correct this and also go over the manuscript carefully in the revision
> for other possible minor mistakes.
>
> For the lack of **definition of $p_t$ and $q_t$ in the statement of Theorem 4.3**, we will recall that in the revision.
>
> It was stated that there is an error in Theorem 2.2 in the polynomial dependence of $d$. The dependence is in fact $d^{3.25}$ instead of $d^{2.5}$. We will fix that in the revision.
>
> Regarding the suggestions about certain assumptions and Appendix, thanks for the suggestions; we will take them into account in the revision.

---

### Official Review · Reviewer_JAKh · 2022-07-11

**Rating:** 7
**Confidence:** 2
**Soundness:** 3 good
**Presentation:** 3 good
**Contribution:** 3 good

**Summary:**

This work provides the first polynomial convergence guarantee for score-based geneative models. The theoreical results supports the current progress in score-based generative models that use an annealed procedure with the combination of predictor-corrector during training.

**Questions:**

1. Does the claim in Theorem 4.2 means the convergence rate of DDPM/SMLD can be further improved when combined with a corrector? If so, some simulation results might be helpful to verify this claim.

2. Does the improvement on the convergence result only apply to TV distance? What about Wasserstein distance, etc?

**Limitations:**

The limitations of the assumptions made in the paper are discussed in the last section.

**Strengths And Weaknesses:**

Strengths:

1. The work is the first to provide polynomial complexity in the convergence of score-based generative models to my best knowledge. Polynomial dependency on time, dimension and smoothness of the distribution is a significant improvement compared to existing works.

2. The presentation of the paper is clear. The comparison w.r.t. to previous results in the introduction shows the significant improvement in the convergence rate analysis. Examples of exisiting approaches (SMLD, DDPM) also support those design choices.

3. Key ideas in the proofs are introduced in section 4 without goinging into the details, which makes it easier to follow.

Weaknesses:

1. The relationship between the predictor-corrector method and the current training approach of score-based generative models (for example, DDPM) is not discussed in a very comprehensive way. See Questions 1.

---

> ### Author Response · Authors · 2022-08-02
> **Response to Reviewer JAKh**
>
> Thank you very much for your very encouraging comments and feedback. The detailed response to each point is as follows:
>
> **Re: "Does the claim in Theorem 4.2 means the convergence rate of DDPM/SMLD can be further improved when combined with a corrector? If so, some simulation results might be helpful to verify this claim."** Yes. As for the simulation results, please see Section 4.2 of [1] for details. It shows that for all predictors, adding one corrector step for each predictor step doubles computation but always improves sample quality.
>
> **Re: "Does the improvement on the convergence result only apply to TV distance? What about Wasserstein distance, etc?"** The main idea of our analysis framework is to convert a $L^2$ error guarantee to a $L^\infty$ error guarantee by excluding a bad set, formalized in Theorem 4.1. However, this bad set idea only applies to TV distance, and the convergence of LMC Wasserstein distance with $L^2$-accurate score function is still an open question.
>
> [1] Yang Song, Jascha Sohl-Dickstein, Diederik P Kingma, Abhishek Kumar, Stefano Ermon, and Ben Poole. “Score-Based Generative Modeling through Stochastic Differential Equations”. In: International Conference on Learning Representations. 2020.

---

> > ### Comment · Reviewer_JAKh · 2022-08-07
> > **Response**
> >
> > Thanks the authors for the update. I keep my score unchanged.

---

### Official Review · Reviewer_8Jhx · 2022-07-12

**Rating:** 9
**Confidence:** 3
**Soundness:** 4 excellent
**Presentation:** 4 excellent
**Contribution:** 4 excellent

**Summary:**

This work provides a rigorous theoretical analysis on score-based generative modeling, giving the first polynomial convergence guarantees for multiple sampling methods. It is proved that under suitable assumptions, annealed Langevin dynamics converge better than Langevin dynamics (without the need of warm starts), and predictor-corrector converges faster than either predictor-only or corrector-only approaches.

**Questions:**

As a byproduct of the proof for Theorem 2.2, authors have a way to choose $M$ and the sequence $\sigma_1 < \cdots < \sigma_M$ for the convenience of theoretical derivation. I wonder whether choosing noise levels in this way has any empirical significance. Can you outperform existing noise schemes with the one used in your theoretical analysis?

**Limitations:**

Authors have a very comprehensive list of future research directions in conclusion, which is an implicit discussion on limitations. This is a purely theoretical work and I do not see any immediate negative societal impacts.

**Strengths And Weaknesses:**

This is an excellent work and I can see many strengths:

1. Theoretical results in this work have weaker assumptions than prior arts. It does not require $L^\infty$-accurate score function ($L^2(p)$-accurate is sufficient), and assumes no dissipativity. The rates do not have exponential dependency on dimensionality, a significant improvement over existing results.

2. The score estimate has no need to be conservative. This work therefore provides a solid theoretical footing for the common trick in practical applications that employs unconstrained neural networks to directly model and estimate the score function.

3. Theorem 2.1 & 2.2 provide a concrete justification on the importance of annealed Langevin dynamics in terms of avoiding warm starts. Theorem 2.2, 3.1 & 3.2 additionally shows the benefit of predictor-corrector sampling compared to using either part alone. Both are significant theoretical advancements that provide solid understanding for previous empirically driven methods in score-based generative modeling.

4. Very clear writing. A comprehensive conclusion on future directions.

There are no major weaknesses. A few typos need to be corrected in the revision:

line 53: duplicated "the"

line 171: "corrector step" -> "predictor step"

line 242: "to the be" -> "to be the"

Equation below line 246: missing $\sum_{k=0}^{n-1}$ after $\leq$ on the second row.

---

> ### Author Response · Authors · 2022-08-02
> **Response to Reviewer 8Jhx**
>
> Thanks a lot for your very encouraging comments and feedback. For the typos pointed out, we will correct them and also go over the manuscript carefully in the revision for other possible minor mistakes.
>
> **Re: "I wonder whether choosing noise levels in this way has any empirical significance."**
> We note that in practice, the sequence is indeed chosen as a geometric sequence: [1] recommends choosing $\{\sigma_i\}$ as a geometric sequence based on empirical observations (see Section 3). However, we don't make any claims about whether our suggested spacing is empirically optimal, as this choice is mainly for the purpose of theoretical analysis.
>
> [1] Yang Song and Stefano Ermon. “Improved techniques for training score-based generative models”. In: arXiv preprint arXiv:2006.09011 (2020).

---

> > ### Comment · Reviewer_8Jhx · 2022-08-09
> > **Thanks for the response**
> >
> > Thanks for your answer. My question was simply out of curiosity and I agree that a full investigation is out of the scope of this paper.

---

### Official Review · Reviewer_ru9p · 2022-07-15

**Rating:** 6
**Confidence:** 2
**Soundness:** 2 fair
**Presentation:** 3 good
**Contribution:** 3 good

**Summary:**

Score-based generative modeling (SGM) refers to an increasingly popular class of generative models, which implicitly learn a target distribution by using diffusion processes that gradually add noise to the observed samples. An important component of such models is the score function, namely the gradient of the log probability density. This work proposes a theoretical analysis on the convergence of SGM when the score function is estimated, assuming the induced approximation error is bounded in $L^2$ (see Assumption 2.2). The authors motivate their study by pointing out that the literature on that topic exhibit convergence rates that scale exponentially in the dimension or the running time. The main contribution of this paper is then the derivation of convergence rates with polynomial complexity in the parameters of the problem, under a log-Sobolev condition on the data distribution. The contributions of this paper are theoretical results, as explained more precisely below.

1) The error between the target probability density $p$ and the one returned by a SGD based on Langevin diffusion is bounded (Theorem 2.1). As discussed by the authors, the bound in Theorem 2.1 emphasizes the importance of a "warm start" and motivates the study of SGD with annealed Langevin dynamics (Algorithm 1). The error bound is refined for the latter approach in Theorem 2.2.

2) Prior work demonstrated that running the diffusion process in reverse can help improve the empirical performance. The authors thus derive the error bound for denoising diffusion probabilistic modeling (DDPM), an existing generative model based on a reverse-time diffusion process (Theorem 3.1). They also show that using a "predictor-corrector" generative model, by combining DDPM steps ("predictor" steps) with Langevin diffusion steps ("corrector" steps), yield to an error bound with improved dependence on the parameters (Theorem 3.2).

Note that the four main error bounds (Theorems 2.1, 2.2, 3.1, 3.2) are in TV distance and hold under specific assumptions on the target $p$ and the estimate score function $s$ (Assumptions 1 and 2): in particular, $p$ must satisfy a log-Sobolev inequality, the score function is $L$-Lipschitz, and $s$ is bounded in $L^2$. The proofs of these results are detailed in the appendix and summarized in the main document (Section 4). The authors conclude by pointing out the limitations of their analysis (Section 5).

**Questions:**

For now, what prevents me from giving a higher score is, as explained above, the lack of illustration on the main contributions: I encourage the authors to,
1) If possible, relate their theoretical bounds with existing empirical observations reported in the literature,
2) Discuss whether their proof sketches (Section 4) are original to better emphasize their contributions,
3) Discuss if the theoretical framework can foster interesting follow-up work, given that the core assumptions seem unrealistic (according to Section 5),
4) Clarify the impact of the main parameters on the bounds. In particular, if I am not mistaken, the rates do not seem to avoid the curse of dimensionality mentioned in the introduction: while the dependence on $d$ is not exponential in the derived bounds, one can still observe that $d$ highly impacts the speed and quality of the results (for example, see the accuracy $\varepsilon$ in all Theorems, or the length of the sequence $M$ in Theorem 2.2).

Minor comments:

Typos: l.42 "an denoising", l.242: "the be"

l.118: the definition of $W_t$ is missing

l.146-150: I am confused about the interpretations: without warm start, $K_{\chi}^2$ can be arbitrarily large, therefore $\varepsilon$ is small; then, why is the error on $s$ not guaranteed to be small on average (l.147-148)?

l.235: $\mathcal{F}_n$ should be defined

l.246: The sum over $k$ seems to be missing in the derivations, when upper-bounding $\sum_{k=0}^{n-1} E_{q_k} 1_{B_k}$

l.265: Please give more explanations on why $(1+\chi^2(\bar{q}_{kh} || p))^{1/2}$ can be bounded that way. Theorem 4.2 is used but it seems like some factors are missing.

**Limitations:**

The limitations of this work are clearly addressed in Section 5. The potential negative societal impact of the contributions is not discussed.


**Strengths And Weaknesses:**

Score-based generative models have attracted significant interest in the machine learning community over the past few years, but some successful approaches have been proposed very recently and lack theoretical guarantees. Deriving the convergence rate and bounding the error made by these models according to the parameters of the problem is important, since it helps identify the setups for which such algorithms will perform well or fail in practice. In that sense, the problem addressed by this paper is highly relevant for NeurIPS.

I found the paper very well written and easy to follow: the motivations are clear, prior work is adequately discussed, and the authors put some effort in explaining their main theorems and their proofs from a high level perspective. In light of the discussions and interpretations provided throughout the paper, the contributions of this work seem sound to me; but due to a lack of time and expertise on SGD, I was unable to thoroughly check the proofs in the appendix.

In my opinion, this work would significantly benefit from more illustration on the derived rates: I am not familiar with related work and after reading this paper, it is still unclear to me how original the proof techniques are and whether they are consistent with empirical analysis in previous work. I also listed some questions in the next section of my review. Besides, this paper does not contain any experiments, although this could have been useful to illustrate and confirm the theoretical contributions.

On a related note, the authors discuss the limitations of their work in their conclusion, which I appreciated – that being said, it highlights that the assumptions of the theorems do not reflect well what happens in practice, so there is still an important gap between this study and the practice. Specifically, the central assumptions seem too restrictive: bounding the accuracy of the estimated score function in $L^2$ is described as "a strong condition in practice that seems unlikely to be satisfied (and difficult to check) when learning complex distributions" (l.326-328) and it is unclear "when we can find such an estimate" (l.330); the log-Sobolev condition on the target $p$ implies that $p$ is unimodal, which strongly limits the scope of the theoretical analysis. Hence, I am wondering to what extent the proposed framework is realistic and useful.

---

> ### Author Response · Authors · 2022-08-02
> **Response**
>
> Thanks for your insightful comments and careful correction. We will correct the typos in the revision. We respond to the comments in the review:
>
> **Relation with empirical observations**: Our theoretical results are qualitatively consistent with empirical observations in prior works:
>
> * Using a corrector can improve the convergence rate of DDPM/SMLD: Song et al. give a side-by-side comparison of predictor-only (P) and predictor-corrector (PC) methods in Section 4.2 of [3] (see Table 1) and show that the PC methods outperform the P methods.
>
> * Necessity of using annealed Langevin dynamics when we have an estimated score: In Langevin dynamics with estimated score, a larger $K_\chi$ (colder start) necessitates a more accurate score estimate, and this is potentially a serious problem when $K_\chi$ is exponential from a cold start. Annealing avoids this problem both theoretically and empirically.
>
> **Originality of proofs**.
> The main innovation in our analysis is to convert a $L^2$ error guarantee to a $L^\infty$ error guarantee by excluding a bad set, formalized in Theorem 4.1. This 'bad set' idea is original, while for the $L^\infty$ error analysis, we use existing techniques from [1]. Additionally, we note that the analysis for DDPM/SMLD is technically more involved than prior work due to the fact that we are comparing the empirical distribution to a distribution $p_t$ that is itself evolving over time, rather than stationary. Upon differentiating $\chi^2(q_t||p_t)$, this results in more terms that must be bounded individually (see proof of Lemma C.3).
>
> **Flexibility of the framework to accommodate follow-up work**. We believe our framework can be used to analyze convergence beyond distributions satisfying a log-Sobolev inequality, as suggested in our conclusion. In the proof of Theorem 3.1 (Predictor with $L^2$-accurate score estimate), the main use of the LSI condition is to bound $KL(\psi_t q_t||p_t)$ in Lemma C.8. More generally, any bound of the form $KL(\psi_t q_t||p_t)\le \frac{A}{\chi^2(q_t||p_t)+1}\mathscr E_{p_t}(q_t/p_t)+B$ plus subgaussianity of $p_t$ (used in Lemma C.9) allows to the rest of the proof to go through. In this way, we are currently working on deriving similar bounds for some multimodal distributions.
>
> We also emphasize that although the logarithmic-Sobolev inequality (LSI) is hard to check for real-world data, this
> assumption is standard for analysis of Langevin Monte Carlo, and weaker than previous works on SGM's; for example, [2] assumes strong dissipativity, which is a stronger condition. The accuracy of score-estimation is an interesting question
> that is beyond the scope of this paper, and we refer to [2] for details.
>
> **Impact of the main parameters on the bounds.** By "avoiding the curse of dimensionality", we mean that the necessary settings of parameters to achieve error $\epsilon$ do not depend exponentially on $d$.
> Our dependence on $d$ is only polynomial instead of exponential and hence we avoid the curse of dimensionality. Note that the exponents in our theorems can likely be improved, but we leave the problem of determining optimal dependencies to future work. While we don't claim exact dependencies, we can still obtain qualitative information. For example, a larger $K_\chi$ necessitates a more accurate score estimate.
>
> As a concrete illustration of our parameter choices, in Theorem 3.2 we show that in order to be $O(1/d^{1/3})$-close to the target distribution in the sense of TV-distance, we first need out estimated score to be $O(1/d^{7/3})$-accurate in $L^2(p)$ sense and run the predictor-corrector algorithm for $T=O(\ln(d))$. Here we omit the dependence of other parameters. We will make the impact of the main parameters more clear in the revision.
>
> **Interpretations after Theorem 2.1 (line 146-150)**: If $K_{\chi}^2$ is large, we need $\epsilon$ (the score estimation error) to be small (see equation (2)) to guarantee the accuracy. By our comment we mean that for a *fixed* value of $\epsilon$, when $K_\chi$ is larger, the score estimate along the algorithm is not guaranteed to be as small on average, hence the necessity of a smaller $\epsilon$.
>
> **Why can $(1+\chi^2(\bar{q}_{kh}||p))^{1/2}$ can be bounded that way?** This is a typo and the last inequality should read '$\leq \sum_{k=0}^{n-1} (\exp(-\frac{kh}{8C_{LS}})\chi^2(q_0||p)^{1/2} + 1)\delta^{1/2}$'.
>
> [1] Sinho Chewi, Murat A Erdogdu, Mufan Bill Li, Ruoqi Shen, and Matthew Zhang. “Analysis of Langevin Monte Carlo from Poincare to Log-Sobolev”. In: ´ arXiv preprint arXiv:2112.12662 (2021)
>
> [2] Adam Block, Youssef Mroueh, and Alexander Rakhlin. “Generative modeling with denoising auto-
> encoders and Langevin sampling”. In: arXiv preprint arXiv:2002.00107 (2020)
>
> [3] Yang Song, Jascha Sohl-Dickstein, Diederik P Kingma, Abhishek Kumar, Stefano Ermon, and Ben
> Poole. “Score-Based Generative Modeling through Stochastic Differential Equations”. In: International
> Conference on Learning Representations. 2020.

---

### Author Response · Authors · 2022-08-02
**Sincere Thanks**

We thank all the reviewers for carefully reading our work and for the helpful and constructive comments.
Please see below for separate replies. We will revise the manuscript accordingly.

---

### Meta-Review · Area_Chair_zNkb · 2022-08-24

**Recommendation:** Accept
**Confidence:** Certain

**Metareview:**

The reviewers and I agree that the contributions of the paper are of interest and useful addition to the literature. Therefore, I recommend accepting the paper.

Please consider the reviewers' comments when preparing the camera-ready version.


**Award:**

No

---

### Decision · Program_Chairs · 2022-09-14

Accept